# On the Convergence of Shallow Neural Network Training with Randomly Masked Neurons

**Fangshuo Liao**                                                    *Fangshuo.Liao@rice.edu*
*Department of Computer Science*
*Rice University*

**Anastasios Kyrillidis**                                            *anastasios@rice.edu*
*Department of Computer Science*
*Rice University*

**Reviewed on OpenReview:** *https: // openreview. net/ forum? id= e7mYYMSyZH*

## Abstract

With the motive of training all the parameters of a neural network, we study why and when one can achieve this by iteratively creating, training, and combining randomly selected subnetworks. Such scenarios have either implicitly or explicitly emerged in the recent literature: see e.g., the Dropout family of regularization techniques, or some distributed ML training protocols that reduce communication/computation complexities, such as the Independent Subnet Training protocol. While these methods are studied empirically and utilized in practice, they often enjoy partial or no theoretical support, especially when applied on neural network-based objectives.

In this manuscript, our focus is on overparameterized single hidden layer neural networks with ReLU activations in the lazy training regime. By carefully analyzing $i$) the subnetworks' neural tangent kernel, $ii$) the surrogate functions' gradient, and $iii$) how we sample and combine the surrogate functions, we prove linear convergence rate of the training error –up to a neighborhood around the optimal point– for an overparameterized single-hidden layer perceptron with a regression loss. Our analysis reveals a dependency of the size of the neighborhood around the optimal point on the number of surrogate models and the number of local training steps for each selected subnetwork. Moreover, the considered framework generalizes and provides new insights on dropout training, multi-sample dropout training, as well as Independent Subnet Training; for each case, we provide convergence results as corollaries of our main theorem.

## 1 Introduction

Overparameterized neural networks have led to both unexpected empirical success in deep learning *(Zhang et al., 2021; Goodfellow et al., 2016; Arpit et al., 2017; Recht et al., 2019; Toneva et al., 2018)*, and new techniques in analyzing neural network training *(Kawaguchi et al., 2017; Bartlett et al., 2017; Neyshabur et al., 2017; Golowich et al., 2018; Liang et al., 2019; Arora et al., 2018; Dziugaite & Roy, 2017; Neyshabur et al., 2018; Zhou et al., 2018; Soudry et al., 2018; Shah et al., 2020; Belkin et al., 2019; 2018; Feldman, 2020; Ma et al., 2018; Spigler et al., 2019; Belkin, 2021; Bartlett et al., 2021; Jacot et al., 2018)*. While theoretical work in this field has led to a diverse set of new overparameterized neural network architectures *(Frei et al., 2020; Fang et al., 2021; Lu et al., 2020; Huang et al., 2020; Allen-Zhu et al., 2019a; Gu et al., 2020; Cao et al., 2020)* and training algorithms *(Du et al., 2018; Zou et al., 2020; Soltanolkotabi et al., 2018; Oymak & Soltanolkotabi, 2019; Li et al., 2020; Oymak & Soltanolkotabi, 2020)*, most efforts fall under the following scenario: in each iteration, we perform a gradient-based update that involves all parameters of the neural network in both the forward and backward propagation. Yet, advances in regularization techniques

*(Srivastava et al., 2014; Wan et al., 2013; Gal & Ghahramani, 2016; Courbariaux et al., 2015; Labach et al., 2019)*, computationally-efficient *(Shazeer et al., 2017; Fedus et al., 2021; Lepikhin et al., 2020; LeJeune et al., 2020; Yao et al., 2021; Yu et al., 2018; Mohtashami et al., 2021; Yuan et al., 2020; Dun et al., 2021; Wolfe et al., 2021)* and communication-efficient distributed training methods *(Vogels et al., 2019; Wang et al., 2021; Yuan et al., 2020)* favor a different narrative: one would –explicitly or implicitly– train smaller and randomly-selected models within a large model, iteratively. This brings up the following question:

> "*Can one meaningfully train an overparameterized ML model by iteratively training and combining together smaller versions of it?*"

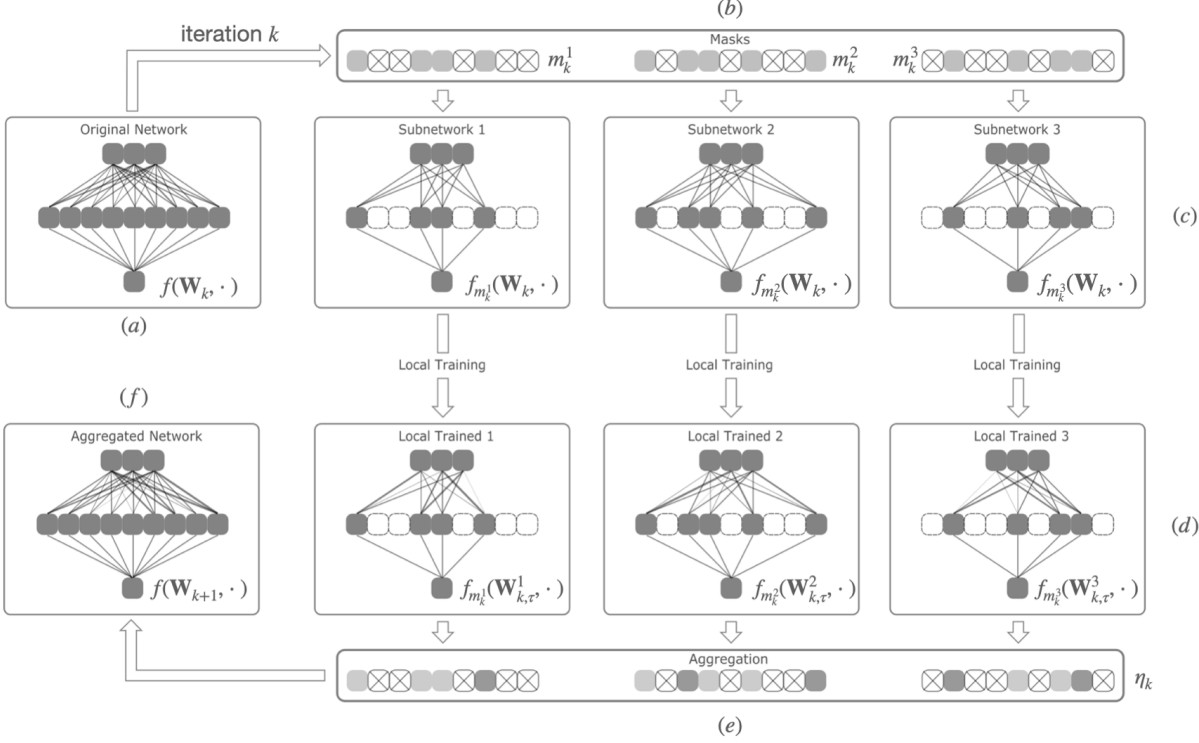

Figure 1: Training a single hidden-layer perceptron using multiple randomly masked subnetworks. Here, $f(\mathbf{W}, \cdot)$ denotes the full model with $\mathbf{W}$ parameters, and $f_{\mathbf{m}_k^l}(\mathbf{W}, \cdot)$ denotes the surrogate model (subnetwork) with only active neurons as dictated by the mask $\mathbf{m}_k^l$ at the $k$-th iteration for subnetwork $l$. Moreover, $\mathbf{W}_k$ denotes the parameter at the start of the iteration, while $\mathbf{W}_{k,\tau}^l$ is the trained parameter of subnetwork $l$.

This question closely relates to multiple existing training algorithms, as we discuss below; our goal is to work towards a unified training scheme, and seek for rigorous theoretical analysis of such a framework. Focusing on this objective based on shallow feedforward neural networks, we provide a positive answer, accompanied with theoretical guarantees that are supported by observations in practical scenarios.

To be more specific, the training scheme we consider is depicted in Figure 1. Given a dense neural network, as in Fig.1($a$), we sample masks within one training step; see Fig.1($b$). Each of these masks deactivates a subset of the neurons in the original network's hidden layer. In a way, each mask defines a *surrogate model*, as shown in Fig.1($c$), based on the original network, leading to a collection of *subnetworks*. These surrogate subnetworks independently update their own parameters (possibly on different data shards), by performing (stochastic) gradient descent ((S)GD) steps. Lastly, we aggregate the parameters of the independently trained subnetworks (Fig.1($d$)) to update the weights of the original network, before the next iteration starts; see Fig.1($e$). Note that multiple masks could share active neurons. When aggregating the updates, we take the weighted sum of the updated parameters across all subnetworks, with the aggregation weights computed on the masks of the current iteration.

We mathematically illustrate the difference between traditional training (first expression below) and the considered methodology (second expression below):

$$\mathbf{W}_{k+1} = \texttt{(S)GD}\left(f, \mathbf{W}_k, \tau\right) \qquad \text{vs.}$$

$$\mathbf{W}_{k+1} = \texttt{Reassemble}\left(\texttt{(S)GD}\left(f_{\mathbf{m}_k^1}, \mathbf{W}_k^1, \tau\right), \texttt{(S)GD}\left(f_{\mathbf{m}_k^2}, \mathbf{W}_k^2, \tau\right), \cdots, \texttt{(S)GD}\left(f_{\mathbf{m}_k^p}, \mathbf{W}_k^p, \tau\right)\right)$$

Here, the acronym $\texttt{(S)GD}\left(f, \mathbf{W}, \tau\right)$ indicates the application of (S)GD on function $f$ for $\tau$ iterations, starting from initial parameters $\mathbf{W}$. Consequently, $\texttt{(S)GD}\left(f_{\mathbf{m}_k^l}, \mathbf{W}_k^l, \tau\right)$ indicates the application of (S)GD for $\tau$ iterations on the surrogate function $f_{\mathbf{m}_k^l}$, based on the mask $\mathbf{m}_k^l$ and using only the subset of parameters $\mathbf{W}_k^l$. The function $\texttt{Reassemble}$ involves both aggregation and reassembly of the whole model $\mathbf{W}_{k+1}$.

In this work, we perform a theoretical analysis of this framework, based on a single-hidden layer perceptron with ReLU activations. This is a non-trivial, non-convex setting, that has been used extensively in studying the behavior of training algorithms on neural networks *(Du et al., 2018; Zou et al., 2020; Soltanolkotabi et al., 2018; Oymak & Soltanolkotabi, 2019; Li et al., 2020; Oymak & Soltanolkotabi, 2020; Song & Yang, 2020; Ji & Telgarsky, 2020; Mianjy & Arora, 2020)*.

**Challenges.** Much work has been devoted to analyzing the convergence of neural networks based on the Neural Tangent Kernel (NTK) perspective *(Jacot et al., 2018)*; see the Related Works section below. The literature in this direction notice that the NTK remains roughly stable throughout training. Therefore, the neural network output can be approximated well by the linearization defined by the NTK. Yet, training with randomly masked neurons poses additional challenges: *i)* With a randomly generated mask, the NTK changes even with the same set of weights, leading to more instability of the kernel; *ii)* the gradient of the subnetworks introduces both randomness and bias towards optimizing the loss of the full network; and, *iii)* the non-linear activation makes the aggregated network function no longer a linear combination of the subnetwork functions. The three challenges complicate the analysis, driving us to treat the NTK, gradient, and combined network function with special care. We will tackle these difficulties in the proof of the theorems.

**Motivation and connection to existing methods.** The study of partial models/subnetworks that reside in a large dense network have drawn increasing attention.

*Dropout regularization.* Dropout *(Srivastava et al., 2014; Wan et al., 2013; Gal & Ghahramani, 2016; Courbariaux et al., 2015)* is a widely-accepted technique against overfitting in deep learning. In each training step, a random mask is generated from some pre-defined distribution, and used to mask-out part of the neurons in the neural network. Later variants of dropout include the drop-connect *(Wan et al., 2013)*, multi-sample dropout *(Inoue, 2019)*, Gaussian dropout *(Wang & Manning, 2013)*, and the variational dropout *(Kingma et al., 2015)*. Here, we restrict our attention to the vanilla dropout, and the multi-sample dropout. The vanilla dropout corresponds to our framework, if in the latter we sample only one mask per iteration, and let the subnetwork perform only one gradient descent update. The multi-sample dropout extends the vanilla dropout in that it samples multiple masks per iteration. *For regression tasks, our theoretical result implies convergence guarantees for these two scenarios on a single hidden-layer perceptron.*

*Distributed ML training.* Recent advances in distributed model/parallel training have led to variants of distributed gradient descent protocols *(Mcdonald et al., 2009; Zinkevich et al., 2010; Zhang & Ré, 2014; Zhang et al., 2016)*. Yet, all training parameters are updated per outer step, which could be computationally and communication inefficient, especially in cases of high communication costs per round. The *Independent Subnetwork Training (IST)* protocol *(Yuan et al., 2020)* goes one step further: IST splits the model vertically, where each machine contains all layers of the neural network, but only with a (non-overlapping) subset of neurons being active in each layer. Multiple local SGD steps can be performed without the workers having to communicate. Methods in this line of work achieves higher communication efficiency and accuracy that is comparable to centralized training.*(Wolfe et al., 2021; Dun et al., 2021; Yuan et al., 2020) Yet, the theoretical understanding of IST is currently missing. Our theoretical result implies convergence guarantees for IST for a single hidden-layer perceptron under the simplified assumption that every worker has full data access, and provides insights on how the number of compute nodes affects the performance of the overall protocol.*

**Contributions.** The present training framework naturally generalizes the approaches above. Yet, current literature –more often than not– omits any theoretical understanding for these scenarios, even for the case of shallow MLPs. While handling multiple layers is a more desirable scenario (and is, indeed, considered as future work), our presented theory illustrates how training and combining multiple randomly masked surrogate models behaves. Our findings can be summarized as follows:

- We provide convergence rate guarantees for $i$) dropout regularization *(Srivastava et al., 2014)*, $ii$) multi-sample dropout *(Inoue, 2019)*, $iii$) and multi-worker IST *(Yuan et al., 2020)*, given a regression task on a single-hidden layer perceptron.

- We show that the NTK of surrogate models stays close to the infinite width NTK, thus being positive definite. Consequently, our work shows that training over surrogate models still enjoys linear convergence.

- For subnetworks defined by Bernoulli masks with a fixed distribution parameter, we show that aggregated gradient in the first local step is a biased estimator of the desirable gradient of the whole network, with the bias term decreasing as the number of subnetworks grows. Moreover, all aggregated gradients during local training stays close to the aggregated gradient of the first local step. This finding leads to linear convergence of the above training framework with an error term under Bernoulli masks.

- For masks sampled from categorical distribution, we provide tight bounds $i$) on the average loss increase, when sampling a subnetwork from the whole network; $ii$) on the loss decrease, when the independently trained subnetworks are combined into the whole model. This finding leads to linear convergence with a slightly different error term than the Bernoulli mask scenario.

Summarizing the contributions above, the main objective of our work is to provide theoretical support for the following statement:

**Main statement** (*Informal*)**.** *Consider the training scheme shown in Figure 1 and described precisely in Algorithm 1. If the masks are generated from a Bernoulli distribution or categorical distribution, under sufficiently large over-parameterization coefficient, and sufficiently small learning rate, training the large model via surrogate subnetworks still converges linearly, up to an neighborhood around the optimal point.*

## 2 Related Works

**Convergence of Neural Network Training.** Recent study on the properties of over-parameterized neural networks enabled their training error analysis. The NTK-based analysis studies the dynamics of the parameters in the so-called kernel regime under a particular scaling option *(Jacot et al., 2018; Du et al., 2018; Oymak & Soltanolkotabi, 2020; Song & Yang, 2020; Ji & Telgarsky, 2020; Su & Yang, 2019; Arora et al., 2019; Mianjy & Arora, 2020; Huang et al., 2021)*. NTKs can be viewed as the reproducing kernels of the function space defined by the neural network structure, and are constructed using the inner product between gradients of pairs of data points. With the observation of the NTK's stability under sufficient over-parameterization, recent work has shown that (S)GD achieves zero training loss on shallow neural networks for regression task, even if when the data-points are randomly labeled *(Du et al., 2018; Oymak & Soltanolkotabi, 2020; Song & Yang, 2020)*. To study how the labeling of the data affects the convergence, *(Arora et al., 2019)* characterizes the loss update in terms of the NTK-induced inner-product of the label vector, and notices that, when the label vector aligns with the top eigenvectors of the NTK, training achieves a faster convergence rate. *(Su & Yang, 2019)* analyze the convergence of training from a functional approximation perspective, and obtains a meaningful result under infinite sample size limit, where the minimum eigenvalue of the NTK matrix goes to zero.

Later works start to deviate from the NTK-based analysis and aim at reducing the over-parametereization requirement. By using a more refined analysis on the evolution of the Jacobian matrix, *(Oymak & Soltanolkotabi, 2019)* reduce the required hidden-layer width to $n^2$, with $n$ being the sample size. *(Nguyen, 2021)* leverages a property of the gradient that resembles the PL-condition, and provides convergence guarantees for deep neural network under proper initialization. When applied to neural networks with one hidden layer, their over-parameterization requirement also reduces to $n^2$. *(Song et al., 2021)* further study

the PL-condition along the path of the optimization and show that a subquadratic over-parameterization is sufficient to guarantee convergence. *While reducing the over-parameterization is ideal, the focus of our work is to extend the analysis on regular neural network training to a more general training scheme.*

A different line of work explores the structure of the data-distribution in classification tasks, by assuming separability when mapped to the Hilbert space induced by the partial application of the NTK *(Ji & Telgarsky, 2020; Mianjy & Arora, 2020).* Rather than depending on the stability of NTK, the crux of these works relies on the small change in the linearization of the network function. This line of work requires milder overparameterization, and can be easily extended to training stochastic gradient descent without changing the over-parameterization requirement. *The above literature assumes all parameters are updated per iteration.*

**Analysis of Dropout.** There is literature devoted to the analysis of dropout training. For shallow linear neural networks, *(Senen-Cerda & Sanders, 2020a)* give asymptotic convergence rate by carefully characterizing the local minima. For deep neural networks with ReLU activations, *(Senen-Cerda & Sanders, 2020b)* shows that the training dynamics of dropout converge to a unique stationary set of a projected system of differential equations. Under NTK assumptions, *(Mianjy & Arora, 2020)* shows sublinear convergence rate for an online version for dropout in classification tasks. Recently, *(LeJeune et al., 2021)* study the duality of Dropout in a linear regression problem and transform the dropout into a penalty term in the loss. *Our main theorem implies linear convergence rate of the training loss dynamic for the regression task on a shallow neural network with ReLU activations.*

**Federated Learning and Distributed Training.** Traditional analysis on distributed training methods assumes that the objective can be written as a sum or average of a sequence sub-objectives *(Stich, 2018; Li et al., 2019; Haddadpour & Mahdavi, 2019; Khaled et al., 2019a;b).* Each worker/client performs some variant of local (stochastic) gradient descent on a subset of the sub-objectives. This line of work deviates greatly from the scenario we are considering. First, the assumption that the objective can be broken down into linear combination of sub-objectives corresponds to the data-parallel training, in stark contrast to our proposed scheme: the loss computed on the whole network not necessarily equals to the mean of the losses computed on the subnetworks. Second, this line of work assumes that the objective is smooth and usually convex or strongly convex. Moreover, there is recent work on training partially masked neural networks that deviates from the assumption that the objective can be written as a linear combination of sub-objectives *(Mohtashami et al., 2021).* In particular, in this work *(Mohtashami et al., 2021),* the authors consider optimizing a more general class of differentiable objective functions. However, assumptions including Lipschitzness and bounded perturbation made in their work cannot be easily checked for a concrete neural network, especially for the problem of minimizing mean squared error. Lastly, recent advances in the NTK theory also facilitated the theoretical work on Federated Learning (FL) on neural network training. FL-NTK *(Huang et al., 2021)* characterize the asymmetry of the NTK matrix due to the partial data knowledge. For non-i.i.d. data distribution, *(Deng & Mahdavi, 2021)* proves convergence for a shallow neural network by analyzing the semi-Lipschitzness of the hidden layer. *Our work differs since we consider training a partial model with the whole dataset. We consider the more frequently used setting of a one-hidden layer perceptron with non-differentiable activation.*

## 3   Training with Randomly Masked Neurons

We use bold lower-case letters (e.g., $\mathbf{a}$) to denote vectors, bold upper-case letters (e.g., $\mathbf{A}$) to denote matrices, and standard letters (e.g., $a$) for scalars. $\|\mathbf{a}\|_2$ stands for the $\ell_2$ (Euclidean) vector norm, $\|\mathbf{A}\|_2$ stands for the spectral matrix norm, and $\|\mathbf{A}\|_F$ stands for the Frobenius norm. For an integer $a$, we use $[a]$ to denote the enumeration set $\{1, 2, \cdots, a\}$. Unless otherwise stated, $p$ denotes the number of subnetworks, and $l \in [p]$ its index; $K$ denotes the number of global iterations and $k \in [K]$ its index; $\tau$ is used for the number of local iterations and $t \in [\tau]$ its index. We use $\mathbf{M}_k$ to denote the mask at global iteration $k$, and $\mathbb{E}_{[\mathbf{M}_k]}[\cdot] = \mathbb{E}_{\mathbf{M}_0, \ldots, \mathbf{M}_k}[\cdot]$ to denote the total expectation over masks $\mathbf{M}_0, \ldots, \mathbf{M}_k$. We use $\mathbb{P}(\cdot)$ to denote the probability of an event, and $\mathbb{I}\{\cdot\}$ to denote the indicator function of an event. For distributions, we use $\mathcal{N}(\boldsymbol{\mu}, \boldsymbol{\Sigma})$ to denote the Gaussian distribution with mean $\boldsymbol{\mu}$ and variance $\boldsymbol{\Sigma}$. We use $\texttt{Bern}(\xi)$ to denote the Bernoulli distribution with mean $\xi$, and we use $\texttt{Unif}(S)$ to denote the uniform distribution over the set $S$. For a complete list of notation, see Table 1 in the Appendix.

### 3.1 Single Hidden-Layer Neural Network with ReLU activations

We consider the single hidden-layer neural network with ReLU activations, as in:

$$f(\mathbf{W}, \mathbf{a}, \mathbf{x}) = \frac{1}{\sqrt{m}} \sum_{r=1}^{m} a_r \sigma(\langle \mathbf{w}_r, \mathbf{x} \rangle) := f(\mathbf{W}, \mathbf{x}).$$

Here, $\mathbf{W} = \begin{bmatrix} \mathbf{w}_1, \ldots, \mathbf{w}_m \end{bmatrix}^\top \in \mathbb{R}^{m \times d}$ is the weight matrix of the first layer, and $\mathbf{a} = \begin{bmatrix} a_1, \ldots, a_m \end{bmatrix}^\top \in \mathbb{R}^m$ is the weight vector of the second layer. We assume that each $\mathbf{w}_r$ is initialized based on $\mathcal{N}(0, \kappa^2 \mathbf{I})$. Each weight entry $a_r$ in the second layer is initialized uniformly at random from $\{-1, 1\}$. As in *(Du et al., 2018; Zou et al., 2020; Soltanolkotabi et al., 2018; Oymak & Soltanolkotabi, 2019; Li et al., 2020; Oymak & Soltanolkotabi, 2020)*, $\mathbf{a}$ is fixed.

Consider a subnetwork computing scheme with $p$ workers. In the $k$-th global iteration, we consider each binary mask $\mathbf{M}_k \in \{0, 1\}^{m \times p}$ to be composed of subnetwork masks $\mathbf{m}_k^l \in \{0, 1\}^m$ for $l \in [p]$. The $r$-th entry of $\mathbf{m}_k^l$ is denoted

---

**Algorithm 1** Randomly Masked Training

**Input:** Mask Distribution $\mathcal{D}$, local step-size $\eta$, global aggregation weight $\eta_{k,r}$

1: Initialize $\mathbf{W}_0, \mathbf{a}$
2: **for** $k = 0, \ldots, K - 1$ **do**
3:      Sample mask $\mathbf{M}_k \sim \mathcal{D}$
4:      **for** $l = 1, \ldots, p$ **do**
5:          $\mathbf{W}_{k,0}^l \leftarrow \mathbf{W}_k$
6:          **for** $t = 0, \ldots, \tau - 1$ **do**
7:              $\mathbf{W}_{k,t+1}^l \leftarrow \mathbf{W}_{k,t}^l - \eta \frac{\partial L_{\mathbf{m}_k^l}(\mathbf{W}_{k,t}^l)}{\partial \mathbf{W}}$
8:          **end for**
9:          $\Delta \mathbf{W}_k^l \leftarrow \mathbf{W}_{k,\tau}^l - \mathbf{W}_k$
10:      **end for**
11:      **for** $r = 1, \ldots, m$ **do**
12:          $\mathbf{w}_{k+1,r} \leftarrow \mathbf{w}_{k,r} + \eta_{k,r} \sum_{l=1}^{p} \Delta \mathbf{w}_{k,r}^l$
13:      **end for**
14: **end for**

---

as $m_{k,r}^l$, with $m_{k,r}^l = 1$ indicating that neuron $r$ is active in subnetwork $l$ in the $k$th global iteration, and $m_{k,r}^l = 0$ otherwise. We assume that the sampling of the masks for each neuron is independent of other neurons, and further impose the condition that the event $m_{k,r}^l = 1$ happens with a fixed probability for all $k, r$ and $l$. We denote this probability with $\xi = \mathbb{P}\left(m_{k,r}^l = 1\right)$. The surrogate function defined by a subnetwork mask $\mathbf{m}_k^l$ is given by:

$$f_{\mathbf{m}_k^l}(\mathbf{W}, \mathbf{x}) = \frac{1}{\sqrt{m}} \sum_{r=1}^{m} a_r m_{k,r}^l \sigma(\langle \mathbf{w}_r, \mathbf{x} \rangle).$$

With colored text, we highlight the differences between the full model and the surrogate functions. Consider the dataset given by $(\mathbf{X}, \mathbf{y}) = \{(\mathbf{x}_i, y_i)\}_{i=1}^n$. We make the following assumption on the dataset:

**Assumption 1.** *For any $i \in [n]$, it holds that $\|\mathbf{x}_i\|_2 = 1$ and $|y_i| \leq C - 1$ for some constant $C \geq 1$. Moreover, for any $j \neq i$ it holds that the points $\mathbf{x}_i, \mathbf{x}_j$ are not co-aligned, i.e., $\mathbf{x}_i \neq \zeta \mathbf{x}_j$ for any $\zeta \in \mathbb{R}$.*

This assumption is quite standard as in previous literature *(Du et al., 2018; Arora et al., 2019; Song & Yang, 2020)*. We consider training the neural network using the regression loss. Given a dataset $(\mathbf{X}, \mathbf{y})$, the function output on the whole dataset is denoted as $f(\mathbf{W}, \mathbf{X}) = \begin{bmatrix} f(\mathbf{W}, \mathbf{x}_1), \ldots, f(\mathbf{W}, \mathbf{x}_n) \end{bmatrix}$. Then, the (scaled) mean squared error (MSE) of a surrogate model is given by:

$$L_{\mathbf{m}_k^l}(\mathbf{W}) = \left\| \mathbf{y} - f_{\mathbf{m}_k^l}(\mathbf{W}, \mathbf{X}) \right\|_2^2.$$

The surrogate gradient is computed as:

$$\frac{\partial L_{\mathbf{m}_k^l}(\mathbf{W})}{\partial \mathbf{w}_r} = \frac{1}{\sqrt{m}} \sum_{i=1}^{n} a_r m_{k,r}^l \left( f_{\mathbf{m}_k^l}(\mathbf{W}, \mathbf{x}_i) - y_i \right) \mathbf{x}_i \mathbb{I}\{\langle \mathbf{w}_r, \mathbf{x}_i \rangle \geq 0\}.$$

Let $\eta$ be a constant subnetwork training learning rate, and let the aggregation weight $\eta_{k,r}$ be zero if neuron $r$ is active in no subnetwork in the $k$th iteration; otherwise $\eta_{k,r}$ is set to the inverse of the number of subnets in which it is active. Within this setting, the general training algorithm is given by Algorithm 1.

## 4 Convergence on Two-Layer ReLU Neural Network

We assume that $m_{k,r}^l = 1$ happens with a fixed probability for all $k, r$ and $l$, and such probability is denoted by $\xi$. Consequently, the forward pass of the surrogate function is a linear combination of $\xi$-proportion of

the neurons' output. To keep the pre-activation of the hidden layer at the same scale for both the whole network and the subnetwork, we multiply the weight of the whole network with a factor of $\xi$. Due to the homogeneity of the ReLU activation, this is equivalent to scaling the output of each neuron, as in *(Mianjy & Arora, 2020)*. For notation clarity, we define:

$$u_k^{(i)} = \frac{1}{\sqrt{m}} \sum_{r=1}^{m} a_r \xi \sigma(\langle \mathbf{w}_{k,r}, \mathbf{x}_i \rangle) = \frac{\xi}{\sqrt{m}} \sum_{r=1}^{m} a_r \sigma(\langle \mathbf{w}_{k,r}, \mathbf{x}_i \rangle).$$

Adding this scaling factor gives the property that $\mathbb{E}_{\mathbf{M}_k} \left[ f_{\mathbf{m}_k^l}(\mathbf{W}_k, \mathbf{x}_i) \right] = u_k^{(i)}$, meaning that the sampled subnetworks in the global iteration $k$ are unbiased estimators of the aggregated network in the global iteration $k-1$. Here $u_k^{(i)}$ is both the initial whole network output in global iteration $k$ and the aggregated network output in global iteration $k-1$. We focus on the behavior of the following loss, computed on the scaled whole network over iterations $k$:

$$L_k = \|\mathbf{y} - \mathbf{u}_k\|_2^2, \quad \text{where} \quad \mathbf{u}_k = \left[ u_k^{(1)}, \ldots, u_k^{(n)} \right].$$

This is the regression loss over iterations $k$ between observations $\mathbf{y}$ and the learned model $\mathbf{u}_k$.

**Properties of subnetwork NTK.** Recent works on analyzing the convergence of gradient descent for neural networks consider approximating the function output $\mathbf{u}_k$ with the first order Taylor expansion *(Du et al., 2018; Arora et al., 2019; Song & Yang, 2020)*. For constant step size $\eta$, taking the gradient descent's (i.e., $\mathbf{W}_{k+1} = \mathbf{W}_k - \eta \nabla_{\mathbf{W}} L(\mathbf{W}_k)$) first-order Taylor expansion, we get:

$$u_{k+1}^{(i)} \approx u_k^{(i)} + \left\langle \nabla_{\mathbf{W}} u_k^{(i)}, \mathbf{W}_{k+1} - \mathbf{W}_k \right\rangle \approx u_k^{(i)} - \xi \eta \sum_{j=1}^{n} \mathbf{H}(k)_{ij} (u_k^{(j)} - y_j), \tag{1}$$

where $\mathbf{H}(k) \in \mathbb{R}^{n \times n}$ is the finite-width NTK matrix of iteration $k$, given by

$$\mathbf{H}(k)_{ij} = \frac{\xi}{m} \langle \mathbf{x}_i, \mathbf{x}_j \rangle \sum_{r=1}^{m} \mathbb{I}\{\langle \mathbf{w}_{k,r}, \mathbf{x}_i \rangle \geq 0, \langle \mathbf{w}_{k,r}, \mathbf{x}_j \rangle \geq 0\}. \tag{2}$$

Compared with the previous definition of finite-width NTK, we have an additional scaling factor $\xi$. This is because, based on our later definition of masked-NTK, we would like the masked-NTK to be an unbiased estimator of the finite-width NTK. In the overparameterized regime, the change of the network's weights is controlled in a small region around initialization. Therefore, the change of $\mathbf{H}(k)$ is small, staying close to the NTK at initialization. Moreover, the latter can be well approximated by the infinite-width NTK:

$$\mathbf{H}_{ij}^{\infty} = \xi \cdot \mathbb{E}_{\mathbf{w} \sim \mathcal{N}(0, \mathbf{I})} \left[ \langle \mathbf{x}_i, \mathbf{x}_j \rangle \mathbb{I}\{\langle \mathbf{w}, \mathbf{x}_i \rangle \geq 0, \langle \mathbf{w}, \mathbf{x}_j \rangle \geq 0\} \right].$$

*(Du et al., 2018)* shows that $\mathbf{H}^{\infty}$ is positive definite.

**Theorem 1.** *(Du et al., 2018) Denote $\lambda_0 := \lambda_{\min}(\mathbf{H}^{\infty})$, the minimum eigenvalue of $\mathbf{H}^{\infty}$. Then we have $\lambda_0 > 0$ as long as assumption (1) holds.*

With $\mathbf{H}(k)$ staying sufficiently close to $\mathbf{H}^{\infty}$, *(Du et al., 2018; Arora et al., 2019; Song & Yang, 2020)* show that $\lambda_{\min}(\mathbf{H}(k)) \geq \frac{\lambda_0}{2} > 0$. Moreover, Equation 1 implies that

$$\mathbf{u}_{k+1} - \mathbf{u}_k \approx -\xi \eta \mathbf{H}(k)(\mathbf{u}_k - \mathbf{y}),$$

that further leads to linear convergence rate:

$$L_{k+1} \approx L_k + \langle \nabla_{\mathbf{u}_k} L_k, \mathbf{u}_{k+1} - \mathbf{u}_k \rangle \approx L_k - \xi \eta \langle \mathbf{u}_k - \mathbf{y}, \mathbf{H}(k)(\mathbf{u}_k - \mathbf{y}) \rangle \approx (1 - \xi \eta \lambda_0) L_k.$$

In NTK analysis, the Taylor expansion for both $\mathbf{u}_k$ and $L_k$ produces an error term that improves the convergence rate from $\eta \lambda_0$ to $\gamma \eta \lambda_0$ with $\gamma \in (0, 1)$ being a constant.

For our scenario, the randomly sampled subnetworks bring a trickier situation onto the table: in each iteration, due to the different masks, the NTK changes even when the weights stay the same. To tackle this difficulty, we provide a generalization of the definition of the finite-width NTK that takes both the mask and the weight into consideration:

**Definition 1.** *Let $\mathbf{m}_{k'}^l$ be the mask of subnetwork $l$ in iteration $k'$. We define the masked-NTK in global iteration $k$ and local iteration $t$ induced by $\mathbf{m}_{k'}^l$ as:*

$$\left(\mathbf{m}_{k'}^l \circ \mathbf{H}(k,t)\right)_{ij} = \tfrac{1}{m}\langle \mathbf{x}_i, \mathbf{x}_j\rangle \sum_{r=1}^{m} m_{k',r}^l \mathbb{I}\{\langle \mathbf{w}_{k,t,r}, \mathbf{x}_i\rangle \geq 0, \langle \mathbf{w}_{k,t,r}, \mathbf{x}_i\rangle \geq 0\}.$$

Here, with colored text we highlight the main differences to the common NTK definition. Although we are only interested in the masked-NTK with $k = k'$, to facilitate our analysis on the minimum eigenvalue of masked-NTK, we also allow $k \neq k'$. We point out two connections between our masked-NTK and the vanilla NTK: *i)* the masked-NTK is an unbiased estimator of the whole network's NTK; *ii)* when $\xi = 1$, the masked-NTK reduce to the vanilla NTK as in equation (2). Throughout iterations of the algorithm, the following theorem shows that all masked-NTKs stay sufficiently close to the infinite-width NTK.

**Theorem 2.** *Suppose the number of hidden nodes satisfies $m = \Omega\left(n^2 \log(Kpn/\delta)/\xi\lambda_0^2\right)$. If for all $k,t$ it holds that $\|\mathbf{w}_{k,t,r} - \mathbf{w}_{0,r}\|_2 \leq R := \frac{\kappa\lambda_0}{8n}$, then with probability at least $1 - \delta$, for all $k, k' \in [K]$ we have:*

$$\lambda_{\min}(\mathbf{m}_{k'}^l \circ \mathbf{H}(k,t)) \geq \tfrac{\lambda_0}{2}.$$

The above theorem relies on the small weight change in iteration $(k,t)$. Such assumption is also made in previous work *(Du et al., 2018; Song & Yang, 2020)* to show the positive definiteness of the NTK matrix. In order to guarantee each subnetwork's loss decrease, we need to ensure that the *i)* the weight change is bounded up to global iteration $k$ (this implies that when a subnetwork is sampled from the whole network, its weights do not deviate much from the initialization); and, *ii)* the weight change during the local training of the subnetwork is also bounded. The following hypothesis establishes these two conditions, and sets up the "skeleton" to construct different theorems, based on problems considered. *The aim of this work is to prove this hypothesis for several cases.*

**Hypothesis 1.** *Fix the number of global iterations $K$. Suppose the number of hidden nodes satisfies $m = \Omega\left(n^2 \log(Kpn/\delta)/\xi\lambda_0^2\right)$, and suppose we use a constant step size $\eta = O\left(\lambda_0/n^2\right)$. If the weight perturbation before iteration $k$ is bounded by*

$$\|\mathbf{w}_{k,r} - \mathbf{w}_{0,r}\|_2 + 2\eta\tau\sqrt{\tfrac{nK}{m\delta}}\mathbb{E}_{[\mathbf{M}_{k-1}],\mathbf{W}_0,\mathbf{a}}\left[\|\mathbf{y} - \mathbf{u}_k\|_2\right] + (K-k)\kappa\sqrt{\xi(1-\xi)pn} \leq R, \tag{3}$$

*then, for all $t \in [\tau]$, with probability at least $1 - 4\delta$, we have:*

$$\left\|\mathbf{y} - f_{\mathbf{m}_k^l}\left(\mathbf{W}_{k,t+1}^l, \mathbf{X}\right)\right\|_2^2 \leq \left(1 - \tfrac{\eta\lambda_0}{2}\right)\left\|\mathbf{y} - f_{\mathbf{m}_k^l}\left(\mathbf{W}_{k,t}^l, \mathbf{X}\right)\right\|_2^2, \tag{4}$$

*and the local weight perturbation satisfies:*

$$\|\mathbf{w}_{k,t,r} - \mathbf{w}_{k,r}\| \leq \tfrac{\eta\tau\sqrt{2nK}}{\sqrt{m\delta}}\mathbb{E}_{[\mathbf{M}_{k-1}],\mathbf{W}_0,\mathbf{a}}\left[\|\mathbf{y} - \mathbf{u}_k\|_2\right] + 2\eta\kappa n\sqrt{\tfrac{2\xi(1-\xi)pK}{m\delta}}. \tag{5}$$

The hypothesis above states that, in a given global step $k$, given a small weight perturbation guarantee (Equation 3) up to the current global iterations, each subnetwork's local loss also decreases linearly (Equation 4), as well as the weight perturbation remains bounded (Equation 5). *Yet, the above hypothesis does not connect the subnetwork's loss with the whole network's loss through the sampling and aggregation process.* Our aim is to turn Hypothesis 1 into a series of specific theorems that cover different cases. In particular, we prove using induction the condition for which Hypothesis (1) holds under: *i)* masks with i.i.d. Bernoulli; and *ii)* masks with i.i.d categorical rows. Utilizing these results, we provide convergence results for the two scenarios. This is the goal in the following section.

## 4.1 Generic Convergence Result under Bernoulli Mask

While the local gradient descent for each subnetwork is guaranteed to make progress with high probability, when a large network is split into small subnetworks, the expected loss on the dataset increases. Since $\mathbb{E}_{\mathbf{M}_k}\left[f_{\mathbf{m}_k^l}\left(\mathbf{W}_k, \mathbf{x}_i\right)\right] = u_k^{(i)}$, simply expanding the MSE reveals that:

$$\mathbb{E}_{\mathbf{M}_k}\left[\left\|\mathbf{y} - f_{\mathbf{m}_k^l}\left(\mathbf{W}_k, \mathbf{x}_i\right)\right\|_2^2\right] = \|\mathbf{y} - \mathbf{u}_k\|_2^2 + \mathbb{E}_{\mathbf{M}_k}\left[\left\|f_{\mathbf{m}_k^l}\left(\mathbf{W}_k, \mathbf{x}_i\right) - \mathbf{u}_k\right\|_2^2\right].$$

When analyzing the convergence, the second term on the right-hand side needs to be carefully dealt with. It is non-trivial to show that, when combining the updated network of the local steps, the loss computed on the whole network is smaller than or equal to the error of each sub-network. We will solve these technical difficulties for the training procedure with subnetworks created using masks sampled from two types of distribution. In this section, we focus on masks satisfies the following Bernoulli assumption:

**Assumption 2.** *(Bernoulli Mask) Each mask entry $m_{k,r}^l$ is independently from a Bernoulli distribution with mean $\xi$, i.e., $m_{k,r}^l \sim \text{Bern}(\xi)$.*

Masks sampled in this fashion allow a neuron to be active in more than one subnetworks, or none of the subnetworks. For convenience, we denote the probability that a neuron is active in at least one subnetwork with $\theta = 1 - (1-\xi)^p$. In the meantime, subnetworks created using Bernoulli masks enjoy full independence, and thus have nice concentration properties. By carefully analyzing the aggregated gradient of each local step, we arrive at the following generic convergence theorem, under the Bernoulli mask assumption.

**Theorem 3.** *Let assumptions (1) and (2) hold. Then $\lambda_0 > 0$. Fix the number of global iterations to $K$ and the number of local iterations to $\tau$. Let the number of hidden neurons satisfy:*

$$m = \Omega \left( \frac{K}{\delta} \max \left\{ \frac{n^4}{\kappa^2 \xi \theta \lambda_0^4}, \frac{nK^2 B_1}{\kappa^2 \theta \lambda_0^2}, K^2 p \right\} \right). \tag{6}$$

*Then Algorithm (1) with a constant step-size $\eta = O\left(\frac{\lambda_0}{\max\{n,p\}n\tau}\right)$ converges with probability at least $1 - \delta$, according to:*

$$\mathbb{E}_{[\mathbf{M}_{k-1}]} \left[ \|\mathbf{y} - \mathbf{u}_k\|_2^2 \right] \le \left(1 - \tfrac{1}{4}\eta\theta\tau\lambda_0\right)^k \|\mathbf{y} - \mathbf{u}_0\|_2^2 + B_1, \tag{7}$$

*for some error region level $B_1 > 0$, defined as:*

$$B_1 = O\left( \frac{(1-\xi)^2 n^3 d}{m\lambda_0^2} + \frac{(\theta - \xi^2)n\kappa^2}{p} + \left(1 - \frac{1}{\tau}\right)^2 \theta^2 (1-\xi)n\kappa^2 \right). \tag{8}$$

Overall, given the overparameterization requirement in Equation 6, the neural network training error, as expressed in Equation 7, drops linearly up to a neighborhood around the optimal point, defined by $B_1$ in Equation 8. We notice that $B_1$ has three terms that all reduce to zero when $\xi = 1$. In the first term of $B_1$, $m$ appears in the denominator, implying that this term can be arbitrarily decreased as the cost of increasing the number of hidden neurons. The second term is kept at a constant scale, as long as the initialization scale $\kappa$ is small enough. The third term disappears when the number of local steps is one. However, the loss decreases more in each global iteration, when $\tau$ is larger, since the convergence rate is $1 - O(\eta\theta\tau\lambda_0)$. In the case of $\xi = 1, p = 1$ and $\tau = 1$, the proposed framework reduces to the whole network training. Choosing $\kappa = 1$, Theorem 3 reduces to a form similar to *(Song & Yang, 2020)*, with the same convergence rate and over-parameterization requirement.

***Remark.*** Compared to *(Du et al., 2018; Song & Yang, 2020)*, the scenario considered in our work involves an additional randomness introduced by the mask. Thus, our convergence result is based on the expectation of the loss: we derive the bound of the loss from the bound of its expectation, using concentration inequalities, and apply a union bound over all iterations $k \in [K]$. Therefore, the required over-parameterization on $m$ grows as we increase the number of global iterations, meaning that, under a fixed $m$, the convergence is only guaranteed for a bounded number of iterations. This is not a concern in general since to guarantee $\epsilon$ small training error we only need $K$ to be $\frac{\log \epsilon^{-1} + \log n}{\log(1 - O(\eta\theta\tau\lambda_0)^{-1})}$. This is termed as early-stopping, and is used in previous literature *(Su & Yang, 2019; Allen-Zhu et al., 2018)*.

The complete proof of this theorem is defered to Appendix E, and we sketch the proof below:

1. Let $X_{k,r} = \sum_{l=1}^p m_{k,r}^l$ denote the number of subnetworks that update neuron $r$ in global iteration $k$. Let $N_{k,r} = \max\{X_{k,r}, 1\}$ to be the normalizer of the aggregated gradient, $N_{k,r}^\perp = \min\{X_{k,r}, 1\}$

to be the indicator of whether a neuron is selected by at least one subnetwork. Then, the update of each weight vector can be written as:

$$\mathbf{w}_{k+1,r} = \mathbf{w}_{k,r} - \eta \cdot \frac{N_{k,r}^{\perp}}{N_{k,r}} \sum_{t=0}^{\tau-1} \sum_{l=1}^{p} \frac{\partial L\left(\mathbf{W}_{k,t,r}^{l}\right)}{\partial \mathbf{w}_r}. \tag{9}$$

2. We first focus on the aggregated gradient of the first local step $\frac{N_{k,r}^{\perp}}{N_{k,r}} \sum_{l=1}^{p} \frac{\partial L_{\mathbf{m}_k^l}\left(\mathbf{W}_{k,t,r}^{l}\right)}{\partial \mathbf{w}_r}$, and show that this aggregated gradient satisfies a concentration property around a point near the ideal gradient $\frac{\partial L(\mathbf{W}_k)}{\partial \mathbf{w}_r}$, and such concentration is closer if $p$ is larger.

3. We notice that the difference between the aggregated gradient in the later local steps and the aggregated gradient in the first local step depends on how much the local weight of each subnetwork in the later local step deviates from the weight of the first local step. We then show that the local weight change is bounded, implying that the aggregated gradient in all local steps lie near to the aggregated gradient in the first local step.

4. Lastly, we use the standard NTK technique to show that the aggregated gradient update in equation (9) leads to linear convergence per each global step, with an additional error term.

To interpret the theorem, we choose $\kappa = n^{-\frac{1}{2}}$, make mild assumptions and simplify the key messages of the form in Theorem (3). Note that this choice of $\kappa$ is the same as in (Arora et al., 2019).

**Assumption 3.** *For the simplicity of our theorem, we assume that* $\max\{K, d, p\} \leq n$ *and* $\lambda_0 \leq 1$.

Notice that for all $m$ that satisfies equation (6), the first term in $B_1$ is upper bounded by $O(1)$. Moreover, since $p \geq 1$ and $\tau \geq 1$, by choosing $\kappa = n^{-\frac{1}{2}}$, the second and third term are also upper bounded by $O(1)$. Therefore, $B_1$ is upper bounded by $O(1)$. Moreover, since $\lambda_0 \geq 1$ and $\max\{K, p\} \leq n$, we have that both $\frac{nK^2 B_1}{\kappa^2 \theta \lambda_0^2}$ and $K^2 p$ are smaller than $\frac{n^4}{\kappa^2 \xi \theta \lambda_0^4}$, so the over-parameterization requirement in equation (6) reduces to $m = \frac{n^5 K}{\delta \xi \theta \lambda_0^4}$. For different choice of $\tau$ and $p$, our considered scenario reduces to different existing algorithms. In the following, we provide convergence results of these algorithms, as corollaries of Theorem (3), by considering different $\tau$ and $p$ values.

**Dropout.** The dropout algorithm *(Srivastava et al., 2014)* corresponds to the case $\tau = 1, p = 1$. For this assignment, we arrive at the following corollary.

**Corollary 1.** *Let assumptions (1), (2), and (3) holds. Fix the number of dropout iterations to $K$, the step size to $\eta = O\left(\lambda_0/n^2\right)$, and let the number of hidden neurons satisfies $m = \Theta\left(n^5 K/\xi^2 \lambda_0^4 \delta\right)$. Then, the dropout algorithm on a two-layer ReLU neural network converges with probability at least $1 - \delta$, according to:*

$$\mathbb{E}_{[\mathbf{M}_{k-1}]}\left[\|\mathbf{y} - \mathbf{u}_k\|_2^2\right] \leq \left(1 - \frac{1}{4}\eta\xi\lambda_0\right)^k \|\mathbf{y} - \mathbf{u}_0\|_2^2 + O\left(1 - \xi\right).$$

Typically, $1 - \xi$ is usually referred to as the "dropout rate". In our result, as $\xi$ approaches 0, which corresponds to the scenario that no neurons are selected, the convergence rate approaches 1, meaning that the loss hardly decreases. In the mean time, the error term remains constant. On the contrary, as $\xi$ approaches 1, which corresponds to the scenario that all neurons are selected, we get the same convergence rate of $1 - O\left(\eta\lambda_0\right)$ as in previous literature *(Du et al., 2018; Song & Yang, 2020)*, and the error term decreases to 0. Moreover, we should note that the over-parameterization requirement also depends on $\xi$. In particular, as $\xi$ becomes smaller, we need a larger number of hidden neurons to guarantee convergence.

**Multi-Sample Dropout.** The multi-sample dropout *(Inoue, 2019)* corresponds to the scenario where $\tau = 1, p \geq 1$. Our corollary below indicates how increasing $p$ helps the convergence.

**Corollary 2.** *Let assumptions (1), (2), and (3) hold. Fix the number of dropout iterations to $K$, the step size to $\eta = O\left(\lambda_0/n^2\right)$, and let the number of hidden neurons satisfy $m = \Theta\left(n^5 K/\xi \theta \lambda_0^4 \delta\right)$. Then the p-sample*

*dropout algorithm on a two-layer ReLU neural network converges with probability at least $1 - \delta$, according to:*

$$\mathbb{E}_{[\mathbf{M}_{k'-1}]} \left[ \|\mathbf{y} - \mathbf{u}_{k'}\|_2^2 \right] \leq \left( 1 - \frac{1}{4}\eta\theta\lambda_0 \right)^{k'} \|\mathbf{y} - \mathbf{u}_0\|_2^2 + O\left( \frac{(1-\xi)^2}{nK} + \frac{\theta - \xi^2}{p} \right).$$

Recall that $\theta = 1 - (1 - \xi)^p$ denotes the probability that a neuron is selected by at least one subnetwork. Based on this corollary, increasing the number of subnetworks $p$ improve the convergence rate and the over-parameterization requirement since $\theta$ increases as $p$ increases. Moreover, increasing the number of subnetworks help decreasing the error term even when the dropout rate $\xi$ is fixed. After $p$ is as large as $nK$, the error term stops decreasing, dominated by the term $O\left(\frac{(1-\xi)^2}{nK}\right)$. Lastly, compared with the result of dropout, the over-parameterization depends not only on $\xi$, but also on $\theta$.

**Multi-Worker IST.** The multi-worker IST algorithm *(Yuan et al., 2020)* is very similar to the general scheme with $p \geq 1$ and $\tau \geq 1$, but with the additional assumption that $\max\{K, d, p\} \leq n$, and a special choice of initialization $\kappa = n^{-\frac{1}{2}}$.

**Corollary 3.** *Let assumptions (1), (2), and (3) hold. Fix the number of dropout iterations to $K$, the step size to $\eta = O\left(\lambda_0/n\tau \max\{n, p\}\right)$, and let the number of hidden neurons satisfy $m = \Theta\left(n^5 K/\xi\theta\lambda_0^4\delta\right)$. Then the IST algorithm on a two-layer ReLU neural network converges with probability at least $1 - \delta$, according to:*

$$\mathbb{E}_{[\mathbf{M}_{k-1}]} \left[ \|\mathbf{y} - \mathbf{u}_k\|_2^2 \right] \leq \left( 1 - \frac{1}{4}\eta\theta\tau\lambda_0 \right)^{k} \|\mathbf{y} - \mathbf{u}_0\|_2^2 + O\left( \frac{(1-\xi)^2}{nK} + \frac{\theta - \xi^2}{p} + \left(1 - \frac{1}{\tau}\right)\theta^2(1-\xi) \right)$$

While IST with subnetworks constructed using Bernoulli masks , it allows a neuron to be active in more than one or none of the subnetworks. In the next section, we consider another mask sampling approach that fits better into the scenario of the original IST, where each hidden neuron is distributed to one and only one subnetwork with uniform probability.

### 4.2 Multi-Subnetwork Convergence Result for Categorical Mask

We consider masks sampled from categorical distribution, as explained by the assumption below:

**Assumption 4.** *We assume that $\mathbf{M}_k \sim \texttt{Categorical}(p)$. To be specific, for each $r \in [m]$, let $l'_r \sim \texttt{Unif}([p])$, and we define $m_{k,r}^l = 1$ if $l = l'_r$ and $m_{k,r}^l = 0$ otherwise.*

In this way, the masks endorsed by each worker are non-overlapping (as stated in *(Yuan et al., 2020)*), and the union of the masks covers the whole set of hidden neurons. However, we note that the subnetworks created by the masks sampled according to this fashion are no longer independent. The following theorem presents the convergence result under this setting.

**Theorem 4.** *Let assumptions (1) and (4) hold. Then $\lambda_0 > 0$. Moreover, let $\lambda_{\max}$ denote the maximum eigenvalue of $\mathbf{H}^\infty$. Fix the number of global iterations to $K$ and the number of local iterations to $\tau$. Let the number of hidden neurons be $m = \Omega\left(\frac{n^5\tau^2 K\lambda_{\max}}{\lambda_0^6\delta}\right)$, and choose the initialization scale $\kappa = \sqrt{n\lambda_{\max}}\lambda_0^{-1}$. Let $\gamma = \left(1 - p^{-1}\right)^{\frac{1}{3}}$. Then, Algorithm (1) with a constant step-size $\eta = O\left(\frac{\lambda_0}{n^2}\min\left\{\frac{p}{\gamma^2\tau}, 1\right\}\right)$ converges with probability at least $1 - \delta$, according to:*

$$\mathbb{E}_{[\mathbf{M}_{k-1}]} \left[ \|\mathbf{y} - \mathbf{u}_k\|_2^2 \right] \leq \left( \gamma + (1-\gamma)\left(1 - \frac{\eta\lambda_0}{2}\right)^\tau \right)^{k} \|\mathbf{y} - \mathbf{u}_0\|_2^2 + O\left( \frac{\gamma\tau n\kappa^2\lambda_{\max}}{\lambda_0^2} \right).$$

We defer the proof of this theorem to Appendix F. This theorem has a couple noticeable properties. First, when the number of workers $p = 1$, i.e., the scenario of multi-worker IST reduces to the full-network training, we achieve $\gamma = 0$, which implies that the error term disappears, driving further connections between regular and IST training. Second, when the number of subnetworks $p$ increases, $\gamma$ also increases, leading to a slower decreasing of the training MSE and convergence to a bigger error neighborhood. In particular, as we increase

the number of subnetworks, the number of active neurons in each subnetwork becomes smaller, which makes the subnetworks both harder to train and harder to synchronize. We defer the complete proof of this theorem to Appendix F, and sketch the proof below:

1. Let $\hat{\mathbf{u}}_{k,t}^l = f_{\mathbf{m}_k^l}\left(\mathbf{W}_{k,\tau}^l, \mathbf{X}\right)$. We notice that $f = \frac{1}{p}\sum_{l=1}^p f_{\mathbf{m}_k^l}$. Using this property, we show that $L_{k+1} = \frac{1}{p}\sum_{l=1}^p \|\mathbf{y} - \hat{\mathbf{u}}_{k,\tau}^l\|_2^2 - \frac{1}{p}\sum_{l=1}^p \|\mathbf{u}_{k+1} - \hat{\mathbf{u}}_{k,\tau}^l\|_2^2$. The first term here enjoys linear convergence starting from an initial value of $\|\mathbf{y} - \hat{\mathbf{u}}_{k,0}^l\|_2^2$.

2. It then follows that $\mathbb{E}_{\mathbf{M}_k}[\frac{1}{p}\sum_{l=1}^p \|\mathbf{y}-\hat{\mathbf{u}}_k^l\|_2^2] = L_k + \mathbb{E}_{\mathbf{M}_k}[\frac{1}{p}\sum_{l=1}^p \|\mathbf{u}_k - \hat{\mathbf{u}}_k^l\|_2^2]$. Putting things together, we have that $\mathbb{E}_{\mathbf{M}_k}[L_{k+1}] \le (1-\alpha)^\tau L_k + \iota_k$, where $\iota_k = \mathbb{E}_{\mathbf{M}_k}[\frac{1}{p}\sum_{l=1}^p \|\mathbf{u}_k - \hat{\mathbf{u}}_k^l\|_2^2 - \|\mathbf{u}_{k+1} - \hat{\mathbf{u}}_{k,\tau}^l\|_2^2]$, where $\alpha \in (0,1)$ is some convergence rate achieved by invoking Hypothesis (1).

3. We then use the small weight perturbation induced by over-parameterization, which means that $\|\hat{\mathbf{u}}_k^{l'} - \hat{\mathbf{u}}_{k,\tau}^{l'}\|_2$ is small. This allows us to bound the term $\iota_k$, and arrive at the final convergence.

## 5 Experiments

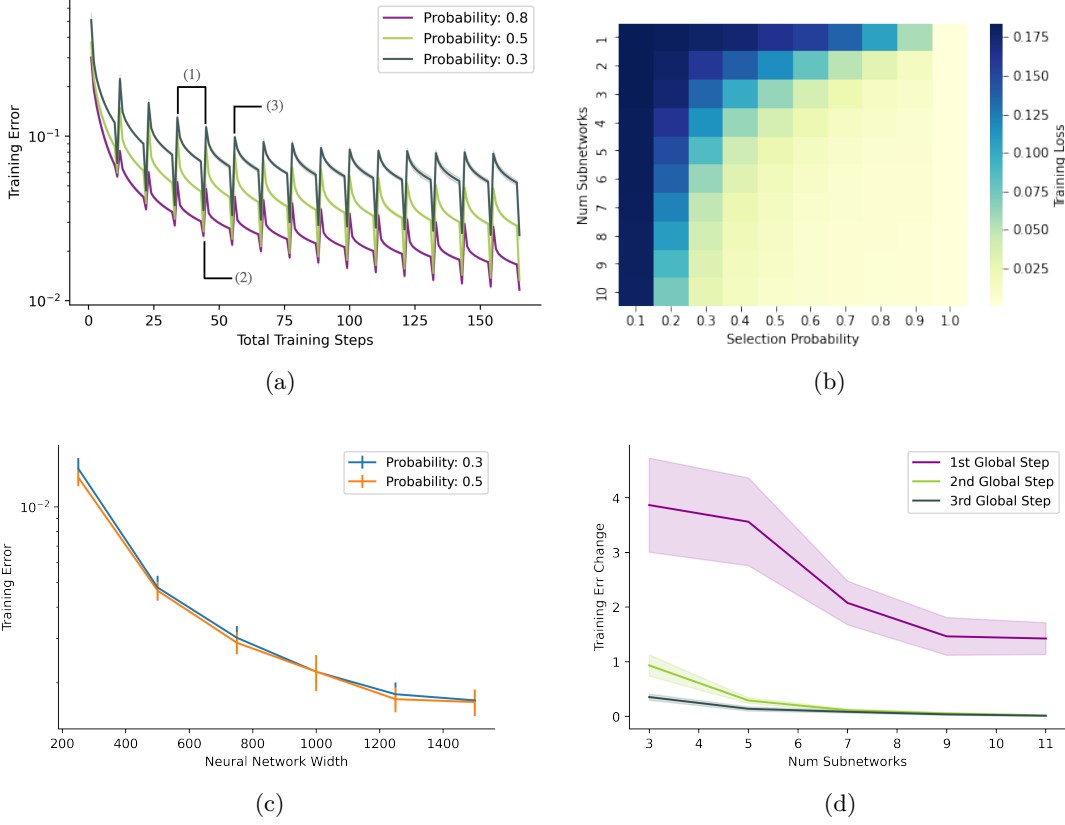

Figure 2: Validation experiments on a single hidden layer perceptron.

We empirically validate our main theorems on a one-hidden-layer, ReLU- activated neural network. With the purpose of performing the experiments on a task that is both widely used and representative, and possess some simplicity to be solved by the one-hidden-layer MLP, we use the features/embeddings extracted from a convolutional-based neural network *(Krizhevsky, 2009)*. This is a common practice: E.g., the recent work in Chowdhury et al. (2021) proposes the use of a library of pre-trained networks to extract useful features, which later on are processed by added topmost layers used for classification. This results in a procedure that takes an image, creates an embedding, and then uses that embedding to build a classifier, by feeding the embedding into a multi-layer perceptron with a single/multiple hidden layers. In this work, we take a

ResNet-50 model *(He et al., 2015)* pretrained on ImageNet as our feature extractor and concatenate it with two fully-connected layers. We then train this combined models on the CIFAR-10 dataset, and take the outputs of the re-trained ResNet-50 model as the input features, and use the logits output of the combined model as the labels. The obtained input feature has a dimension of 2048, and we choose a constant learning rate and a sample size of 1000.

In Figure 2a, we plot the logarithm of the mean and variance of the training error dynamic with respect to the $K$ (for clarity, we only plot the first 160 iterations), which includes the sampling step, local training steps, as well as the gradient aggregation step. Since the algorithm converges with a stable/similar manner across all trials, the variance is too small to be observed from the figure. Notice that there are three types of dynamics, as annotated in the figure: (1) *A smooth decrease of training error:* This corresponds to subnetworks' local training, which is supported by our theory that each subnetwork makes local progress. (2) *A sudden decrease of training error:* This corresponds to the aggregation of locally-trained subnetworks, and is consistent with our proof in Theorem 4. (3) *A sudden increase of training error:* This corresponds to re-sampling subnetworks; according to our theory, the expected average training error increases after sampling.

Figure 2b provides heatmap results that demonstrate the change of the error term as we vary the number of subnetworks, and the selection probability. In Figure 2b, the subnetworks are generated using Bernoulli masks, and the training process assumes a fixed number of local steps. Note that, as we fix the number of subnetworks and increase the selection probability, the error decreases (lighter colors in heatmap). Moreover, if we fix the number of selection probability and increase the number of subnetworks, the training error also decreases. This is consistent with Theorem 3.

Figure 2c studies how the error term changes as we increase the width of the neural network. In Theorem (3), we notice that the error term decreases as we choose a smaller initialization scale $\kappa$, while a smaller $\kappa$ would require a larger over-parameterization. This is consistent with our experiments in figure 2c: as we increase the number of hidden neurons and adjust the initialization scale, we observe that the training converges to a smaller error.

Figure 2d shows how the convergence rate changes as we increase the number of subnetworks under the categorical mask assumption. In particular, the y-axis denotes the training error improvement in the first, second, and third global step, respectively. The training error improvement is defined to be the $\frac{\text{training error in last step} - \text{training error in current step}}{\text{training error in last step}}$. We observe that the training error improvement decreases consistently across the first three global steps, as we increase the number of subnetworks. This corresponds to what we have shown in Theorem (4), where $1 - \gamma$ decreases as we increase $p$.

## 6 Conclusion

We prove linear convergence up to a neighborhood around the optimal point when training and combining subnetworks in a single hidden-layer perceptron scenario. Our work extends results on dropout, multi-sample dropout, and the Independent Subnet Training, and has broad implications on how the sampling method, the number of subnetworks, and the number of local steps affect the convergence rate and the size of the neighborhood around the optimal point. While our work focus on the single hidden-layer perceptron, we consider multi-layer perceptrons as an interesting direction: we conjecture that a more refined analysis of each layer's output is required *(Du et al., 2019; Allen-Zhu et al., 2019b)*. Moreover, focusing on the convergence of a stochastic algorithm for our framework, as well as considering different losses (e.g., classification tasks or even generic generalization losses) are interesting future research directions. Lastly, training with randomly sampled subnetwork may result into a regularization benefit. Theoretically studying how the proposed training scheme affects the generalization ability of the neural network is an interesting next step.

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

# A   Notation

Table 1: Notations

| SYMBOL | DESCRIPTION | MATHEMATICAL DEFINITION |
|---|---|---|
| $K$ | Number of global iterations | $K \in \mathbb{N}_+$ |
| $k$ | Index of global iterations | $k \in [K]$ |
| $\tau$ | Number of local iterations | $\tau \in \mathbb{N}_+$ |
| $t$ | Index of local iterations | $t \in [\tau]$ |
| $p$ | Number of subnetworks | $p \in \mathbb{N}_+$ |
| $l$ | Index of subnetworks | $l \in [p]$ |
| $\xi$ | Probability of selecting a neuron | $\xi \in (0,1]$ |
| $\boldsymbol{\xi}$ | Vector probability of selection a neuron by each worker | $\boldsymbol{\xi} \in (0,1]^p$ |
| $\eta$ | Constant step size for local gradient update | $\eta \in \mathbb{R}$ |
| $\mathbf{M}_k$ | Binary mask in iteration $k$ | $\mathbf{M}_k \in \{0,1\}^{p \times m}$ |
| $\mathbf{m}_{k,r}$ | Binary mask for neuron r in iteration $k$ | $\mathbf{m}_{k,r} \in \{0,1\}^p$, the vector of $r$th column of $\mathbf{M}_k$ |
| $\mathbf{m}_k^l$ | Binary mask for subnetwork $l$ in iteration $k$ | $\mathbf{m}_k^l \in \{0,1\}^m$, the vector of $l$th row of $\mathbf{M}_k$ |
| $m_{k,r}^l$ | Binary mask for neuron $r$ in subnetwork $l$ in iteration $k$ | $m_{k,r}^l \in \{0,1\}$ the $(l,r)$th entry of $\mathbf{M}_k$ |
| $X_{k,r}$ | Number of subnetworks selecting neuron $r$ in iteration $k$ | $X_{k,r} = \sum_{l=1}^p m_{k,r}^l$ |
| $N_{k,r}$ | Aggregated gradient normalizer for neuron $r$ in iteration $k$ | $N_{k,r} = \max\{X_{k,r}, 1\}$ |
| $N_{k,r}^\perp$ | Indicator of gradient existing for neuron $r$ in iteration $k$ | $N_{k,r}^\perp = \min\{X_{k,r}, 1\}$ |
| $\eta_{k,r}$ | Global gradient aggregation step size for neuron $r$ in iteration $k$ | $\eta_{k,r} = N_{k,r}^\perp / N_{k,r}$ |
| $u_k^{(i)}$ | Output of the whole network at global iteration $k$ for sample $i$ | $u_k^{(i)} = \frac{\xi}{\sqrt{m}} \sum_{r=1}^m a_r \sigma(\langle \mathbf{w}_{k,r}, \mathbf{x}_i \rangle)$ |
| $\mathbf{u}_k$ | Output of the whole network at global iteration $k$ for all $\mathbf{X}$ | $\mathbf{u}_k = \left[ u_k^{(1)}, \ldots, u_k^{(n)} \right]$ |
| $\hat{u}_{k,t}^{l(i)}$ | Output of subnetwork $l$ at iteration $(k,t)$ for sample $i$ | $\hat{u}_{k,t}^{l(i)} = \frac{1}{\sqrt{m}} \sum_{r=1}^m a_r m_{k,r}^l \sigma\left( \left\langle \mathbf{w}_{k,t,r}^l, \mathbf{x}_i \right\rangle \right)$ |
| $\hat{\mathbf{u}}_{k,t}^l$ | Output of subnetwork $l$ at iteration $(k,t)$ for all $\mathbf{X}$ | $\hat{\mathbf{u}}_{k,t}^l = \left[ \hat{u}_{k,t}^{l(1)}, \ldots, \hat{u}_{k,t}^{l(n)} \right]$ |
| $\hat{u}_k^{l(i)}$ | Output of subnetwork $l$ at iteration $(k,0)$ for sample $i$ | $\hat{u}_k^{l(i)} = \hat{u}_{k,0}^{l(i)}$ |
| $\hat{\mathbf{u}}_k^l$ | Output of subnetwork $l$ at iteration $(k,0)$ for all $\mathbf{X}$ | $\hat{\mathbf{u}}_k^l = \hat{\mathbf{u}}_{k,0}^l$ |
| $L_k$ | Global loss at iteration $k$ | $L_k = \|\mathbf{y} - \mathbf{u}_k\|_2^2$ |
| $L_{\mathbf{m}_k^l}(\mathbf{W}_{k,t}^l)$ | Local loss for subnetwork $l$ at iteration $(k,t)$ | $L_{\mathbf{m}_k^l}(\mathbf{W}_{k,t}^l) = \|\mathbf{y} - f_{\mathbf{m}_k^l}(\mathbf{W}_{k,t}^l)\|_2^2$ |

## B    Preliminary and Definition

In the proofs of our theorems, we use extensively the following tools.

**Definition 2.** *(Sub-Gaussian Random Variable) A random variable $X$ is $\kappa^2$-sub-Gaussian if*

$$\mathbb{E}[e^{tx}] \leq e^{\frac{\kappa^2 t^2}{2}}$$

**Definition 3.** *(Sub-Exponential Random Variable) A random variable with mean $\mathbb{E}[X] = \mu$ is $(\kappa', \alpha)$-sub-exponential if there exists non-negative $(\kappa', \alpha)$ such that for all $t \leq \alpha^{-1}$*

$$\mathbb{E}[e^{t(X-\mu)}] \leq e^{-\frac{t^2 \kappa'^2}{2}}$$

**Property 1.** *(Sub-Exponential Tail Bound) For a $(\kappa', \alpha)$-sub-exponential random variable $X$ with $\mathbb{E}[X] = \mu$ we have*

$$P(X > \mu + t) \leq \begin{cases} e^{-\frac{t^2}{2\kappa'^2}} & \text{if } 0 \leq t \leq \frac{\kappa'^2}{\alpha} \\ e^{-\frac{t^2}{2\alpha}} & \text{if } t > \frac{\kappa'^2}{\alpha} \end{cases}$$

**Property 2.** *(Markov's Inequality) For a non-negative random variable $X$, we have*

$$P(X \geq a) \leq \frac{1}{a}\mathbb{E}[X]$$

**Property 3.** *(Hoeffding's Inequality for Bounded Random Variables) Let $X_1, \ldots, X_n$ be independent random variables bounded by $|X_i| \leq 1$ for all $i \in [n]$. Then we have*

$$P\left(\left|\frac{1}{n}\sum_{i=1}^{n} X_i\right| \geq t\right) \leq e^{-2nt^2}$$

**Property 4.** *(Berstein's Inequality) Let $X_1, \ldots, X_n$ be random variables with $\mathbb{E}[X_i] = 0$ for all $i \in [n]$. If $|X_i| \leq M$ almost surely, then*

$$P\left(\sum_{i=1}^{n} X_i > t\right) \leq e^{-\frac{t^2/2}{\sum_{j=1}^{n} \mathbb{E}[X_j^2] + Mt/3}}$$

**Property 5.** *(Jensen's Inequality for Expectation) For a non-negative random variable $X$, we have*

$$\mathbb{E}\left[X^{\frac{1}{2}}\right] \leq (\mathbb{E}[X])^{\frac{1}{2}}$$

Apart from the properties above, we also need the following definitions to facilitate our analysis. First, we note that, in the following proofs, we let $R = \frac{\kappa \lambda_0}{192n}$. Define

$$A_{ir} = \{\exists \mathbf{w} \in \mathcal{B}(\mathbf{w}_{0,r}, R) : \mathbb{I}\{\langle \mathbf{w}, \mathbf{x}_i \rangle \geq 0\} \neq \mathbb{I}\{\langle \mathbf{w}_{0,r}, \mathbf{x}_i \rangle \geq 0\}\}$$

to denote the event that for sample $\mathbf{x}_i$, the activation pattern of neuron $r$ may change through training if the weight vector change is bounded in the $R$-ball centered at initialization. Moreover, let

$$S_i = \{r \in [m] : \neg A_{ir}\}$$
$$S_i^{\perp} = [m] \setminus S_i$$

to be the set of neurons whose activation pattern does not change for sample $\mathbf{x}_i$ if the weight vector change is bounded by the $R$-ball centered at initialization. Moreover, since we are interested in the loss dynamic computed on the following function

$$u_k^{(i)} = \frac{\xi}{\sqrt{m}} \sum_{r=1}^{m} a_r \sigma(\langle \mathbf{w}_{k,r}, \mathbf{x}_i \rangle).$$

we denote the full gradient of loss with respect to each weight vector $\mathbf{w}_r$ as

$$\frac{\partial L(\mathbf{W}_k)}{\partial \mathbf{w}_r} = \frac{\xi}{\sqrt{m}} \sum_{i=1}^{n} a_r \mathbf{x}_i (u_k^{(i)} - y_i) \mathbb{I}\{\langle \mathbf{w}_{k,r}, \mathbf{x}_i \rangle \geq 0\}$$

## C   Proof of Theorem 2

Recall the definition of masked-NTK

$$(\mathbf{m}_{k'}^l \circ \mathbf{H}(k,t))_{ij} = \frac{1}{m} \langle \mathbf{x}_i, \mathbf{x}_j \rangle \sum_{r=1}^m m_{k',r}^l \mathbb{I}\{\langle \mathbf{w}_{k,t,r}, \mathbf{x}_i \rangle \geq 0, \langle \mathbf{w}_{k,t,r}, \mathbf{x}_i \rangle \geq 0\}$$

To start with, we fix $k' \in [K], l \in [p]$, and $i, j \in [n]$. In this case, let

$$h_r = m_{k',r}^l \langle \mathbf{x}_i, \mathbf{x}_j \rangle \mathbb{I}\{\langle \mathbf{w}_{0,r}, \mathbf{x}_i \rangle \geq 0, \langle \mathbf{w}_{0,r}, \mathbf{x}_i \rangle \geq 0\}$$

Then we have

$$(\mathbf{m}_{k'}^l \circ \mathbf{H}(0,0))_{ij} = \frac{1}{m} \sum_{r=1}^m h_r$$

Also we have

$$\mathbb{E}_{\mathbf{M}_k,\mathbf{W}} [h_r] = \mathbb{E}_{\mathbf{w}\sim\mathcal{N}(0,\mathbf{I})} [\mathbb{E}_{\mathbf{M}_k} [h_r]] = \mathbf{H}_{ij}^\infty$$

Note that for all $r$ we have $|h_r| \leq 1$. Thus we apply Hoeffding's inequality for bounded random variables and get

$$P\left(\left|(\mathbf{m}_{k'}^l \circ \mathbf{H}(0,0))_{ij} - \mathbf{H}_{ij}^\infty\right| \geq t\right) = P\left(\left|\frac{1}{m} \sum_{r=1}^m h_r - \mathbf{H}_{ij}^\infty\right| \geq t\right) \leq 2e^{-2mt^2}$$

Apply a union bound over $i, j$ gives that with probability at least $1 - 2n^2 e^{-2mt^2}$ it holds that

$$\left|(\mathbf{m}_{k'}^l \circ \mathbf{H}(0,0))_{ij} - \mathbf{H}_{ij}^\infty\right| \leq t$$

for all $i, j \in [n]$. Therefore,

$$\left\|\mathbf{m}_{k'}^l \circ \mathbf{H}(0,0) - \mathbf{H}^\infty\right\|_2^2 \leq \left\|\mathbf{m}_{k'}^l \circ \mathbf{H}(0,0) - \mathbf{H}^\infty\right\|_F^2$$

$$\leq \sum_{i,j=1}^n \left|(\mathbf{m}_{k'}^l \circ \mathbf{H}(0,0))_{ij} - \mathbf{H}_{ij}^\infty\right|^2$$

$$\leq n^2 t^2$$

Let $t = \frac{\lambda_0}{4n}$ gives

$$\left\|\mathbf{m}_{k'}^l \circ \mathbf{H}(0,0) - \mathbf{H}^\infty\right\|_2 \leq \frac{\lambda_0}{4}$$

holds with probability at least $1 - 2n^2 e^{-\frac{m\lambda_0^2}{8n^2}}$. Next we show that for all $k \in [k]$ and $t \in [\tau]$, as long as $\|\mathbf{w}_{k,t,r} - \mathbf{w}_{0,r}\|_2 \leq R$ for all $r \in [m]$, then it holds that

$$\left\|\mathbf{m}_{k'}^l \circ \mathbf{H}(k,t) - \mathbf{m}_{k'}^l \circ \mathbf{H}(0,0)\right\|_2 \leq 2n\kappa^{-1}R$$

Following the argument of *(Song & Yang, 2020)*, lemma 3.2, we have

$$\left\|\mathbf{m}_{k'}^l \circ \mathbf{H}(k,t) - \mathbf{m}_{k'}^l \circ \mathbf{H}(0,0)\right\|_F^2 \leq \frac{1}{m^2} \sum_{i,j=1}^n \left(\sum_{r=1}^m s_{r,i,j}\right)^2$$

with

$$s_{r,i,j} = m_{k',r}^l \left(\mathbb{I}\{\langle \mathbf{w}_{0,r}, \mathbf{x}_i \rangle \geq 0; \langle \mathbf{w}_{0,r}, \mathbf{x}_i \rangle \geq 0\} - \mathbb{I}\{\langle \mathbf{w}_{r,k,t}, \mathbf{x}_j \rangle \geq 0; \langle \mathbf{w}_{r,k,t}, \mathbf{x}_j \rangle \geq 0\}\right)$$

Then $s_{r,i,j} = 0$ if $\neg A_{ir}$ and $\neg A_{jr}$ happend. In other cases we have $|s_{r,i,j}| \leq 1$. Thus we have that for all $i, j \in [n]$

$$\mathbb{E}_{\mathbf{M}_k, \mathbf{W}_0}[s_{r,i,j}] = \xi P\left(A_{ir} \cup A_{jr}\right) \leq \frac{4\xi R}{\kappa\sqrt{2\pi}} \leq 2\xi\kappa^{-1}R$$

and

$$\mathbb{E}_{\mathbf{M}_k, \mathbf{W}_0}\left[(s_{r,i,j} - \mathbb{E}_{\mathbf{M}_k, \mathbf{W}_0}[s_{r,i,j}])^2\right] \leq \mathbb{E}_{\mathbf{M}_k, \mathbf{w}_{0,r}}[s_{r,i,j}^2] \leq \frac{4\xi R}{\kappa\sqrt{2\pi}} \leq 2\xi\kappa^{-1}R$$

Thus applying Bernstein inequality with $t = \xi\kappa^{-1}R$ gives

$$P\left(\frac{1}{m^2}\sum_{r=1}^{m} s_{r,i,j} \geq 3\xi\kappa^{-1}R\right) \leq \exp\left(-\frac{m\xi R}{10\kappa}\right)$$

Therefore, taking a union bound gives that, with probability at least $1 - n^2 e^{-\frac{m\xi R}{10\kappa}}$ we have that

$$\left\|\mathbf{m}_{k'}^l \circ \mathbf{H}(k,t) - \mathbf{m}_{k'}^l \circ \mathbf{H}(0,0)\right\|_2 \leq \left\|\mathbf{m}_{k'}^l \circ \mathbf{H}(k,t) - \mathbf{m}_{k'}^l \circ \mathbf{H}(0,0)\right\|_F \leq 3\xi n\kappa^{-1}R$$

Using $R \leq \frac{\kappa\lambda_0}{12n}$ gives $\|\mathbf{m}_{k'}^l \circ \mathbf{H}(k,t) - \mathbf{m}_{k'}^l \circ \mathbf{H}(0,0)\|_2 \leq \frac{\xi\lambda_0}{4} \leq \frac{\lambda_0}{4}$ with probability at least $1 - n^2 e^{-\frac{m\xi\lambda_0}{12n}}$. Therefore, we have

$$\left\|\mathbf{m}_{k'}^l \circ \mathbf{H}(k,t) - \mathbf{H}^\infty\right\|_2 \leq \frac{\lambda_0}{2}$$

which implies that $\lambda_{\min}\left(\mathbf{m}_{k'}^l \circ \mathbf{H}(k,t)\right) \geq \frac{\lambda_0}{2}$ holds with probability at least $1 - n^2\left(e^{-\frac{m\xi\lambda_0}{12n}} - 2e^{-\frac{m\lambda_0^2}{8n^2}}\right)$ for a fixed $k' \in [K]$ and $l \in [p]$. Taking a union bound over all $k'$ and $l$ and plugging in the requirement $m = \Omega\left(\frac{n^2 \log Kpn/\delta}{\xi\lambda_0}\right)$ gives the desired result.

# D   Proof of Hypothesis 1

In this proof, we follow the idea of *(Du et al., 2018)*. However, the difference is that $i$) we use our masked-NTK during the analysis, and $ii$) we use a different technique for bounding the weight perturbation. We repeat the key requirement stated in the theorem here: for all $r \in [m]$

$$\|\mathbf{w}_{k,r} - \mathbf{w}_{0,r}\|_2 + 2\eta\tau\sqrt{\frac{nK}{m\delta}}\mathbb{E}_{[\mathbf{M}_{k-1}],\mathbf{W}_0,\mathbf{a}}\left[\|\mathbf{y} - \mathbf{u}_k\|_2\right] + (K - k)\kappa\sqrt{\xi(1-\xi)pn} \le R \tag{10}$$

To start, we notice that, using the required over-parameterization, lemma 23 holds with probability at least $1 - O(\delta)$. We use induction on the following two conditions to prove the theorem:

$$\left\|\mathbf{y} - \mathbf{u}_{k,t+1}^l\right\|_2^2 \le \left(1 - \frac{\eta\lambda_0}{2}\right)\left\|\mathbf{y} - \mathbf{u}_{k,t}^l\right\|_2^2 \tag{11}$$

$$\left\|\mathbf{w}_{k,t,r}^l - \mathbf{w}_{k,r}\right\| \le \frac{\eta\tau\sqrt{2nK}}{\sqrt{m\delta}}\mathbb{E}_{[\mathbf{M}_{k-1}],\mathbf{W}_0,\mathbf{a}}\left[\|\mathbf{y} - \mathbf{u}_k\|_2\right] + 2\eta\kappa n\sqrt{\frac{2\xi(1-\xi)pK}{m\delta}} \tag{12}$$

$$\left\|\mathbf{w}_{k,t,r}^l - \mathbf{w}_{0,r}\right\| \le R \tag{13}$$

**Base Case**: For the case of $t = 0$, we notice that equation (11) and (12) naturally holds. Moreover, equation (10) implies equation (12).

**Inductive Case**: the inductive case is divided into three parts.

**(12)→(13)**: assume that equation (12) holds in local iteration $t$. Combine the result with equation (10) gives that equation (13) holds in iteration $t$.

**(13→11)**: assume that equation (13) holds in local iteration $t$. We are going to prove that equation (11) holds. In particular, we are interested in

$$\|\mathbf{y} - \hat{\mathbf{u}}_{k,t+1}^l\|_2^2 = \|\mathbf{y} - \hat{\mathbf{u}}_{k,t}^l\|_2^2 - 2\left\langle\mathbf{y} - \mathbf{u}_{k,t}^l, \hat{\mathbf{u}}_{k,t+1}^l - \hat{\mathbf{u}}_{k,t}^l\right\rangle + \|\hat{\mathbf{u}}_{k,t+1}^l - \hat{\mathbf{u}}_{k,t}^l\|_2^2$$

We define $\hat{u}_{k,t+1}^{l(i)} - \hat{u}_{k,t}^{l(i)} = I_{1,k,t}^{l(i)} + I_{2,k,t}^{l(i)}$ with

$$I_{1,k,t}^{l(i)} = \frac{1}{\sqrt{m}}\sum_{r \in S_i} a_r m_{k,r}^l \left(\sigma\left(\langle\mathbf{w}_{k,t+1,r}^l, \mathbf{x}_i\rangle\right) - \sigma\left(\langle\mathbf{w}_{k,t,r}^l, \mathbf{x}_i\rangle\right)\right)$$

$$I_{2,k,t}^{l(i)} = \frac{1}{\sqrt{m}}\sum_{r \in S_i^{\perp}} a_r m_{k,r}^l \left(\sigma\left(\langle\mathbf{w}_{k,t+1,r}^l, \mathbf{x}_i\rangle\right) - \sigma\left(\langle\mathbf{w}_{k,t,r}^l, \mathbf{x}_i\rangle\right)\right)$$

and notice that, with the 1-Lipchitzness of ReLU,

$$\left|I_{2,k,t}^{l(i)}\right| \le \frac{1}{\sqrt{m}}\sum_{r \in S_i^{\perp}}\left|\sigma\left(\langle\mathbf{w}_{k,t+1,r}^l, \mathbf{x}_i\rangle\right) - \sigma\left(\langle\mathbf{w}_{k,t,r}^l, \mathbf{x}_i\rangle\right)\right|$$

$$\le \frac{1}{\sqrt{m}}\sum_{r \in S_i^{\perp}}\left\|\mathbf{w}_{k,t+1,r}^l - \mathbf{w}_{k,t,r}\right\|_2$$

$$\le \frac{\eta}{\sqrt{m}}\sum_{r \in S_i^{\perp}}\left\|\frac{\partial L_{\mathbf{m}_k^l}\left(\mathbf{W}_{k,t}^l\right)}{\partial\mathbf{w}_r}\right\|_2$$

$$\le \frac{\eta\sqrt{n}}{m}\sum_{r \in S_i^{\perp}}\left\|\mathbf{y} - \hat{\mathbf{u}}_{k,t}^l\right\|_2$$

$$\le 4\eta\kappa^{-1}\sqrt{n}R\|\mathbf{y} - \hat{\mathbf{u}}_{k,t}^l\|_2$$

where the last inequality uses $|S_i^\perp| \leq 4m\kappa^{-1}R$ from Lemma 16. Therefore,

$$\left|\left\langle \mathbf{y} - \hat{\mathbf{u}}_{k,t}^l, \mathbf{I}_{2,k,t}^l \right\rangle\right| \leq \sqrt{n} \max_{i \in [n]} \left|I_{2,k,t}^{l(i)}\right| \cdot \|\mathbf{y} - \hat{\mathbf{u}}_{k,t}^l\|_2 \leq 4\eta\kappa^{-1}nR\|\mathbf{y} - \hat{\mathbf{u}}_{k,t}^l\|_2^2$$

Similarly, we have

$$\left(\hat{u}_{k,t+1}^{l(i)} - \hat{u}_{k,t}^{l(i)}\right)^2 \leq \frac{1}{m}\left(\sum_{r=1}^m \left\|\mathbf{w}_{k,t+1,r}^l - \mathbf{w}_{k,t,r}^l\right\|_2\right)^2$$

$$\leq \sum_{r=1}^m \left\|\mathbf{w}_{k,t+1,r}^l - \mathbf{w}_{k,t,r}^l\right\|_2^2$$

$$\leq \eta^2 \sum_{r=1}^m \left\|\frac{\partial L_{\mathbf{m}_k^l}\left(\mathbf{W}_{k,t}^l\right)}{\partial \mathbf{w}_r}\right\|_2^2$$

$$\leq \eta^2 n^2 \|\mathbf{y} - \hat{\mathbf{u}}_{k,t}^l\|_2^2$$

Lastly, we define $\mathbf{m}_k^l \circ \mathbf{H}(k,t)^\perp$ with

$$\left(\mathbf{m}_k^l \circ \mathbf{H}(k,t)^\perp\right)_{ij} = \frac{1}{m}\left\langle \mathbf{x}_i, \mathbf{x}_j \right\rangle \sum_{r \in S_i^\perp} m_{k,r}^l \mathbb{I}\{\langle \mathbf{w}_{k,r}, \mathbf{x}_i\rangle \geq 0, \langle \mathbf{w}_{k,r}, \mathbf{x}_j\rangle \geq 0\}$$

and we have

$$I_{1,k,t}^{l(i)} = \frac{1}{\sqrt{m}} \sum_{r \in S_i} a_r m_{k,r}^l \left\langle \mathbf{w}_{k,t+1,r}^l - \mathbf{w}_{k,t,r}^l, \mathbf{x}_i\right\rangle \mathbb{I}\{\langle \mathbf{w}_{k,r}, \mathbf{x}_i\rangle \geq 0\}$$

$$= -\frac{\eta}{\sqrt{m}} \sum_{r \in S_i} a_r m_{k,r}^l \left\langle \frac{\partial L_{\mathbf{m}_k^l}\left(\mathbf{W}_{k,t}^l\right)}{\partial \mathbf{w}_r}, \mathbf{x}_i\right\rangle \mathbb{I}\{\langle \mathbf{w}_{k,r}, \mathbf{x}_i\rangle \geq 0\}$$

$$= \frac{\eta}{m} \sum_{r \in S_i} \sum_{j=1}^n m_{k,r}^l \left(y_j - \hat{u}_{k,t}^{l(j)}\right) \langle \mathbf{x}_i, \mathbf{x}_j\rangle \mathbb{I}\{\langle \mathbf{w}_{k,r}, \mathbf{x}_i\rangle \geq 0, \langle \mathbf{w}_{k,r}, \mathbf{x}_j\rangle \geq 0\}$$

$$= \eta \sum_{j=1}^n \left(\mathbf{m}_k^l \circ \mathbf{H}(k,t) - \mathbf{m}_k^l \circ \mathbf{H}(k,t)^\perp\right)_{ij} \left(y_j - \hat{u}_{k,t}^{l(j)}\right)$$

Therefore,

$$\left\langle \mathbf{y} - \hat{\mathbf{u}}_{k,t}^l, \mathbf{I}_{1,k,t}^l \right\rangle = \eta \sum_{i,j=1}^n \left(y_i - \hat{u}_{k,t}^{l(i)}\right)\left(\mathbf{m}_k^l \circ \mathbf{H}(k,t) - \mathbf{m}_k^l \circ \mathbf{H}(k,t)^\perp\right)_{ij}\left(y_j - \hat{u}_{k,t}^{l(j)}\right)$$

$$= \eta \left\langle \mathbf{y} - \hat{\mathbf{u}}_{k,t}, \left(\mathbf{m}_k^l \circ \mathbf{H}(k,t) - \mathbf{m}_k^l \circ \mathbf{H}(k,t)^\perp\right)(\mathbf{y} - \hat{\mathbf{u}}_{k,t})\right\rangle$$

$$\geq \frac{\eta\lambda_0}{2}\|\mathbf{y} - \hat{\mathbf{u}}_{k,t}^l\|_2^2 - \eta\|\mathbf{m}_k^l \circ \mathbf{H}(k,t)^\perp\|_2\|\mathbf{y} - \hat{\mathbf{u}}_{k,t}^l\|_2^2$$

$$\geq \left(\frac{\eta\lambda_0}{2} - 4\eta\kappa^{-1}nR\right)\|\mathbf{y} - \hat{\mathbf{u}}_{k,t}^l\|_2^2$$

where the last inequality follows from the fact that

$$\|\mathbf{m}_k^l \circ \mathbf{H}(k,t)^\perp\|_2^2 \leq \|\mathbf{m}_k^l \circ \mathbf{H}(k,t)^\perp\|_F^2$$

$$= \frac{1}{m^2} \sum_{i,j=1}^n \left(\langle \mathbf{x}_i, \mathbf{x}_j\rangle \sum_{r \in S_i^\perp} \mathbb{I}\{\langle \mathbf{w}_{k,r}, \mathbf{x}_i\rangle \geq 0, \langle \mathbf{w}_{k,r}, \mathbf{x}_j\rangle \geq 0\}\right)^2$$

$$\leq \frac{n^2}{m^2}|S_i^\perp|^2$$

$$= 16n^2\kappa^{-2}R^2$$

Putting things together gives

$$\|\mathbf{y} - \hat{\mathbf{u}}_{k,t+1}^l\|_2^2 \leq \left(1 - \eta\lambda_0 + 16\eta\kappa^{-1}nR + \eta^2 n^2\right) \|\mathbf{y} - \hat{\mathbf{u}}_{k,t}^l\|_2^2$$

Choose $R \leq \frac{\kappa\lambda_0}{64n}$ and $\eta \leq \frac{\lambda_0}{4n^2}$ gives

$$\|\mathbf{y} - \hat{\mathbf{u}}_{k,t+1}^l\|_2^2 \leq \left(1 - \frac{\eta\lambda_0}{2}\right) \|\mathbf{y} - \hat{\mathbf{u}}_{k,t}^l\|_2^2$$

**(11)→(12)**: Assume that equation (11) holds for local iteration $0, \ldots, t$. We are going to prove equation (12) for local iteration $t + 1$. We start by noticing that equation (11) implies that for all local iteration $t' \in [t]$, we have that $\|\mathbf{y} - \hat{\mathbf{u}}_{k,t'}^l\|_2 \leq \|\mathbf{y} - \hat{\mathbf{u}}_k^l\|_2$. Moreover, we notice that

$$\left\|\frac{\partial L_{\mathbf{m}_k}\left(\mathbf{W}_{k,t}^l\right)}{\partial \mathbf{w}_r}\right\|_2 \leq \frac{1}{\sqrt{m}} \sum_{i=1}^{n} \left\|\left(\hat{u}_{k,t}^{l,(i)} - y_i\right) a_r m_{k,r}^l \mathbf{x}_i \mathbb{I}\{\langle \mathbf{w}_{k,t,r}^l, \mathbf{x}_i\rangle\}\right\|_2$$

$$\leq \frac{1}{\sqrt{m}} \sum_{i=1}^{n} \left|\hat{u}_{k,t}^{l,(i)} - y_i\right|$$

$$\leq \frac{\sqrt{n}}{\sqrt{m}} \left\|\mathbf{y} - \hat{\mathbf{u}}_{k,t}^l\right\|_2$$

Therefore,

$$\|\mathbf{w}_{k,t,r} - \mathbf{w}_{k,r}\|_2 \leq \eta \sum_{t'=0}^{t-1} \left\|\frac{\partial L_{\mathbf{m}_k}\left(\mathbf{W}_{k,t}^l\right)}{\partial \mathbf{w}_r}\right\|_2$$

$$\leq \eta\frac{\sqrt{n}}{\sqrt{m}} \sum_{t'=0}^{t-1} \|\mathbf{y} - \hat{\mathbf{u}}_{k,t'}^l\|_2$$

$$\leq \eta\tau\frac{\sqrt{n}}{\sqrt{m}} \|\mathbf{y} - \hat{\mathbf{u}}_k^l\|_2$$

$$\leq \eta\tau\frac{\sqrt{n}}{\sqrt{m}} \left(\|\mathbf{y} - \mathbf{u}_k\|_2 + \|\mathbf{u}_k - \hat{\mathbf{u}}_k^l\|_2\right)$$

Applying Markov's inequality to the global convergence, with probabiltiy at least $1 - \frac{\delta}{2K}$, it holds that

$$\|\mathbf{y} - \mathbf{u}_k\|_2 \leq \sqrt{2K/\delta}\mathbb{E}_{[\mathbf{M}_{k-1}],\mathbf{W}_0,\mathbf{a}}\left[\|\mathbf{y} - \mathbf{u}_k\|_2\right]$$

By Lemma 25, we have

$$\mathbb{E}_{\mathbf{M}_k}\left[\|\mathbf{u}_k - \hat{\mathbf{u}}_k^l\|_2^2\right] \leq 4\xi(1-\xi)n\kappa^2$$

Thus with probability at least $1 - \frac{\delta}{2pK}$ it holds that

$$\|\mathbf{u}_k - \hat{\mathbf{u}}_k^l\|_2 \leq 2\kappa\sqrt{2\xi(1-\xi)npK/\delta}$$

Plugging in gives

$$\|\mathbf{w}_{k,t,r} - \mathbf{w}_{k,r}\| \leq \frac{\eta\tau\sqrt{2nK}}{\sqrt{m\delta}}\mathbb{E}_{[\mathbf{M}_{k-1}],\mathbf{W}_0,\mathbf{a}}\left[\|\mathbf{y} - \mathbf{u}_k\|_2\right] + 2\eta\tau\kappa n\sqrt{\frac{2\xi(1-\xi)pK}{m\delta}}$$

which completes the proof.

# E   Proof of Theorem 3

Before we start the proof, we introduce several notations. Define

$$I_{1,k}^{(i)} = \frac{\xi}{\sqrt{m}} \sum_{r \in S_i} a_r \left( \sigma(\langle \mathbf{w}_{k+1,r}, \mathbf{x}_i \rangle) - \sigma(\langle \mathbf{w}_{k,r}, \mathbf{x}_i \rangle) \right)$$

$$I_{2,k}^{(i)} = \frac{\xi}{\sqrt{m}} \sum_{r \in S_i^{\perp}} a_r \left( \sigma(\langle \mathbf{w}_{k+1,r}, \mathbf{x}_i \rangle) - \sigma(\langle \mathbf{w}_{k,r}, \mathbf{x}_i \rangle) \right)$$

Let

$$\mathbf{I}_{1,k} = \left[ I_{1,k}^{(1)}, \dots, I_{1,k}^{(n)} \right]$$

and similarly,

$$\mathbf{I}_{2,k} = \left[ I_{2,k}^{(1)}, \dots, I_{2,k}^{(n)} \right]$$

Then we have $u_{k+1}^{(i)} - u_k^{(i)} = I_1^{(i)} + I_2^{(i)}$ and $\mathbf{u}_{k+1} - \mathbf{u}_k = \mathbf{I}_{1,k} + \mathbf{I}_{2,k}$. Also, we define $\mathbf{H}(k)^{\perp}$ to be

$$\mathbf{H}(k)_{ij}^{\perp} = \frac{\xi}{m} \sum_{r \in S_i^{\perp}} \langle \mathbf{x}_i, \mathbf{x}_j \rangle \, \mathbb{I}\{\langle \mathbf{w}_{k,r}, \mathbf{x}_i \rangle \geq 0, \langle \mathbf{w}_{k,r}, \mathbf{x}_j \rangle \geq 0\}$$

For $k$-th global iteration, first local iteration, we define the mixing gradient as

$$\begin{aligned}
\mathbf{g}_{k,r} &= \eta_{k,r} \sum_{l=1}^{p} \frac{\partial L_{\mathbf{m}_k^l}\left(\mathbf{W}_{k,0}^l\right)}{\partial \mathbf{w}_r} \\
&= \eta_{k,r} \sum_{l=1}^{p} \frac{\partial L_{\mathbf{m}_k^l}\left(\mathbf{W}_k\right)}{\partial \mathbf{w}_r} \\
&= \frac{\eta_{k,r}}{\sqrt{m}} \sum_{l=1}^{p} \sum_{i=1}^{n} m_{k,r}^l (\hat{u}_k^{l(i)} - y_i) a_r \mathbf{x}_i \mathbb{I}\{\langle \mathbf{w}_{k,r}, \mathbf{x}_i \rangle \geq 0\} \\
&= \frac{1}{\sqrt{m}} \sum_{i=1}^{m} (f_{k,r}^{(i)} - N_{k,r}^{\perp} y_i) a_r \mathbf{x}_i \mathbb{I}\{\langle \mathbf{w}_{k,r}, \mathbf{x}_i \rangle \geq 0\}
\end{aligned}$$

where we define the mixing function as

$$f_{k,r}^{(i)} = \eta_{k,r} \sum_{l=1}^{p} m_{k,r}^l \hat{u}_k^{l(i)}$$

As usual, we let $\mathbf{f}_{k,r} = \left[ f_{k,r}^{(1)}, \dots, f_{k,r}^{(n)} \right]$. We note that $f_{k,r}^{(i)}$ has the form

$$\begin{aligned}
f_{k,r}^{(i)} &= \eta_{k,r} \sum_{l=1}^{p} m_{k,r}^l \hat{u}_k^{(l(i)} \\
&= \frac{1}{\sqrt{m}} \sum_{r'=1}^{m} a_r \left( \eta_{k,r} \sum_{l=1}^{p} m_{k,r}^l m_{k,r'}^l \right) \sigma(\langle \mathbf{w}_{k,r}, \mathbf{x}_i \rangle)
\end{aligned}$$

Let $\nu_{k,r,r'} = \eta_{k,r} \sum_{l=1}^{p} m_{k,r}^l m_{k,r'}^l$. The mixing function reduce to the form

$$f_{k,r}^{(i)} = \frac{1}{\sqrt{m}} \sum_{r=1}^{m} a_r \nu_{k,r,r'} \sigma(\langle \mathbf{w}_{k,r}, \mathbf{x}_i \rangle)$$

Also, note that if $N_{k,r}^\perp = 0$, we have $\nu_{k,r,r'} = 0$. We prove Theorem 3 by a fashion of induction, with the two conditions we consider stated below:

$$\mathbb{E}_{[\mathbf{M}_{k-1}]}\left[\|\mathbf{y} - \mathbf{u}_k\|_2^2\right] \leq \left(1 - \frac{1}{4}\eta\theta\tau\lambda_0\right)^k \|\mathbf{y} - \mathbf{u}_0\|_2^2 + B_1 \tag{14}$$

$$\|\mathbf{w}_{k,r} - \mathbf{w}_{0,r}\|_2 + 2\eta\tau\sqrt{\frac{2nK}{m\delta}}\left(\frac{4}{\eta\theta\tau\lambda_0}\mathbb{E}_{[\mathbf{M}_{k-1}]}\left[\|\mathbf{y} - \mathbf{u}_k\|_2\right] + (K - k)B\right) \leq R \tag{15}$$

$$\left\|\mathbf{w}_{k,t,r}^l - \mathbf{w}_{0,r}\right\|_2 \leq R \tag{16}$$

with

$$B = \sqrt{\frac{4B_1}{\eta\theta\tau\lambda_0}} + \kappa\sqrt{\xi(1-\xi)pn}$$

**Base Case:** Note that equation (14) and equation (16) holds naturally for $t = 0$. To show that equation (15) holds, we need to use the over-parameterization property. In particular, we want to show that

$$2\eta\tau\sqrt{\frac{2nK}{m\delta}}\left(\frac{4}{\eta\theta\tau\lambda_0}\mathbb{E}_{\mathbf{W}_0,\mathbf{a}}\left[\|\mathbf{y} - \mathbf{u}_0\|_2\right] + KB\right) \leq R \leq \frac{\kappa\lambda_0}{144n}$$

Apply lemma 26 and move the factor on the left hand side of the equation to the right. Then we equivalently want

$$\frac{\kappa\lambda_0}{\eta\tau}\sqrt{\frac{m\delta}{n^3K}} = \Omega\left(\max\left\{\frac{4C^2n}{\eta\theta\tau\lambda_0}, KB\right\}\right)$$

Plugging in the value of $B$ and the requirement of $\eta$, and solve for $m$ to get that

$$m = \Omega\left(\frac{K}{\delta}\max\left\{\frac{n^4}{\kappa^2\xi\theta\lambda_0^4}, \frac{nK^2B_1}{\kappa^2\theta\lambda_0^2}, K^2p\right\}\right)$$

**Inductive Case:** again the inductive case is divided into three parts.

**(15)→(16)** Observe that $B \geq \kappa\sqrt{\xi(1-\xi)pn}$. Thus, if equation (15) is satisfied and the over-parameterization requirement holds, then Hypothesis (1) holds. Thus equation (16) holds naturally.

**(14)→(15)**. Assume that equation (14) holds, and (15) holds for global iteration $k$, we want to show that equation (15) holds for global iteration $t + 1$. In particular, we would like to show that

$$\|\mathbf{w}_{k+1,r} - \mathbf{w}_{0,r}\|_2 + 2\eta\tau\sqrt{\frac{2nK}{m\delta}}\left(\frac{4}{\eta\theta\tau\lambda_0}\mathbb{E}_{[\mathbf{M}_k],\mathbf{W}_0,\mathbf{a}}\left[\|\mathbf{y} - \mathbf{u}_{k+1}\|_2\right] + (K - k - 1)B\right) \leq R$$

it suffice to show that

$$\|\mathbf{w}_{k+1,r} - \mathbf{w}_{k,r}\|_2 \leq \eta\tau\sqrt{\frac{2nK}{m\delta}}\mathbb{E}_{[\mathbf{M}_{k-1}],\mathbf{W}_0,\mathbf{a}}\left[\|\mathbf{y} - \mathbf{u}_k\|_2\right] + 2\eta\tau\sqrt{\frac{2nK}{m\delta}}B - 2\eta\tau\sqrt{\frac{8nKB_1}{m\delta\eta\theta\tau\lambda_0}}$$

Recall the definition of $B$ as

$$B = \sqrt{\frac{B_1}{\alpha}} + \kappa\sqrt{\xi(1-\xi)pn}$$

It then suffice to show that

$$\|\mathbf{w}_{k+1,r} - \mathbf{w}_{0,r}\|_2 \leq \eta\tau\sqrt{\frac{2nK}{m\delta}}\mathbb{E}_{[\mathbf{M}_{k-1}],\mathbf{W}_0,\mathbf{a}}\left[\|\mathbf{y} - \mathbf{u}_k\|_2\right] + 2\eta\tau\kappa n\sqrt{\frac{2\xi(1-\xi)pK}{m\delta}}$$

Note that under these two conditions and the over-parameterization requirement, Hypothesis (1) holds. Thus, we have

$$\|\mathbf{w}_{k,t,r}^l - \mathbf{w}_{k,r}\|_2 \leq \eta\tau\sqrt{\frac{2nK}{m\delta}}\mathbb{E}_{[\mathbf{M}_{k-1}],\mathbf{W}_0,\mathbf{a}}\left[\|\mathbf{y} - \mathbf{u}_k\|_2\right] + 2\eta\tau\kappa n\sqrt{\frac{2\xi(1-\xi)pK}{m\delta}}$$

for all $l \in [p]$ and $t \in [\tau]$. Using the definition that $\eta_{k,r} = \frac{N_{k,r}^\perp}{N_{k,r}}$, we have

$$\|\mathbf{w}_{k+1,r} - \mathbf{w}_{k,r}\|_2 \leq \eta_{k,r}\sum_{l=1}^p m_{k,r}^l\|\mathbf{w}_{k,\tau,r}^l - \mathbf{w}_{k,r}\|_2$$

$$\leq \eta\tau\sqrt{\frac{2nK}{m\delta}}\mathbb{E}_{[\mathbf{M}_{k-1}],\mathbf{W}_0,\mathbf{a}}\left[\|\mathbf{y} - \mathbf{u}_k\|_2\right] + 2\eta\tau\kappa n\sqrt{\frac{2\xi(1-\xi)pK}{m\delta}}$$

**(16), (15)→(14)** Assume that equation (16) and (15) holds for iteration $k$. We want to show (14) for up to iteration $k + 1$. Under these conditions, we have that with probability at least $1 - 4\delta$, Hypothesis 1 holds. Throughout this proof, we assume that

$$\sum_{i=1}^n \sum_{r'=1}^m \langle \mathbf{w}_{0,r}, \mathbf{x}_i \rangle^2 \leq 2mn\kappa^2 - mnR^2$$

and that

$$\|W_0\|_F \leq \sqrt{2md} - \sqrt{m}R$$

Note that Lemma 22 and Lemma 23 shows that, as long as $m = \Omega\left(\log\frac{n}{\delta}\right)$, the above assumption holds with probability at least $1 - \delta$ over initialization. Moreover, Lemma 16 shows that as long as $m = \left(\frac{n\log\frac{n}{\delta}}{\xi\lambda_0}\right)$, with probability at least $1 - \delta$ over initialization we have

$$|S_i^\perp| \leq 4m\kappa^{-1}R$$

To start, expanding the loss at iteration $k + 1$ gives

$$\mathbb{E}_{\mathbf{M}_k}\left[\|\mathbf{y} - \mathbf{u}_{k+1}\|_2^2\right] = \|\mathbf{y} - \mathbf{u}_k\|_2^2 - 2\langle\mathbf{y} - \mathbf{u}_k, \mathbb{E}_{\mathbf{M}_k}\left[\mathbf{u}_{k+1} - \mathbf{u}_k\right]\rangle + \mathbb{E}_{\mathbf{M}_k}\left[\|\mathbf{u}_{k+1} - \mathbf{u}_k\|_2^2\right]$$
$$= \|\mathbf{y} - \mathbf{u}_k\|_2^2 - 2\langle\mathbf{y} - \mathbf{u}_k, \mathbb{E}_{\mathbf{M}_k}\left[\mathbf{I}_{1,k}\right]\rangle - 2\langle\mathbf{y} - \mathbf{u}_k, \mathbb{E}_{\mathbf{M}_k}\left[\mathbf{I}_{2,k}\right]\rangle +$$
$$\mathbb{E}_{\mathbf{M}_k}\left[\|\mathbf{u}_{k+1} - \mathbf{u}_k\|_2^2\right]$$

Following previous work, we bound the second, third, and fourth term separately. However, the second term requires a more detailed analysis. In particular, we let

$$\mathbf{I}_{1,k}' = \mathbf{I}_{1,k} - \eta\theta\tau\mathbf{H}(k)(\mathbf{y} - \mathbf{u}_k)$$

Then the loss at iteration $k + 1$ has the form

$$\mathbb{E}_{\mathbf{M}_k}\left[\|\mathbf{y} - \mathbf{u}_{k+1}\|_2^2\right] = \|\mathbf{y} - \mathbf{u}_k\|_2^2 - 2\eta\theta\tau\langle\mathbf{y} - \mathbf{u}_k, \mathbf{H}(k)(\mathbf{y} - \mathbf{u}_k)\rangle + \mathbb{E}_{\mathbf{M}_k}\left[\|\mathbf{u}_{k+1} - \mathbf{u}_k\|_2^2\right] -$$
$$2\langle\mathbf{y} - \mathbf{u}_k, \mathbb{E}_{\mathbf{M}_k}\left[\mathbf{I}_{1,k}'\right]\rangle - 2\langle\mathbf{y} - \mathbf{u}_k, \mathbb{E}_{\mathbf{M}_k}\left[\mathbf{I}_{2,k}\right]\rangle$$
$$\leq (1 - \eta\theta\tau\lambda_0)\|\mathbf{y} - \mathbf{u}_k\|_2^2 + 2\left|\langle\mathbf{y} - \mathbf{u}_k, \mathbb{E}_{\mathbf{M}_k}\left[\mathbf{I}_{1,k}'\right]\rangle\right| +$$
$$2\left|\langle\mathbf{y} - \mathbf{u}_k, \mathbb{E}_{\mathbf{M}_k}\left[\mathbf{I}_{2,k}\right]\rangle\right| + \mathbb{E}_{\mathbf{M}_k}\left[\|\mathbf{u}_{k+1} - \mathbf{u}_k\|_2^2\right]$$

where in the last inequality we use $\lambda_{\min}(\mathbf{H}(k)) \geq \frac{\lambda_0}{2}$ from *(Du et al., 2018)*, Assumption 3.1. Moreover, Lemma 6, Lemma 9, and Lemma 10 shows that under the given assumption, with $\eta = O\left(\frac{\lambda_0}{n\tau\max\{n,p\}}\right)$ we have

$$\left|\langle\mathbf{y} - \mathbf{u}_k, \mathbb{E}_{\mathbf{M}_k}\left[\mathbf{I}_{1,k}'\right]\rangle\right| \leq \frac{1}{8}\eta\theta\tau\lambda_0\|\mathbf{y} - \mathbf{u}_k\|_2^2 + \frac{16\eta\theta\tau\xi^2(1-\xi)^2\kappa^2n^3d}{m\lambda_0} + \frac{2\eta^3\xi^2\tau(\tau-1)^2n^4pC_1}{\theta\lambda_0}$$

$$|\langle \mathbf{y} - \mathbf{u}_k, \mathbb{E}_{\mathbf{M}_k}[\mathbf{I}_{2,k}]\rangle| \leq \frac{1}{8}\eta\theta\tau\lambda_0\|\mathbf{y} - \mathbf{u}_k\|_2^2 + \frac{\eta\lambda_0\xi^2(\theta - \xi^2)n\kappa^2}{24p\tau} + \frac{\eta\lambda_0\xi^2(\tau - 1)^2pC_1}{96\tau\theta}$$

$$\mathbb{E}_{\mathbf{M}_k}\left[\|\mathbf{u}_{k+1} - \mathbf{u}_k\|_2^2\right] \leq \frac{1}{4}\eta\theta\tau\lambda_0\|\mathbf{y} - \mathbf{u}_k\|_2^2 + \frac{17\eta^2\xi^2\tau^2\theta(\theta - \xi^2)n^3\kappa^2}{p} + \eta^2\xi^2\lambda_0(\tau - 1)^2pnC_1$$

Putting things together gives

$$\begin{aligned}
\mathbb{E}_{\mathbf{M}_k}\left[\|\mathbf{y} - \mathbf{u}_{k+1}\|_2^2\right] &\leq \left(1 - \frac{1}{4}\eta\theta\tau\lambda_0\right)\|\mathbf{y} - \mathbf{u}_k\|_2^2 + \frac{16\eta\theta\tau\xi^2(1 - \xi)^2n^3d}{m\lambda_0} + \frac{2\eta^3\xi^2\tau(\tau - 1)^2n^4pC_1}{\theta\lambda_0} \\
&\quad \frac{\eta\lambda_0\xi^2(\theta - \xi^2)n\kappa^2}{24\tau p} + \frac{\eta\lambda_0\xi^2(\tau - 1)^2pC_1}{96\tau\theta} + \frac{17\eta^2\xi^2\tau^2\theta(\theta - \xi^2)n^3\kappa^2}{p} + \\
&\quad \eta^2\xi^2\lambda_0\tau(\tau - 1)pnC_1 \\
&\leq \left(1 - \frac{1}{4}\eta\theta\tau\lambda_0\right)\|\mathbf{y} - \mathbf{u}_k\|_2^2 + \frac{1}{4}\eta\theta\tau\lambda_0 B_1
\end{aligned}$$

Therefore, we have

$$\mathbb{E}_{[\mathbf{M}_{k-1}]}\left[\|\mathbf{y} - \mathbf{u}_k\|_2^2\right] \leq \left(1 - \frac{1}{4}\eta\theta\tau\lambda_0\right)^k\|\mathbf{y} - \mathbf{u}_0\|_2^2 + B_1$$

This completes the proof.

# F    Proof of Theorem 4

Again we use induction to prove the theorem. Consider the following conditions

$$\mathbb{E}_{\mathbf{M}_k}\left[\|\mathbf{y}-\mathbf{u}_k\|_2^2\right] \leq (1-\alpha)\|\mathbf{y}-\mathbf{u}_k\|_2^2 + B_1 \tag{17}$$

$$\|\mathbf{w}_{k,r}-\mathbf{w}_{0,r}\|_2 + 2\eta\tau\sqrt{\frac{2nK}{m\delta}}\left(\frac{1}{\alpha}\mathbb{E}_{[\mathbf{M}_{k-1}],\mathbf{W}_0,\mathbf{a}}\left[\|\mathbf{y}-\mathbf{u}_k\|_2\right] + (K-k)B\right) \leq R \tag{18}$$

with $\alpha$ and $B$ defined as below

$$\alpha = \left(1-\left(1-p^{-1}\right)^{\frac{1}{3}}\right)\left(1-\left(1-\frac{\eta\lambda_0}{2}\right)^\tau\right); \quad B = \frac{B_1}{\alpha} + \kappa\sqrt{\xi(1-\xi)pn}; \quad B_1 = O\left(\left(1-p^{-1}\right)^{\frac{2}{3}}n\kappa^2\eta\tau\frac{\lambda_{\max}}{\lambda_0}\right)$$

with $\xi = p^{-1}$. For notation clarity we will not plug in the value of $\xi$ for now. Notice that equation (17) implies the convergence result as in the theorem statement as long as $\gamma = \frac{1}{2}$ and that $\max\{K,d,p\} \geq n$. Thus, as long as we establish the inductive relations as above we are done.

**Base Case:** Notice by lemma 26, equation (17) holds. We use the over-parameterization requirement to show equation (18). In particular, we show

$$2\eta\tau\sqrt{\frac{2nK}{m\delta}}\left(\frac{1}{\alpha}\mathbb{E}_{\mathbf{W}_0,\mathbf{a}}\left[\|\mathbf{y}-\mathbf{u}_0\|_2\right] + KB\right) \leq R = O\left(\frac{\kappa\lambda_0}{n}\right)$$

As before, using Lemma 26, we have

$$\mathbb{E}_{\mathbf{W}_0,\mathbf{a}}\left[\|\mathbf{y}-\mathbf{u}_0\|_2\right] \leq \left(\mathbb{E}_{\mathbf{W}_0,\mathbf{a}}\left[\|\mathbf{y}-\mathbf{u}_0\|_2^2\right]\right)^{\frac{1}{2}} = C\sqrt{n}$$

Plugging in this value and the value of $\kappa, B$ and $\alpha$, we arrive at the over-parameterization requirement

$$m = \Omega\left(\frac{n^5\tau^2K\lambda_{\max}}{\lambda_0^6\delta}\right)$$

**Inductive Case:** the proof is again divided into two parts.

**(17)→(18)** Assume that equation (17) holds, and equation (18) holds for global iteration $k$. Similar to the proof of Theorem 3, it suffice to show that

$$\|\mathbf{w}_{k+1,r}-\mathbf{w}_{0,r}\|_2 \leq \eta\tau\sqrt{\frac{2nK}{m\delta}}\mathbb{E}_{[\mathbf{M}_{k-1}],\mathbf{W}_0,\mathbf{a}}\left[\|\mathbf{y}-\mathbf{u}_k\|_2\right] + 2\eta\tau\sqrt{\frac{2nK}{m\delta}}B - 2\eta\tau\sqrt{\frac{2nKB_1}{m\delta\alpha}}$$

Plugging in the value of $B$ and $\alpha$, it then suffice to show that

$$\|\mathbf{w}_{k+1,r}-\mathbf{w}_{0,r}\|_2 \leq \eta\tau\sqrt{\frac{2nK}{m\delta}}\mathbb{E}_{[\mathbf{M}_{k-1}],\mathbf{W}_0,\mathbf{a}}\left[\|\mathbf{y}-\mathbf{u}_k\|_2\right] + 2\eta\tau\kappa n\sqrt{\frac{2\xi(1-\xi)pK}{m\delta}}$$

By Hypothesis 1 we have

$$\|\mathbf{w}_{k,t,r}^l-\mathbf{w}_{k,r}\|_2 \leq \eta\tau\sqrt{\frac{2nK}{m\delta}}\mathbb{E}_{[\mathbf{M}_{k-1}],\mathbf{W}_0,\mathbf{a}}\left[\|\mathbf{y}-\mathbf{u}_k\|_2\right] + 2\eta\tau\kappa n\sqrt{\frac{2\xi(1-\xi)pK}{m\delta}}$$

for all $l \in [p]$ and $t \in [\tau]$. Then we have

$$\|\mathbf{w}_{k+1,r}-\mathbf{w}_{k,r}\|_2 \leq \eta_{k,r}\sum_{l=1}^p m_{k,r}^l\|\mathbf{w}_{k,\tau,r}^l-\mathbf{w}_{k,r}\|_2$$

$$\leq \eta\tau\sqrt{\frac{2nK}{m\delta}}\mathbb{E}_{[\mathbf{M}_{k-1}],\mathbf{W}_0,\mathbf{a}}\left[\|\mathbf{y}-\mathbf{u}_k\|_2\right] + 2\eta\tau\kappa n\sqrt{\frac{\xi(1-\xi)pK}{m\delta}}$$

**(18)→(17)** Now, assume that equation (18) holds. Then the result of Hypothesis (D) holds. Our target is to show equation (17). As in previous theorem, we start by studying $\|\mathbf{y} - \mathbf{u}_{k+1}\|_2^2$. In the case of a categorical mask, we have the nice property that the average of the sub-networks equals to the full network $\mathbf{u}_{k+1} = \frac{1}{p} \sum_{l=1}^{p} \hat{\mathbf{u}}_{k,\tau}^l$. Using this property, Lemma 13 characterize $\|\mathbf{y} - \mathbf{u}_{k+1}\|_2^2$ as

$$\|\mathbf{y} - \mathbf{u}_{k+1}\|_2^2 = \frac{1}{p} \sum_{l=1}^{p} \|\mathbf{y} - \hat{\mathbf{u}}_{k,\tau}^l\|_2^2 - \frac{1}{p^2} \sum_{l=1}^{p} \sum_{l'=1}^{l-1} \|\hat{\mathbf{u}}_{k,\tau}^l - \hat{\mathbf{u}}_{k,\tau}^{l'}\|_2^2$$

We start by assuming the condition of Hypothesis 1 holds. We proceed by proving the convergence, then we prove the weight perturbation bound with a fashion of induction. Hypothesis 1 implies that

$$\|\mathbf{y} - \hat{\mathbf{u}}_{k,\tau}^l\|_2^2 \leq \left(1 - \frac{\eta\lambda_0}{2}\right)^{\tau} \|\mathbf{y} - \hat{\mathbf{u}}_k^l\|_2^2$$

$$= \|\mathbf{y} - \hat{\mathbf{u}}_k^l\|_2^2 - \frac{\eta\lambda_0}{2} \sum_{t=0}^{\tau-1} \left(1 - \frac{\eta\lambda_0}{2}\right)^{t} \|\mathbf{y} - \hat{\mathbf{u}}_k^l\|_2^2$$

Using the fact that $\mathbb{E}_{\mathbf{M}_k}\left[\hat{\mathbf{u}}_k^l\right] = \mathbf{u}_k$, we have that

$$\mathbb{E}_{\mathbf{M}_k}\left[\|\mathbf{y} - \hat{\mathbf{u}}_k\|_2^2\right] = \|\mathbf{y} - \mathbf{u}_k\|_2^2 + \mathbb{E}_{\mathbf{M}_k}\left[\|\mathbf{u}_k - \hat{\mathbf{u}}_k^l\|_2^2\right]$$

Therefore, we have

$$\mathbb{E}_{\mathbf{M}_k}\left[\|\mathbf{y} - \mathbf{u}_{k+1}\|_2^2\right] = \frac{1}{p} \sum_{l=1}^{p} \mathbb{E}_{\mathbf{M}_k}\left[\|\mathbf{y} - \hat{\mathbf{u}}_k^l\|_2^2\right] - \frac{\eta\lambda_0}{2p} \sum_{t=0}^{\tau-1} \sum_{l=1}^{p} \left(1 - \frac{\eta\lambda_0}{2}\right)^{t} \mathbb{E}_{\mathbf{M}_k}\left[\|\mathbf{y} - \hat{\mathbf{u}}_k^l\|_2^2\right] -$$

$$\frac{1}{p^2} \sum_{l=1}^{p} \sum_{l'=1}^{l} \mathbb{E}_{\mathbf{M}_k}\left[\|\hat{\mathbf{u}}_{k,\tau}^l - \hat{\mathbf{u}}_{k,\tau}^{l'}\|_2^2\right]$$

$$= \|\mathbf{y} - \mathbf{u}_k\|_2^2 - \frac{\eta\lambda_0}{2p} \sum_{t=0}^{\tau-1} \sum_{l=1}^{p} \left(1 - \frac{\eta\lambda_0}{2}\right)^{t} \mathbb{E}_{\mathbf{M}_k}\left[\|\mathbf{y} - \hat{\mathbf{u}}_k^l\|_2^2\right] +$$

$$\frac{1}{p} \sum_{l=1}^{p} \mathbb{E}_{\mathbf{M}_k}\left[\|\mathbf{u}_k - \hat{\mathbf{u}}_k^l\|_2^2\right] - \frac{1}{p^2} \sum_{l=1}^{p} \sum_{l'=1}^{l} \mathbb{E}_{\mathbf{M}_k}\left[\|\hat{\mathbf{u}}_{k,\tau}^l - \hat{\mathbf{u}}_{k,\tau}^{l'}\|_2^2\right]$$

Lemma 14 studies the error term $\|\mathbf{u}_k - \hat{\mathbf{u}}_k^l\|_2^2$ and gives

$$\sum_{l=1}^{p} \|\mathbf{u}_k - \hat{\mathbf{u}}_k^l\|_2^2 = \frac{1}{p} \sum_{l=1}^{p} \sum_{l'=1}^{l-1} \|\hat{\mathbf{u}}_k^l - \hat{\mathbf{u}}_k^{l'}\|_2^2$$

Plugging in we have

$$\mathbb{E}_{\mathbf{M}_k}\left[\|\mathbf{y} - \mathbf{u}_{k+1}\|_2^2\right] \leq \|\mathbf{y} - \mathbf{u}_k\|_2^2 - -\frac{\eta\lambda_0}{2p} \sum_{t=0}^{\tau-1} \left(1 - \frac{\eta\lambda_0}{2}\right)^{t} \sum_{l=1}^{p} \mathbb{E}_{\mathbf{M}_k}\left[\|\mathbf{y} - \hat{\mathbf{u}}_k^l\|_2^2\right] +$$

$$\frac{1}{p^2} \sum_{l=1}^{p} \sum_{l'=1}^{l} \mathbb{E}_{\mathbf{M}_k}\left[\|\hat{\mathbf{u}}_k^l - \hat{\mathbf{u}}_k^{l'}\|_2^2 - \|\hat{\mathbf{u}}_{k,\tau}^l - \hat{\mathbf{u}}_{k,\tau}^{l'}\|_2^2\right]$$

Denote the last term in the right hand side as $\iota_k$. Lemma 15 shows bound of the expectation of $\iota_k$ with respect to the initialization. In particular, if lemma 23 holds for some $R \geq 0$, and the weight perturbation

is bounded by $\|\mathbf{w}_{k,r}^l - \mathbf{w}_{0,r}\|_2 \leq R$ for all $r \in [m]$, then we have

$$\iota_k \leq \frac{\left(1 - p^{-1}\right)^{\frac{1}{2}} \eta\lambda_0}{2p} \sum_{l=1}^{p} \sum_{t=0}^{\tau-1} \mathbb{E}_{\mathbf{M}_k} \left[\left\|\mathbf{y} - \hat{\mathbf{u}}_{k,t}^l\right\|_2^2\right] + 24 \left(1 - p^{-1}\right)^{\frac{1}{2}} n\kappa^2 \eta\tau \frac{\lambda_{\max}}{\lambda_0}$$

Using this result, we have that

$$\begin{aligned}
\mathbb{E}_{\mathbf{M}_k} \left[\|\mathbf{y} - \mathbf{u}_{k+1}\|_2^2\right] &= \|\mathbf{y} - \mathbf{u}_k\|_2^2 - -\frac{\eta\lambda_0}{2p} \sum_{t=0}^{\tau-1} \left(1 - \frac{\eta\lambda_0}{2}\right)^t \sum_{l=1}^{p} \mathbb{E}_{\mathbf{M}_k} \left[\|\mathbf{y} - \hat{\mathbf{u}}_k^l\|_2^2\right] + \iota_k \\
&= \|\mathbf{y} - \mathbf{u}_k\|_2^2 - \frac{\eta\lambda_0}{2p} \sum_{l=1}^{p} \sum_{t=0}^{\tau-1} \left(1 - \frac{\eta\lambda_0}{2}\right)^t \mathbb{E}_{\mathbf{M}_k} \left[\|\mathbf{y} - \hat{\mathbf{u}}_k^l\|_2^2\right] + \\
&\qquad \frac{\left(1 - p^{-1}\right)^{\frac{1}{3}} \eta\lambda_0}{4p} \sum_{l=1}^{p} \sum_{t=0}^{\tau-1} \left(1 - \frac{\eta\lambda_0}{2}\right)^t \mathbb{E}_{\mathbf{M}_k} \left[\|\mathbf{y} - \hat{\mathbf{u}}_k^l\|_2^2\right] + 48 \left(1 - p^{-1}\right)^{\frac{2}{3}} n\kappa^2 \eta\tau \frac{\lambda_{\max}}{\lambda_0} \\
&= \|\mathbf{y} - \mathbf{u}_k\|_2^2 - \frac{\left(1 - p^{-1}\right)^{\frac{1}{3}} \eta\lambda_0}{4p} \sum_{l=1}^{p} \sum_{t=0}^{\tau-1} \left(1 - \frac{\eta\lambda_0}{2}\right)^t \mathbb{E}_{\mathbf{M}_k} \left[\|\mathbf{y} - \hat{\mathbf{u}}_k^l\|_2^2\right] + \\
&\qquad 48 \left(1 - p^{-1}\right)^{\frac{2}{3}} n\kappa^2 \eta\tau \frac{\lambda_{\max}}{\lambda_0} \\
&\leq \|\mathbf{y} - \mathbf{u}_k\|_2^2 - \frac{\left(1 - p^{-1}\right)^{\frac{1}{3}} \eta\lambda_0}{4} \sum_{t=0}^{\tau-1} \left(1 - \frac{\eta\lambda_0}{2}\right)^t \|\mathbf{y} - \mathbf{u}_k\|_2^2 + \\
&\qquad 48 \left(1 - p^{-1}\right)^{\frac{2}{3}} n\kappa^2 \eta\tau \frac{\lambda_{\max}}{\lambda_0} \\
&= \left(\left(1 - p^{-1}\right)^{\frac{1}{3}} + \left(1 - \left(1 - p^{-1}\right)^{\frac{1}{3}}\right) \left(1 - \frac{\eta\lambda_0}{2}\right)^{\tau}\right) \|\mathbf{y} - \mathbf{u}_k\|_2^2 + 48 \left(1 - p^{-1}\right)^{\frac{2}{3}} n\kappa^2 \eta\tau \frac{\lambda_{\max}}{\lambda_0}
\end{aligned}$$

Therefore,

$$\mathbb{E}_{\mathbf{M}_k} \left[\|\mathbf{y} - \mathbf{u}_{k+1}\|_2^2\right] \leq (1 - \alpha) \|\mathbf{y} - \mathbf{u}_k\|_2^2 + B_1$$

This is the same as the form in equation (17). Thus we arrive at the convergence

$$\mathbb{E}_{[\mathbf{M}_k]} \left[\|\mathbf{y} - \mathbf{u}_k\|_2^2\right] \leq (1 - \alpha)^k \|\mathbf{y} - \mathbf{u}_0\|_2^2 + \frac{B_1}{\alpha}$$

This shows the convergence result in the theorem with $\alpha$ and $B_1$ plugged in.

## G   Lemmas for Theorem 3

**Lemma 1.** *The expectation of the mixing function satisfies*

$$\mathbb{E}_{\mathbf{M}_k}\left[f_{k,r}^{(i)}\right] = \theta u_k^{(i)} + \frac{\theta(1-\xi)}{\sqrt{m}}a_r\sigma(\langle\mathbf{w}_{k,r'},\mathbf{x}_i\rangle)$$

*Proof.* Note that if $N_{k,r}^{\perp} = 0$, then we have $f_{k,r}^{(i)} = 0$ for all $i \in [n]$. Thus

$$\mathbb{E}_{\mathbf{M}_k}\left[f_{k,r}^{(i)} \mid N_{k,r}^{\perp} = 0\right] = 0$$

Moreover, if $N_{k,r}^{\perp} = 1$, the expectation can be computed as

$$\mathbb{E}_{\mathbf{M}_k}\left[f_{k,r}^{(i)} \mid N_{k,r}^{\perp} = 1\right] = \mathbb{E}_{\mathbf{M}_k}\left[\frac{\eta_{k,r}}{\sqrt{m}}\sum_{l=1}^{p}\sum_{r'=1}^{m}m_{k,r}^{l}m_{k,r'}^{l}a_r\sigma(\langle\mathbf{w}_{k,r},\mathbf{x}_i\rangle) \mid N_{k,r}^{\perp} = 1\right]$$

$$= \frac{1}{\sqrt{m}}\sum_{r'=1}^{m}\mathbb{E}_{\mathbf{M}_k}\left[\eta_{k,r}\sum_{l=1}^{p}m_{k,r}^{l}m_{k,r'}^{l} \mid N_{k,r}^{\perp} = 1\right]a_{r'}\sigma(\langle\mathbf{w}_{k,r'},\mathbf{x}_i\rangle)$$

$$= \frac{\xi}{\sqrt{m}}\sum_{r'=1}^{m}a_{r'}\sigma(\langle\mathbf{w}_{k,r'},\mathbf{x}_i\rangle) + \frac{1-\xi}{\sqrt{m}}a_r\sigma(\langle\mathbf{w}_{k,r'},\mathbf{x}_i\rangle)$$

by using Lemma 20. Combining the two conditions above gives that

$$\mathbb{E}_{\mathbf{M}_k}\left[f_{k,r}^{(i)}\right] = P(N_{k,r}^{\perp} = 1)\mathbb{E}_{\mathbf{M}_k}\left[f_{k,r}^{(i)} \mid N_{k,r}^{\perp} = 1\right] + P(N_{k,r}^{\perp} = 0)\mathbb{E}_{\mathbf{M}_k}\left[f_{k,r}^{(i)} \mid N_{k,r}^{\perp} = 0\right]$$

$$= \frac{\theta\xi}{\sqrt{m}}\sum_{r'=1}^{m}a_{r'}\sigma(\langle\mathbf{w}_{k,r'},\mathbf{x}_i\rangle) + \frac{\theta(1-\xi)}{\sqrt{m}}a_r\sigma(\langle\mathbf{w}_{k,r'},\mathbf{x}_i\rangle)$$

$$= \theta u_k^{(i)} + \frac{\theta(1-\xi)}{\sqrt{m}}a_r\sigma(\langle\mathbf{w}_{k,r'},\mathbf{x}_i\rangle)$$

$\square$

**Lemma 2.** *The expectation of the mixing gradient satisfies*

$$\mathbb{E}_{\mathbf{M}_k}[\mathbf{g}_{k,r}] = \frac{\theta}{\xi}\frac{\partial L(\mathbf{W}_k)}{\partial\mathbf{w}_r} + \frac{\theta(1-\xi)}{m}\sum_{i=1}^{n}\mathbf{x}_i\sigma(\langle\mathbf{w}_{k,r},\mathbf{x}_i\rangle)$$

*Proof.* With the result from Lemma 1, we have

$$\mathbb{E}_{\mathbf{M}_k}[\mathbf{g}_{k,r}] = \frac{1}{\sqrt{m}}\sum_{i=1}^{n}\left(\mathbb{E}_{\mathbf{M}_k}\left[f_{k,r}^{(i)}\right] - y_i\mathbb{E}_{\mathbf{M}_k}\left[N_{k,r}^{\perp}\right]\right)a_r\mathbf{x}_i\mathbb{I}\{\langle\mathbf{w}_{k,r},\mathbf{x}_i\rangle \geq 0\}$$

$$= \frac{\theta}{\sqrt{m}}\sum_{i=1}^{n}\left(u_k^{(i)} - y_i + \frac{1-\xi}{\sqrt{m}}a_r\sigma(\langle\mathbf{w}_{k,r},\mathbf{x}_i\rangle)\right)a_r\mathbf{x}_i\mathbb{I}\{\langle\mathbf{w}_{k,r},\mathbf{x}_i\rangle \geq 0\}$$

$$= \frac{\theta}{\xi}\frac{\partial L(\mathbf{W}_k)}{\partial\mathbf{w}_r} + \frac{\theta(1-\xi)}{m}\sum_{i=1}^{n}\mathbf{x}_i\sigma(\langle\mathbf{w}_{k,r},\mathbf{x}_i\rangle)$$

$\square$

**Lemma 3.** *Suppose $m \geq p$. If for some $R > 0$ and all $r \in [m]$ the initialization satisfies*

$$\sum_{r=1}^{m}\langle\mathbf{w}_{0,r},\mathbf{x}_i\rangle^2 \leq 2mn\kappa^2 - mnR^2$$

and for all $r \in [m]$, it holds that $\|\mathbf{w}_{k,r} - \mathbf{w}_{0,r}\|_2 \leq R$, the expected norm of the difference between the mixing function and $u_k^{(i)}$ satisfies

$$\mathbb{E}_{\mathbf{M}_k}\left[\|\mathbf{f}_{k,r} - \mathbf{u}_k\|_2^2 \mid N_{k,r}^\perp = 1\right] \leq \frac{8(\theta - \xi^2)n\kappa^2}{p}$$

*Proof.* Since $\mathbb{E}_{\mathbf{M}_k}\left[\nu_{k,r,r'} \mid N_{k,r}^\perp = 1\right] = \xi$ for $r' \neq r$, we have for $r_1 \neq r_2$, there is at least one of $r_1, r_2$ that is not $r$. Thus

$$\mathbb{E}_{\mathbf{M}_k}\left[(\nu_{k,r,r_1} - \xi)(\nu_{k,r,r_2} - \xi) \mid N_{k,r}^\perp = 1\right] = 0$$

and for $r \neq r'$

$$\mathrm{Var}_{\mathbf{M}_k}\left(\nu_{k,r,r'} \mid N_{k,r}^\perp = 1\right) = \mathbb{E}_{\mathbf{M}_k}\left[(\nu_{k,r,r'} - \xi)^2 \mid N_{k,r}^\perp = 1\right]$$

Moreover, for $r = r'$, Lemma 19

$$\mathbb{E}_{\mathbf{M}_k}\left[(\nu_{k,r,r,} - \xi)^2\right] \leq \theta - \xi^2$$

Therefore, using Lemma 21 we have

$$\mathbb{E}_{\mathbf{M}_k}\left[\left(f_{k,r}^{(i)} - u_k^{(i)}\right)^2 \mid N_{k,r}^\perp = 1\right] = \frac{1}{m}\mathbb{E}_{\mathbf{M}_k}\left[\left(\sum_{r' \neq r}^m a_r(\nu_{k,r,r'} - \xi)\sigma(\langle \mathbf{w}_{k,r'}, \mathbf{x}_i\rangle)\right)^2 \mid N_{k,r}^\perp = 1\right]$$

$$= \frac{1}{m}\sum_{r'=1}^m \mathrm{Var}_{\mathbf{M}_k}\left(\nu_{k,r,r'} \mid N_{k,r}^\perp = 1\right)\sigma(\langle \mathbf{w}_{k,r'}, \mathbf{x}_i\rangle)^2 +$$

$$\frac{1}{m}\mathbb{E}_{\mathbf{M}_k}\left[(\nu_{k,r,r,} - \xi)^2\right]\sigma(\langle \mathbf{w}_{k,r}, \mathbf{x}_i\rangle)^2$$

$$\leq \frac{\theta - \xi^2}{pm}\sum_{r' \neq r}^m \langle \mathbf{w}_{k,r'}, \mathbf{x}_i\rangle^2 + \frac{\theta - \xi^2}{m}\sigma(\langle \mathbf{w}_{k,r}, \mathbf{x}_i\rangle)^2$$

$$\leq \frac{2(\theta - \xi^2)}{pm}\left(\sum_{r'=1}^m \langle \mathbf{w}_{0,r}, \mathbf{x}_i\rangle^2 + mR^2\right) + \frac{2(\theta - \xi^2)}{p}\left(\langle \mathbf{w}_{k,r}, \mathbf{x}_i\rangle + R^2\right)$$

$$\leq \frac{8(\theta - \xi^2)\kappa^2}{p}$$

Plugging this in gives

$$\mathbb{E}_{\mathbf{M}_k}\left[\|\mathbf{f}_{k,r} - \mathbf{u}_k\|_2^2 \mid N_{k,r}^\perp = 1\right] \leq \frac{8(\theta - \xi^2)n\kappa^2}{p}$$

$\square$

**Lemma 4.** *Under the condition of Lemma 3, the expected norm and squared-norm of the mixing gradient is bounded by*

$$\mathbb{E}_{\mathbf{M}_k}\left[\|\mathbf{g}_{k,r}\|_2^2\right] \leq \frac{2n\theta}{m}\|\mathbf{y} - \mathbf{u}_k\|_2^2 + \frac{16\theta(\theta - \xi^2)n^2\kappa^2}{pm}$$

$$\mathbb{E}_{\mathbf{M}_k}\left[\|\mathbf{g}_{k,r}\|_2\right] \leq \frac{\sqrt{n}\theta}{\sqrt{m}}\|\mathbf{y} - \mathbf{u}_k\|_2 + 4n\kappa\sqrt{\frac{\theta(\theta - \xi^2)}{pm}}$$

*Proof.* Using Lemma 3, we have

$$\mathbb{E}_{\mathbf{M}_k}\left[N_{k,r}^\perp \|\mathbf{f}_{k,r} - \mathbf{u}_k\|_2^2\right] = P(N_{k,r}^\perp = 1)\mathbb{E}_{\mathbf{M}_k}\left[\|\mathbf{f}_{k,r} - \mathbf{u}_k\|_2^2\right] \leq \frac{8\theta(\theta - \xi^2)n\kappa^2}{p}$$

According to Jensen's inequality, we also have

$$\mathbb{E}_{\mathbf{M}_k}\left[N_{k,r}^{\perp}\|\mathbf{f}_{k,r} - \mathbf{u}_k\|_2\right] \le 2\kappa \cdot \sqrt{\frac{2\theta(\theta - \xi^2)n}{p}}$$

Moreover, we have

$$\mathbf{g}_{k,r} = \frac{1}{\sqrt{m}} \sum_{i=1}^{n} \left(f_{k,r}^{(i)} - y_i\right) a_r N_{k,r}^{\perp} \mathbf{x}_i \mathbb{I}\{\langle \mathbf{w}_{k,r}, \mathbf{x}_i \rangle \ge 0\}$$

$$= \frac{1}{\sqrt{m}} \sum_{i=1}^{n} \left(f_{k,r}^{(i)} - u_k^{(i)}\right) a_r N_{k,r}^{\perp} \mathbf{x}_i \mathbb{I}\{\langle \mathbf{w}_{k,r}, \mathbf{x}_i \rangle \ge 0\} + \frac{N_{k,r}^{\perp}}{\xi} \cdot \frac{\partial L(\mathbf{W}_k)}{\partial \mathbf{w}_r}$$

Therefore,

$$\mathbb{E}_{\mathbf{M}_k}\left[\|\mathbf{g}_{k,r}\|_2^2\right] \le \frac{2\mathbb{E}_{\mathbf{M}_k}\left[N_{k,r}^{\perp}\right]}{\xi^2}\left\|\frac{\partial L(\mathbf{W}_k)}{\partial \mathbf{w}_r}\right\|_2^2 + \frac{2}{m}\mathbb{E}_{\mathbf{M}_k}\left[\left\|\sum_{i=1}^{n}\left(f_{k,r}^{(i)} - u_k^{(i)}\right) a_r N_{k,r}^{\perp} \mathbf{x}_i \mathbb{I}\{\langle \mathbf{w}_{k,r}, \mathbf{x}_i \rangle \ge 0\}\right\|_2^2\right]$$

$$\le \frac{2n}{m}\mathbb{E}_{\mathbf{M}_k}\left[N_{k,r}^{\perp}\sum_{i=1}^{n}\left(f_{k,r}^{(i)} - u_k^{(i)}\right)^2\right] + \frac{2n\theta}{m}\|\mathbf{y} - \mathbf{u}_k\|_2^2$$

$$\le \frac{2n}{m}\left(\mathbb{E}_{\mathbf{M}_k}\left[N_{k,r}^{\perp}\|\mathbf{f}_{k,r} - \mathbf{u}_k\|_2^2\right] + \theta\|\mathbf{y} - \mathbf{u}_k\|_2^2\right)$$

$$\le \frac{2n\theta}{m}\|\mathbf{y} - \mathbf{u}_k\|_2^2 + \frac{16\theta(\theta - \xi^2)n^2\kappa^2}{pm}$$

This shows the first inequality. To show the second, similarly we have

$$\mathbb{E}_{\mathbf{M}_k}\left[\|\mathbf{g}_{k,r}\|_2\right] \le \frac{\mathbb{E}_{\mathbf{M}_k}\left[N_{k,r}^{\perp}\right]}{\xi}\left\|\frac{\partial L(\mathbf{W}_k)}{\partial \mathbf{w}_r}\right\|_2 + \frac{1}{\sqrt{m}}\mathbb{E}_{\mathbf{M}_k}\left[\left\|\sum_{i=1}^{n}\left(f_{k,r}^{(i)} - u_k^{(i)}\right) a_r N_{k,r}^{\perp} \mathbf{x}_i \mathbb{I}\{\langle \mathbf{w}_{k,r}, \mathbf{x}_i \rangle \ge 0\}\right\|_2\right]$$

$$\le \frac{1}{\sqrt{m}}\mathbb{E}_{\mathbf{M}_k}\left[N_{k,r}^{\perp}\sum_{i=1}^{n}\left|f_{k,r}^{(i)} - u_k^{(i)}\right|\right] + \frac{\sqrt{n}\theta}{\sqrt{m}}\|\mathbf{y} - \mathbf{u}_k\|_2$$

$$\le \sqrt{\frac{n}{m}}\mathbb{E}_{\mathbf{M}_k}\left[N_{k,r}^{\perp}\|\mathbf{f}_{k,r} - \mathbf{u}_k\|_2\right] + \frac{\sqrt{n}\theta}{\sqrt{m}}\|\mathbf{y} - \mathbf{u}_k\|_2$$

$$\le \frac{\sqrt{n}\theta}{\sqrt{m}}\|\mathbf{y} - \mathbf{u}_k\|_2 + 4n\kappa\sqrt{\frac{\theta(\theta - \xi^2)}{pm}}$$

$\square$

**Lemma 5.** *Under the condition of Theorem 3, we have*

$$\left|\hat{u}_{k,t}^{l(i)} - \hat{u}_k^{l(i)}\right| \le \eta t \sqrt{n}\|\mathbf{y} - \hat{\mathbf{u}}_k^l\|_2$$

*and therefore,*

$$\left|\hat{u}_{k,t}^{l(i)} - \hat{u}_k^{l(i)}\right| \le \eta t \sqrt{n}\left(\|\mathbf{y} - \mathbf{u}_k\|_2 + \|\mathbf{u}_k - \hat{\mathbf{u}}_k^l\|_2\right)$$

$$\left(\hat{u}_{k,t}^{l(i)} - \hat{u}_k^{l(i)}\right)^2 \le 2\eta^2 t^2 n\left(\|\mathbf{y} - \mathbf{u}_k\|_2^2 + \|\mathbf{u}_k - \hat{\mathbf{u}}_k^l\|_2^2\right)$$

*Proof.* We have

$$\left| \hat{u}_{k,t}^{l(i)} - \hat{u}_k^{l(i)} \right| = \frac{1}{\sqrt{m}} \left| \sum_{r=1}^{m} a_r m_{k,r}^l \left( \sigma \left( \langle \mathbf{w}_{k,t,r}^l, \mathbf{x}_i \rangle \right) - \sigma \left( \langle \mathbf{w}_{k,r}, \mathbf{x}_i \rangle \right) \right) \right|$$

$$\leq \frac{1}{\sqrt{m}} \sum_{r=1}^{m} \left| \sigma \left( \langle \mathbf{w}_{k,t,r}^l, \mathbf{x}_i \rangle \right) - \sigma \left( \langle \mathbf{w}_{k,r}, \mathbf{x}_i \rangle \right) \right|$$

$$\leq \frac{1}{\sqrt{m}} \sum_{r=1}^{m} \left\| \mathbf{w}_{k,t,r}^l - \mathbf{w}_{k,r} \right\|_2$$

$$\leq \frac{\eta}{\sqrt{m}} \sum_{r=1}^{m} \sum_{t'=0}^{t-1} \left\| \frac{\partial L_{\mathbf{m}_k^l} \left( \mathbf{W}_{k,t}^l \right)}{\partial \mathbf{w}_r} \right\|_2$$

$$\leq \eta \sqrt{n} \sum_{t'=0}^{t-1} \| \mathbf{y} - \hat{\mathbf{u}}_{k,t}^l \|_2$$

$$\leq \eta t \sqrt{n} \| \mathbf{y} - \hat{\mathbf{u}}_k^l \|_2$$

Therefore,

$$\left| \hat{u}_{k,t}^{l(i)} - \hat{u}_k^{l(i)} \right| \leq \eta t \sqrt{n} \left( \| \mathbf{y} - \mathbf{u}_k \|_2 + \| \mathbf{u}_k - \hat{\mathbf{u}}_k^l \|_2 \right)$$

Moreover,

$$\left( \hat{u}_{k,t}^{l(i)} - \hat{u}_k^{l(i)} \right)^2 = \left| \hat{u}_{k,t}^{l(i)} - \hat{u}_k^{l(i)} \right|^2 \leq 2\eta^2 t^2 n \left( \| \mathbf{y} - \mathbf{u}_k \|_2^2 + \| \mathbf{u}_k - \hat{\mathbf{u}}_k^l \|_2^2 \right)$$

$\square$

**Lemma 6.** *Under the condition of Lemma 3, with* $\eta \leq \frac{\lambda_0}{16(\tau-1)n^2}$, *we have*

$$\left| \langle \mathbf{y} - \mathbf{u}_k, \mathbb{E}_{\mathbf{M}_k} \left[ \mathbf{I}_{1,k}' \right] \rangle \right| \leq \frac{1}{8} \eta \theta \tau \lambda_0 \| \mathbf{y} - \mathbf{u}_k \|_2^2 + \frac{16\eta \theta \tau \xi^2 (1-\xi)^2 \kappa^2 n^3 d}{m \lambda_0} + \frac{2\eta^3 \xi^2 \tau (\tau-1)^2 n^4 p C_1}{\theta \lambda_0}$$

*Proof.* We start by analyzing $\mathbf{w}_{k+1,r} - \mathbf{w}_{k,r}$. Taking expectation, we have

$$\mathbb{E}_{\mathbf{M}_k} \left[ \mathbf{w}_{k+1,r} - \mathbf{w}_{k,r} \right] = -\eta \mathbb{E}_{\mathbf{M}_k} \left[ \eta_{k,r} \sum_{l=1}^{p} \sum_{t=0}^{\tau-1} \frac{\partial L_{\mathbf{m}_k^l} \left( \mathbf{W}_{k,t}^l \right)}{\partial \mathbf{w}_r} \right]$$

$$= -\eta \mathbb{E}_{\mathbf{M}_k} \left[ \eta_{k,r} \tau \sum_{l=1}^{p} \frac{\partial L_{\mathbf{m}_k^l} (\mathbf{W}_k)}{\partial \mathbf{w}_r} + \eta_{k,r} \sum_{l=1}^{p} \sum_{t=1}^{\tau-1} \left( \frac{\partial L_{\mathbf{m}_k^l} \left( \mathbf{W}_{k,t}^l \right)}{\partial \mathbf{w}_r} - \frac{\partial L_{\mathbf{m}_k^l} (\mathbf{W}_k)}{\partial \mathbf{w}_r} \right) \right]$$

$$= -\eta \tau \mathbb{E}_{\mathbf{M}_k} \left[ \mathbf{g}_{k,r} \right] - \eta \mathbb{E}_{\mathbf{M}_k} \left[ \eta_{k,r} \sum_{l=1}^{p} \sum_{t=1}^{\tau-1} \left( \frac{\partial L_{\mathbf{m}_k^l} \left( \mathbf{W}_{k,t}^l \right)}{\partial \mathbf{w}_r} - \frac{\partial L_{\mathbf{m}_k^l} (\mathbf{W}_k)}{\partial \mathbf{w}_r} \right) \right]$$

$$= -\frac{\eta \theta \tau}{\xi} \frac{\partial L(\mathbf{W}_k)}{\partial \mathbf{w}_r} - \frac{\eta \theta (1-\xi) \tau}{m} \sum_{i=1}^{n} \mathbf{x}_i \sigma(\langle \mathbf{w}_{k,r}, \mathbf{x}_i \rangle) -$$

$$\eta \mathbb{E}_{\mathbf{M}_k} \left[ \eta_{k,r} \sum_{l=1}^{p} \sum_{t=1}^{\tau-1} \left( \frac{\partial L_{\mathbf{m}_k^l} \left( \mathbf{W}_{k,t}^l \right)}{\partial \mathbf{w}_r} - \frac{\partial L_{\mathbf{m}_k^l} (\mathbf{W}_k)}{\partial \mathbf{w}_r} \right) \right]$$

Therefore

$$
\begin{aligned}
\mathbb{E}_{\mathbf{M}_k}\left[I_{1,k}^{(i)}\right] &= \frac{\xi}{\sqrt{m}} \sum_{r \in S_i} a_r \mathbb{E}_{\mathbf{M}_k}\left[\sigma(\langle \mathbf{w}_{k+1,r}, \mathbf{x}_i \rangle) - \sigma(\langle \mathbf{w}_{k,r}, \mathbf{x}_i \rangle)\right] \\
&= \frac{\xi}{\sqrt{m}} \sum_{r \in S_i} a_r \langle \mathbb{E}_{\mathbf{M}_k}\left[\mathbf{w}_{k+1,r} - \mathbf{w}_{k,r}\right], \mathbf{x}_i \rangle \mathbb{I}\{\langle \mathbf{w}_{k,r}, \mathbf{x}_i \rangle \geq 0\} \\
&= -\frac{\eta \theta \tau}{\sqrt{m}} \sum_{r \in S_i} a_r \left\langle \frac{\partial L(\mathbf{W}_k)}{\partial \mathbf{w}_r}, \mathbf{x}_i \right\rangle \mathbb{I}\{\langle \mathbf{w}_{k,r}, \mathbf{x}_i \rangle \geq 0\} - \eta \left(\mathcal{E}_{1,k}^{(i)} + \mathcal{E}_{2,k}^{(i)}\right) \\
&= \frac{\eta \xi \theta \tau}{m} \sum_{r \in S_i} \sum_{j=1}^{n} (y_i - u_k^{(i)}) \langle \mathbf{x}_i, \mathbf{x}_j \rangle \mathbb{I}\{\langle \mathbf{w}_{k,r}, \mathbf{x}_i \rangle \geq 0, \langle \mathbf{w}_{k,r}, \mathbf{x}_j \rangle \geq 0\} - \eta \left(\mathcal{E}_{1,k}^{(i)} + \mathcal{E}_{2,k}^{(i)}\right) \\
&= \eta \theta \tau \sum_{j=1}^{n} \left(\mathbf{H}(k)_{ij} - \mathbf{H}(k)_{ij}^{\perp}\right)(y_j - u_k^{(j)}) - \eta \left(\mathcal{E}_{1,k}^{(i)} + \mathcal{E}_{2,k}^{(i)}\right)
\end{aligned}
$$

where

$$
\mathcal{E}_{1,k}^{(i)} = \frac{\theta \xi (1-\xi) \tau}{m^{\frac{3}{2}}} \sum_{r \in S_i} \sum_{j=1}^{n} a_r \langle \mathbf{x}_i, \mathbf{x}_j \rangle \sigma(\langle \mathbf{w}_{k,r}, \mathbf{x}_j \rangle)
$$

$$
\mathcal{E}_{2,k}^{(i)} = \frac{\xi}{\sqrt{m}} \sum_{r \in S_i} a_r \left\langle \mathbb{E}_{\mathbf{M}_k}\left[\eta_{k,r} \sum_{l=1}^{p} \sum_{t=1}^{\tau-1} \left(\frac{\partial L_{\mathbf{m}_k^l}\left(\mathbf{W}_{k,t}^l\right)}{\partial \mathbf{w}_r} - \frac{\partial L_{\mathbf{m}_k^l}\left(\mathbf{W}_k\right)}{\partial \mathbf{w}_r}\right)\right], \mathbf{x}_i \right\rangle \mathbb{I}\{\langle \mathbf{w}_{k,r}, \mathbf{x}_i \rangle \geq 0\}
$$

Let $\mathcal{E}_{1,k} = \left[\mathcal{E}_{1,k}^{(1)}, \ldots, \mathcal{E}_{1,k}^{(n)}\right]$, and $\mathcal{E}_{2,k} = \left[\mathcal{E}_{2,k}^{(1)}, \ldots, \mathcal{E}_{2,k}^{(n)}\right]$. Then we have

$$
\mathbb{E}_{\mathbf{M}_k}\left[\mathbf{I}_{1,k}\right] = \eta \theta \tau \left(\mathbf{H}(k) - \mathbf{H}(k)^{\perp}\right)(\mathbf{y} - \mathbf{u}_k) - \eta \left(\mathcal{E}_{1,k} + \mathcal{E}_{2,k}\right)
$$

Thus,

$$
\begin{aligned}
\mathbb{E}_{\mathbf{M}_k}\left[\mathbf{I}_{1,k}'\right] &= \mathbb{E}_{\mathbf{M}_k}\left[\mathbf{I}_{1,k}\right] - \eta \theta \tau \mathbf{H}(k)(\mathbf{y} - \mathbf{u}_k) \\
&= \eta \theta \tau \mathbf{H}(k)^{\perp}(\mathbf{y} - \mathbf{u}_k) + \eta \left(\mathcal{E}_{1,k} + \mathcal{E}_{2,k}\right)
\end{aligned}
$$

According to Lemma 7 and Lemma 8, we have the bound of $\mathcal{E}_{1,k}^{(i)}$ and $\mathcal{E}_{2,k}^{(i)}$ as

$$
\left|\mathcal{E}_{1,k}^{(i)}\right| \leq \theta \xi (1-\xi) \tau n \kappa \sqrt{\frac{2d}{m}}
$$

$$
\left|\mathcal{E}_{2,k}^{(i)}\right| \leq \frac{\eta \xi \tau (\tau - 1) n^{\frac{3}{2}}}{2} \left(\theta \|\mathbf{y} - \mathbf{u}_k\|_2 + \sqrt{pC_1}\right)
$$

Moreover, according to Lemma 17, we have

$$
\|\mathbf{H}(k)^{\perp}\|_2 \leq 4\xi n \kappa^{-1} R
$$

Let $R \leq \frac{\kappa \lambda_0}{128n}$, we have

$$
\|\mathbf{H}(k)^{\perp}\|_2 \leq \frac{\lambda_0}{32}
$$

Therefore, we have

$$
\begin{aligned}
\left|\left\langle \mathbf{y} - \mathbf{u}_k, \mathbb{E}_{\mathbf{M}_k}\left[\mathbf{I}'_{1,k}\right]\right\rangle\right| &\leq \eta\theta\tau\left|\left\langle \mathbf{y} - \mathbf{u}_k, \mathbf{H}(k)^{\perp}(\mathbf{y} - \mathbf{u}_k)\right\rangle\right| + \eta\sum_{i=1}^{n}\left(\left|\left(y_i - u_k^{(i)}\right)\mathcal{E}_{1,k}^{(i)}\right| + \left|\left(y_i - u_k^{(i)}\right)\mathcal{E}_{2,k}^{(i)}\right|\right) \\
&= \eta\theta\tau\|\mathbf{H}(k)^{\perp}\|_2\|\mathbf{y} - \mathbf{u}_k\|_2^2 + \max_{i\in[n]}\left(\left|\mathcal{E}_{1,k}^{(i)}\right| + \left|\mathcal{E}_{2,k}^{(i)}\right|\right)\eta\sum_{i=1}^{n}\left|y_i - u_k^{(i)}\right| \\
&\leq \frac{1}{32}\eta\theta\tau\lambda_0\|\mathbf{y} - \mathbf{u}_k\|_2^2 + \max_{i\in[n]}\left(\left|\mathcal{E}_{1,k}^{(i)}\right| + \left|\mathcal{E}_{2,k}^{(i)}\right|\right)\eta\sqrt{n}\|\mathbf{y} - \mathbf{u}_k\|_2 \\
&\leq \frac{1}{32}\eta\theta\tau\lambda_0\|\mathbf{y} - \mathbf{u}_k\|_2^2 + \frac{\eta^2\theta\xi\tau(\tau-1)n^2}{2}\|\mathbf{y} - \mathbf{u}_k\|_2^2 + \\
&\quad \left(\eta\theta\xi(1-\xi)\tau\kappa\sqrt{\frac{2n^3 d}{m}} + \frac{\eta^2\xi\tau(\tau-1)n^2}{2}\sqrt{pC_1}\right)\|\mathbf{y} - \mathbf{u}_k\|_2
\end{aligned}
$$

Using the general inequality that $ab \leq \frac{1}{2}(a^2 + b^2)$, and $\eta \leq \frac{\lambda_0}{16(\tau-1)n^2}$, we get

$$
\left|\left\langle \mathbf{y} - \mathbf{u}_k, \mathbb{E}_{\mathbf{M}_k}\left[\mathbf{I}'_{1,k}\right]\right\rangle\right| \leq \frac{1}{8}\eta\theta\tau\lambda_0\|\mathbf{y} - \mathbf{u}_k\|_2^2 + \frac{16\eta\theta\tau\xi^2(1-\xi)^2\kappa^2 n^3 d}{m\lambda_0} + \frac{2\eta^3\xi^2\tau(\tau-1)^2 n^4 pC_1}{\theta\lambda_0}
$$

$\square$

**Lemma 7.** *Under the assumption of Theorem 3 we have that for all $k \in [K], i \in [n]$, it holds that*

$$
\left|\mathcal{E}_{1,k}^{(i)}\right| \leq \theta\xi(1-\xi)\tau n\kappa\sqrt{\frac{2d}{m}}
$$

*Proof.* We have

$$
\begin{aligned}
\left|\mathcal{E}_{1,k}^{(i)}\right| &\leq \frac{\theta\xi(1-\xi)\tau}{m^{\frac{3}{2}}}\left|\sum_{r\in S_i}\sum_{j=1}^{n}a_r\left\langle \mathbf{x}_i, \mathbf{x}_j\right\rangle\sigma(\left\langle \mathbf{w}_{k,r}, \mathbf{x}_j\right\rangle)\right| \\
&\leq \frac{\theta\xi(1-\xi)\tau}{m^{\frac{3}{2}}}\sum_{r\in S_i}\sum_{j=1}^{n}\left|\left\langle \mathbf{w}_{k,r}, \mathbf{x}_i\right\rangle\right| \\
&\leq \frac{\theta\xi(1-\xi)\tau n}{m^{\frac{3}{2}}}\sum_{r\in S_i}\|\mathbf{w}_{k,r}\|_2 \\
&\leq \frac{\theta\xi(1-\xi)\tau n}{m^{\frac{3}{2}}}\sum_{r\in S_i}\left(\|\mathbf{w}_{0,r}\|_2 + R\right) \\
&\leq \frac{\theta\xi(1-\xi)\tau n}{m}\|\mathbf{W}_0\|_F + \frac{\theta\xi(1-\xi)\tau n R}{\sqrt{m}} \\
&\leq \theta\xi(1-\xi)\tau n\kappa\sqrt{\frac{2d}{m}}
\end{aligned}
$$

where for the bound of $\|\mathbf{W}_0\|_F$ we use Lemma 22. $\square$

**Lemma 8.** *Suppose $\|\mathbf{w}_{k,t,r} - \mathbf{w}_{0,r}\|_2 \leq R$ for all $r \in [m]$. Then we have*

$$
\left|\mathcal{E}_{2,t}^{(i)}\right| \leq \frac{\eta\xi\tau(\tau-1)n^{\frac{3}{2}}}{2}\left(\theta\|\mathbf{y} - \mathbf{u}_k\|_2 + \sqrt{pC_1}\right)
$$

*Proof.* Since $r \in S_i$, the difference between the surrogate gradients of a sub-network has the form

$$\left\| \frac{\partial L_{\mathbf{m}_k^l}\left(\mathbf{W}_{k,t}^l\right)}{\partial \mathbf{w}_r} - \frac{\partial L_{\mathbf{m}_k^l}\left(\mathbf{W}_k\right)}{\partial \mathbf{w}_r} \right\|_2 = \frac{1}{\sqrt{m}} \left\| \sum_{j=1}^n a_r m_{k,r}^l \mathbf{x}_j \left( \hat{u}_{k,t}^{l(j)} - \hat{u}_k^{l(j)} \right) \mathbb{I}\{\langle \mathbf{w}_{k,r}, \mathbf{x}_j \rangle \geq 0\} \right\|_2$$

$$\leq \frac{m_{k,r}^l}{\sqrt{m}} \sum_{j=1}^n \left| \hat{u}_{k,t}^{l(j)} - \hat{u}_k^{l(j)} \right|$$

Therefore, using the convexity of $\ell_2$-norm,

$$\left\| \mathbb{E}_{\mathbf{M}_k} \left[ \eta_{k,r} \sum_{l=1}^p \left( \frac{\partial L_{\mathbf{m}_k^l}\left(\mathbf{W}_{k,t}^l\right)}{\partial \mathbf{w}_r} - \frac{\partial L_{\mathbf{m}_k^l}\left(\mathbf{W}_k\right)}{\partial \mathbf{w}_r} \right) \right] \right\|_2 \leq \mathbb{E}_{\mathbf{M}_k} \left[ \eta_{k,r} \left\| \sum_{l=1}^p \left( \frac{\partial L_{\mathbf{m}_k^l}\left(\mathbf{W}_{k,t}^l\right)}{\partial \mathbf{w}_r} - \frac{\partial L_{\mathbf{m}_k^l}\left(\mathbf{W}_k\right)}{\partial \mathbf{w}_r} \right) \right\|_2 \right]$$

$$\leq \mathbb{E}_{\mathbf{M}_k} \left[ \eta_{k,r} \sum_{l=1}^p \left\| \frac{\partial L_{\mathbf{m}_k^l}\left(\mathbf{W}_{k,t}^l\right)}{\partial \mathbf{w}_r} - \frac{\partial L_{\mathbf{m}_k^l}\left(\mathbf{W}_k\right)}{\partial \mathbf{w}_r} \right\|_2 \right]$$

$$\leq \mathbb{E}_{\mathbf{M}_k} \left[ \frac{\eta_{k,r}}{\sqrt{m}} \sum_{l=1}^p m_{k,r} \sum_{j=1}^n \left| \hat{u}_{k,t}^{l(j)} - \hat{u}_k^{l(j)} \right| \right]$$

By Lemma 5, we have

$$\left| \hat{u}_{k,t}^{l(i)} - \hat{u}_k^{l(i)} \right| \leq \eta t \sqrt{n} \left( \|\mathbf{y} - \mathbf{u}_k\|_2 + \|\mathbf{u}_k - \hat{\mathbf{u}}_k^l\|_2 \right)$$

Therefore,

$$\left| \mathcal{E}_{2,t}^{(i)} \right| \leq \frac{\xi}{\sqrt{m}} \sum_{r \in S_i} \left\| \mathbb{E}_{\mathbf{M}_k} \left[ \eta_{k,r} \sum_{l=1}^p \sum_{t=1}^{\tau-1} \left( \frac{\partial L_{\mathbf{m}_k^l}\left(\mathbf{W}_{k,t}^l\right)}{\partial \mathbf{w}_r} - \frac{\partial L_{\mathbf{m}_k^l}\left(\mathbf{W}_k\right)}{\partial \mathbf{w}_r} \right) \right] \right\|_2$$

$$\leq \frac{\xi}{\sqrt{m}} \sum_{t=1}^{\tau-1} \sum_{r \in S_i} \left\| \mathbb{E}_{\mathbf{M}_k} \left[ \eta_{k,r} \sum_{l=1}^p \left( \frac{\partial L_{\mathbf{m}_k^l}\left(\mathbf{W}_{k,t}^l\right)}{\partial \mathbf{w}_r} - \frac{\partial L_{\mathbf{m}_k^l}\left(\mathbf{W}_k\right)}{\partial \mathbf{w}_r} \right) \right] \right\|_2$$

$$\leq \frac{\xi}{\sqrt{m}} \sum_{t=1}^{\tau-1} \sum_{r \in S_i} \mathbb{E}_{\mathbf{M}_k} \left[ \frac{\eta_{k,r}}{\sqrt{m}} \sum_{l=1}^p m_{k,r} \sum_{j=1}^n \left| \hat{u}_{k,t}^{l(j)} - \hat{u}_k^{l(j)} \right| \right]$$

$$\leq \frac{\eta \xi n^{\frac{3}{2}}}{m} \sum_{t=1}^{\tau-1} t \sum_{r \in S_i} \mathbb{E}_{\mathbf{M}_k} \left[ \eta_{k,r} \sum_{l=1}^p m_{k,r} \left( \|\mathbf{y} - \mathbf{u}_k\|_2 + \|\mathbf{y} - \hat{\mathbf{u}}_k^l\|_2 \right) \right]$$

$$\leq \frac{\eta \xi \tau(\tau-1) n^{\frac{3}{2}}}{2} \left( ]\theta \|\mathbf{y} - \mathbf{u}_k\|_2 + \mathbb{E}_{\mathbf{M}_k} \left[ \eta_{k,r} \sum_{l=1}^p m_{k,r}^l \|\mathbf{u}_k - \hat{\mathbf{u}}_k^l\|_2 \right] \right)$$

$$\leq \frac{\eta \xi \tau(\tau-1) n^{\frac{3}{2}}}{2} \left( \theta \|\mathbf{y} - \mathbf{u}_k\|_2 + \sqrt{pC_1} \right)$$

where the last inequality follows from Lemma 24. $\qquad \square$

**Lemma 9.** *Under the condition of Theorem 3, we have*

$$|\langle \mathbf{y} - \mathbf{u}_k, \mathbb{E}_{\mathbf{M}_k}[\mathbf{I}_2]\rangle| \leq \frac{1}{8}\eta\theta\tau\lambda_0 \|\mathbf{y} - \mathbf{u}_k\|_2^2 + \frac{\eta\lambda_0\xi^2(\theta - \xi^2)n\kappa^2}{24p\tau} + \frac{\eta\lambda_0\xi^2(\tau-1)^2 pC_1}{96\tau\theta}$$

*Proof.* To start, we notice that Using the 1-Lipschitzness of ReLU, we have

$$
\mathbb{E}_{\mathbf{M}_k}\left[\left|I_{2,k}^{(i)}\right|\right] = \frac{\xi}{\sqrt{m}}\mathbb{E}_{\mathbf{M}_k}\left[\left|\sum_{r\in S_i^\perp} a_r\left(\sigma(\langle\mathbf{w}_{k+1,r},\mathbf{x}_i\rangle - \sigma(\langle\mathbf{w}_{k+1,r},\mathbf{x}_i\rangle)\right)\right|\right]
$$

$$
\leq \frac{\xi}{\sqrt{m}}\sum_{r\in S_i^\perp}\mathbb{E}_{\mathbf{M}_k}\left[|\sigma(\langle\mathbf{w}_{k+1,r},\mathbf{x}_i\rangle - \sigma(\langle\mathbf{w}_{k+1,r},\mathbf{x}_i\rangle|\right]
$$

$$
\leq \frac{\xi}{\sqrt{m}}\sum_{r\in S_i^\perp}\mathbb{E}_{\mathbf{M}_k}\left[|\langle\mathbf{w}_{k+1,r}-\mathbf{w}_{k,r},\mathbf{x}_i\rangle|\right]
$$

$$
\leq \frac{\xi}{\sqrt{m}}\sum_{r\in S_i^\perp}\mathbb{E}_{\mathbf{M}_k}\left[\|\mathbf{w}_{k+1,r}-\mathbf{w}_{k,r}\|_2\right]
$$

$$
\leq \frac{\eta\xi}{\sqrt{m}}\sum_{r\in S_i^\perp}\mathbb{E}_{\mathbf{M}_k}\left[\left\|\eta_{k,r}\sum_{t=0}^{\tau-1}\sum_{l=1}^{p}\frac{\partial L_{\mathbf{m}_k^l}\left(\mathbf{W}_{k,t}^l\right)}{\partial\mathbf{w}_r}\right\|_2\right]
$$

$$
\leq \frac{\eta\xi}{\sqrt{m}}\sum_{r\in S_i^\perp}\left(\mathbb{E}_{\mathbf{M}_k}\left[\|\mathbf{g}_{k,r}\|_2\right] + \mathbb{E}_{\mathbf{M}_k}\left[\eta_{k,r}\sum_{t=1}^{\tau-1}\sum_{l=1}^{p}\left\|\frac{\partial L_{\mathbf{m}_k}\left(\mathbf{W}_{k,t}^l\right)}{\partial\mathbf{w}_r}\right\|_2\right]\right)
$$

$$
\leq \frac{\eta\theta\xi\sqrt{n}}{m}|S_i^\perp|\|\mathbf{y}-\mathbf{u}_k\|_2 + \frac{4\eta\xi\kappa n|S_i^\perp|}{m}\sqrt{\frac{\theta(\theta-\xi^2)}{p}} +
$$

$$
\frac{\eta\xi\sqrt{n}}{m}\sum_{r\in S_i^\perp}\mathbb{E}_{\mathbf{M}_k}\left[\eta_{k,r}\sum_{t=1}^{\tau-1}\sum_{l=1}^{p}m_{k,r}^l\|\mathbf{y}-\hat{\mathbf{u}}_{k,t}^l\|_2\right]
$$

$$
\leq \frac{\eta\theta\xi\sqrt{n}}{m}|S_i^\perp|\|\mathbf{y}-\mathbf{u}_k\|_2 + \frac{4\eta\xi\kappa n|S_i^\perp|}{m}\sqrt{\frac{\theta(\theta-\xi^2)}{p}} +
$$

$$
\frac{\eta\xi\sqrt{n}}{m}\sum_{r\in S_i^\perp}\mathbb{E}_{\mathbf{M}_k}\left[\eta_{k,r}\sum_{t=1}^{\tau-1}\sum_{l=1}^{p}m_{k,r}^l\|\mathbf{y}-\hat{\mathbf{u}}_k^l\|_2\right]
$$

$$
\leq \frac{\eta\theta\xi\tau\sqrt{n}}{m}|S_i^\perp|\|\mathbf{y}-\mathbf{u}_k\|_2 + \frac{4\eta\xi\kappa n|S_i^\perp|}{m}\sqrt{\frac{\theta(\theta-\xi^2)}{p}} +
$$

$$
\frac{\eta\xi\sqrt{n}}{m}\sum_{r\in S_i^\perp}\mathbb{E}_{\mathbf{M}_k}\left[\eta_{k,r}\sum_{t=1}^{\tau-1}\sum_{l=1}^{p}m_{k,r}^l\|\mathbf{u}_k-\hat{\mathbf{u}}_k^l\|_2\right]
$$

where in the seventh inequality we use the bound on $\mathbb{E}_{\mathbf{M}_k}\left[\|\mathbf{g}_{k,r}\|_2\right]$ from Lemma 4. Moreover, using Lemma 24 we have

$$
\mathbb{E}_{\mathbf{M}_k}\left[\eta_{k,r}\sum_{l=1}^{p}m_{k,r}^l\|\mathbf{u}_k-\hat{\mathbf{u}}_k^l\|_2\right] \leq \sqrt{pC_1}
$$

Then we have

$$
\mathbb{E}_{\mathbf{M}_k}\left[\left|I_{2,k}^{(i)}\right|\right] \leq \frac{\eta\theta\xi\tau\sqrt{n}}{m}|S_i^\perp|\|\mathbf{y}-\mathbf{u}_k\|_2 + \frac{4\eta\xi\kappa n|S_i^\perp|}{m}\sqrt{\frac{\theta(\theta-\xi^2)}{p}} + \frac{\eta\xi(\tau-1)}{m}\sqrt{npC_1}|S_i^\perp|
$$

$$
\leq 8\eta\theta\xi\tau\sqrt{n}\kappa^{-1}R\|\mathbf{y}-\mathbf{u}_k\|_2 + 16\eta\xi nR\sqrt{\frac{\theta(\theta-\xi^2)}{p}} + 4\eta\xi(\tau-1)\kappa^{-1}R\sqrt{npC_1}
$$

where in the last inequality we use $|S_i^\perp| \le 4m\kappa^{-1}R$. Therefore,

$$
\begin{aligned}
|\langle \mathbf{y} - \mathbf{u}_k, \mathbb{E}_{\mathbf{M}_k}[\mathbf{I}_{2,k}]\rangle| &= \left| \sum_{i=1}^n (y_i - u_k^{(i)}) \mathbb{E}_{\mathbf{M}_k}\left[I_{2,k}^{(i)}\right] \right| \\
&\le \sum_{i=1}^n \left| y_i - u_k^{(i)} \right| \cdot \left| \mathbb{E}_{\mathbf{M}_k}\left[I_{2,k}^{(i)}\right] \right| \\
&\le \max_{i\in[n]} \left| \mathbb{E}_{\mathbf{M}_k}\left[I_{2,k}^{(i)}\right] \right| \sum_{i=1}^n \left| y_i - u_k^{(i)} \right| \\
&\le \sqrt{n} \max_{i\in[n]} \left| \mathbb{E}_{\mathbf{M}_k}\left[I_{2,k}^{(i)}\right] \right| \|\mathbf{y} - \mathbf{u}_k\|_2 \\
&\le 8\eta\theta\xi\tau\kappa^{-1}nR\|\mathbf{y} - \mathbf{u}_k\|_2^2 + 16\eta\xi R\sqrt{\frac{\theta(\theta - \xi^2)n^3}{p}}\|\mathbf{y} - \mathbf{u}_k\|_2 + \\
&\qquad 4\eta\xi(\tau - 1)\kappa^{-1}nR\sqrt{pC_1}\|\mathbf{y} - \mathbf{u}_k\|_2 \\
&\le \frac{1}{8}\eta\theta\tau\lambda_0\|\mathbf{y} - \mathbf{u}_k\|_2^2 + \frac{\eta\lambda_0\xi^2(\theta - \xi^2)n\kappa^2}{24p\tau} + \frac{\eta\lambda_0\xi^2(\tau - 1)^2pC_1}{96\tau\theta}
\end{aligned}
$$

where in the last inequality we use $R \le \frac{\kappa\lambda_0}{192n}$ and $ab \le \frac{1}{2}(a^2 + b^2)$. $\qquad\square$

**Lemma 10.** *Under the condition of Theorem 3, with $\eta \le \frac{\lambda_0}{48n\tau\max\{n,p\}}$, we have*

$$
\mathbb{E}_{\mathbf{M}_k}\left[\|\mathbf{u}_{k+1} - \mathbf{u}_k\|_2^2\right] \le \frac{1}{4}\eta\theta\tau\lambda_0\|\mathbf{y} - \mathbf{u}_k\|_2^2 + \frac{17\eta^2\xi^2\tau^2\theta(\theta - \xi^2)n^3\kappa^2}{p} + \eta^2\xi^2\lambda_0(\tau - 1)^2pnC_1
$$

*Proof.* As in previous lemma, we use the Lipschitzness of ReLU to get

$$
\begin{aligned}
\mathbb{E}_{\mathbf{M}_k}\left[\left(u_{k+1}^{(i)} - u_k^{(i)}\right)^2\right] &\le \frac{\xi^2}{m}\mathbb{E}_{\mathbf{M}_k}\left[\left(\sum_{r=1}^m a_r\left(\sigma(\langle\mathbf{w}_{k+1,r},\mathbf{x}_i\rangle) - \sigma(\langle\mathbf{w}_{k,r},\mathbf{x}_i\rangle)\right)\right)^2\right] \\
&\le \xi^2 \sum_{r=1}^m \mathbb{E}_{\mathbf{M}_k}\left[\left(\sigma(\langle\mathbf{w}_{k+1,r},\mathbf{x}_i\rangle) - \sigma(\langle\mathbf{w}_{k,r},\mathbf{x}_i\rangle)\right)^2\right] \\
&\le \xi^2 \sum_{r=1}^m \mathbb{E}_{\mathbf{M}_k}\left[\langle\mathbf{w}_{k+1,r} - \mathbf{w}_{k,r},\mathbf{x}_i\rangle^2\right] \\
&\le \xi^2 \sum_{r=1}^m \mathbb{E}_{\mathbf{M}_k}\left[\|\mathbf{w}_{k+1,r} - \mathbf{w}_{k,r}\|_2^2\right] \\
&= \xi^2\left(D_{1,k} + D_{2,k}\right)
\end{aligned}
$$

where

$$
D_{1,k} = \sum_{r\in S_i} \mathbb{E}_{\mathbf{M}_k}\left[\|\mathbf{w}_{k+1,r} - \mathbf{w}_{k,r}\|_2^2\right]
$$

$$
D_{2,k} = \sum_{r\in S_i^\perp} \mathbb{E}_{\mathbf{M}_k}\left[\|\mathbf{w}_{k+1,r} - \mathbf{w}_{k,r}\|_2^2\right]
$$

Using Lemma 11 and Lemma 12 we have

$$
D_{1,k} \le \left(4\eta^2\tau^2n\theta + 4\eta^4\theta n^3\tau^3(\tau - 1)p\right)\|\mathbf{y} - \mathbf{u}_k\|_2^2 + \frac{16\eta^2\tau^2\theta(\theta - \xi^2)n^2\kappa^2}{p} + 4\eta^4n^3\tau^2(\tau - 1)^2pC_1
$$

$$
D_{2,k} \le \frac{\eta^2\theta\tau\lambda_0}{18}\left(1 + (\tau - 1)p\right)\|\mathbf{y} - \mathbf{u}_k\|_2^2 + \frac{4\eta^2\lambda_0\theta(\theta - \xi^2)\tau n\kappa^2}{9p} + \frac{\eta^2\tau(\tau - 1)\lambda_0pC_1}{18}
$$

Therefore we have

$$\mathbb{E}_{\mathbf{M}_k}\left[\|\mathbf{u}_{k+1} - \mathbf{u}_k\|_2^2\right] \leq \xi^2 n \left(D_{1,k} + D_{2,k}\right)$$

$$\leq \left(4\eta^2\xi^2\tau^2 n^2\theta + 4\eta^4\theta\xi^2 n^4\tau^3(\tau-1)p + \frac{\eta^2\theta\tau n\lambda_0}{18}\left(1 + (\tau-1)p\right)\right)\|\mathbf{y} - \mathbf{u}_k\|_2^2 +$$

$$\frac{16\eta^2\xi^2\tau^2\theta(\theta-\xi^2)n^3\kappa^2}{p} + 4\eta^4\xi^2 n^4\tau^2(\tau-1)^2 pC_1 + \frac{4\eta^2\lambda_0\xi^2\theta(\theta-\xi^2)\tau n^2\kappa^2}{9p} +$$

$$\frac{\eta^2\xi^2\tau(\tau-1)n\lambda_0 pC_1}{18}$$

With $\eta \leq \frac{\lambda_0}{48n\tau\max\{n,p\}}$, we have

$$\mathbb{E}_{\mathbf{M}_k}\left[\|\mathbf{u}_{k+1} - \mathbf{u}_k\|_2^2\right] \leq \frac{1}{4}\eta\theta\tau\lambda_0\|\mathbf{y} - \mathbf{u}_k\|_2^2 + \frac{17\eta^2\xi^2\tau^2\theta(\theta-\xi^2)n^3\kappa^2}{p} + \eta^2\xi^2\lambda_0(\tau-1)^2 pnC_1$$

$$\square$$

**Lemma 11.**

$$D_{1,k} \leq \left(4\eta^2\tau^2 n\theta + 4\eta^4\theta n^3\tau^3(\tau-1)p\right)\|\mathbf{y} - \mathbf{u}_k\|_2^2 + \frac{16\eta^2\tau^2\theta(\theta-\xi^2)n^2\kappa^2}{p} + 4\eta^4 n^3\tau^2(\tau-1)^2 pC_1$$

*Proof.* We have

$$D_{1,k} = \eta^2\sum_{r\in S_i}\mathbb{E}_{\mathbf{M}_k}\left[\left\|\eta_{k,r}\sum_{t=0}^{\tau-1}\sum_{l=1}^{p}\frac{\partial L_{\mathbf{m}_k^l}\left(\mathbf{W}_{k,t}^l\right)}{\partial\mathbf{w}_r}\right\|_2^2\right]$$

$$\leq \eta^2\sum_{r\in S_i}\mathbb{E}_{\mathbf{M}_k}\left[\left\|\tau\mathbf{g}_{k,r} + \eta_{k,r}\sum_{t=1}^{\tau-1}\sum_{l=1}^{p}\left(\frac{\partial L_{\mathbf{m}_k^l}\left(\mathbf{W}_{k,t}^l\right)}{\partial\mathbf{w}_r} - \frac{\partial L_{\mathbf{m}_k^l}\left(\mathbf{W}_k\right)}{\partial\mathbf{w}_r}\right)\right\|_2^2\right]$$

$$\leq 2\eta^2\tau^2\sum_{r\in S_i}\mathbb{E}_{\mathbf{M}_k}\left[\|\mathbf{g}_{k,r}\|_2^2\right] + 2\eta^2(\tau-1)p\sum_{r\in S_i}\sum_{t=1}^{\tau-1}\mathbb{E}_{\mathbf{M}_k}\left[\eta_{k,r}^2\sum_{l=1}^{p}\left\|\frac{\partial L_{\mathbf{m}_k^l}\left(\mathbf{W}_{k,t}^l\right)}{\partial\mathbf{w}_r} - \frac{\partial L_{\mathbf{m}_k^l}\left(\mathbf{W}_k\right)}{\partial\mathbf{w}_r}\right\|_2^2\right]$$

Note that for $r \in S_i$, we have

$$\left\|\frac{\partial L_{\mathbf{m}_k^l}\left(\mathbf{W}_{k,t}^l\right)}{\partial\mathbf{w}_r} - \frac{\partial L_{\mathbf{m}_k^l}\left(\mathbf{W}_k\right)}{\partial\mathbf{w}_r}\right\|_2^2 = \frac{m_{k,r}}{m}\left\|\sum_{i=1}^{n}a_r\mathbf{x}_i\left(\hat{u}_{k,t}^{l(i)} - \hat{u}_k^{l(i)}\right)\mathbb{I}\{\langle\mathbf{w}_{k,r}, \mathbf{x}_i\rangle \geq 0\}\right\|_2^2$$

$$\leq \frac{nm_{k,r}}{m}\sum_{i=1}^{n}\left(\hat{u}_{k,t}^{l(i)} - \hat{u}_k^{l(i)}\right)^2$$

$$\leq \frac{n^2 m_{k,r}^l}{m}\left(2\eta^2 t^2 n\|\mathbf{y} - \mathbf{u}_k\|_2^2 + 2\eta^2 t^2 n\|\mathbf{u}_k - \hat{\mathbf{u}}_k^l\|_2^2\right)$$

where in the last inequality we use Lemma 5. Plugging in the bound above and the bound on $\mathbb{E}_{\mathbf{M}_k}\left[\|\mathbf{g}_k\|_2^2\right]$ from Lemma 4 gives

$$D_{1,k} \leq 4\eta^2\tau^2 n\theta\|\mathbf{y} - \mathbf{u}_k\|_2^2 + \frac{16\eta^2\tau^2\theta(\theta-\xi^2)n^2\kappa^2}{p} + 4\eta^4\theta n^3\tau^3(\tau-1)p\|\mathbf{y} - \mathbf{u}_k\|_2^2 +$$

$$4\eta^4 n^3\tau^3(\tau-1)p\mathbb{E}_{\mathbf{M}_k}\left[\eta_{k,r}^2\sum_{l=1}^{p}m_{k,r}^l\|\mathbf{u}_k - \hat{\mathbf{u}}_k^l\|_2^2\right]$$

$$\leq \left(4\eta^2\tau^2 n\theta + 4\eta^4\theta n^3\tau^3(\tau-1)p\right)\|\mathbf{y} - \mathbf{u}_k\|_2^2 + \frac{16\eta^2\tau^2\theta(\theta-\xi^2)n^2\kappa^2}{p} + 4\eta^4 n^3\tau^2(\tau-1)^2 pC_1$$

$$\square$$

**Lemma 12.**

$$D_{2,k} \leq \frac{\eta^2 \theta \tau \lambda_0}{18} \left(1 + (\tau - 1)p\right) \|\mathbf{y} - \mathbf{u}_k\|_2^2 + \frac{4\eta^2 \lambda_0 \theta(\theta - \xi^2)\tau n \kappa^2}{9p} + \frac{\eta^2 \tau(\tau - 1)\lambda_0 p C_1}{18}$$

*Proof.*

$$D_{2,k} = \eta^2 \sum_{r \in S_i^\perp} \mathbb{E}_{\mathbf{M}_k} \left[ \left\| \eta_{k,r} \sum_{t=0}^{\tau-1} \sum_{l=1}^{p} \frac{\partial L_{\mathbf{m}_k^l}\left(\mathbf{W}_{k,t}^l\right)}{\partial \mathbf{w}_r} \right\|_2^2 \right]$$

$$\leq \eta^2 \tau \sum_{r \in S_i^\perp} \left( \mathbb{E}_{\mathbf{M}_k} \left[ \|\mathbf{g}_{k,r}\|_2^2 \right] + \sum_{t=1}^{\tau-1} \mathbb{E}_{\mathbf{M}_k} \left[ \left\| \eta_{k,r} \sum_{l=1}^{p} \frac{\partial L_{\mathbf{m}_k^l}\left(\mathbf{W}_{k,t}^l\right)}{\partial \mathbf{w}_r} \right\|_2^2 \right] \right)$$

$$\leq \eta^2 \tau \sum_{r \in S_i^\perp} \left( \mathbb{E}_{\mathbf{M}_k} \left[ \|\mathbf{g}_{k,r}\|_2^2 \right] + p \sum_{t=1}^{\tau-1} \mathbb{E}_{\mathbf{M}_k} \left[ \eta_{k,r}^2 \sum_{l=1}^{p} \left\| \frac{\partial L_{\mathbf{m}_k}\left(\mathbf{W}_{k,t}^l\right)}{\partial \mathbf{w}_r} \right\|_2^2 \right] \right)$$

$$\leq \frac{2\eta^2 \tau n \theta}{m} |S_i^\perp| \|\mathbf{y} - \mathbf{u}_k\|_2^2 + \frac{8\eta^2 \theta(\theta - \xi^2)\tau n^2 \kappa^2}{pm} |S_i^\perp| + \frac{\eta^2 \tau n p}{m} |S_i^\perp| \sum_{t=1}^{\tau-1} \mathbb{E}_{\mathbf{M}_k} \left[ \eta_{k,r}^2 \sum_{l=1}^{p} m_{k,r}^l \|\mathbf{y} - \hat{\mathbf{u}}_{k,t}^l\|_2^2 \right]$$

$$\leq \frac{2\eta^2 \tau n \theta}{m} |S_i^\perp| \|\mathbf{y} - \mathbf{u}_k\|_2^2 + \frac{8\eta^2 \theta(\theta - \xi^2)\tau n^2 \kappa^2}{pm} |S_i^\perp| + \frac{2\eta^2 \theta \tau(\tau - 1)np}{m} |S_i^\perp| \|\mathbf{y} - \mathbf{u}_k\|_2^2 +$$
$$\frac{2\eta^2 \tau(\tau - 1)np}{m} |S_i^\perp| \mathbb{E}_{\mathbf{M}_k} \left[ \eta_{k,r}^2 \sum_{l=1}^{p} m_{k,r}^l \|\mathbf{u}_k - \hat{\mathbf{u}}_k^l\|_2^2 \right]$$

Using $|S_i^\perp| \leq 4m\kappa^{-1}R$ with $R \leq \frac{\xi \kappa \lambda_0}{144n}$ gives

$$D_{2,k} \leq \frac{\eta^2 \theta \tau \lambda_0}{18} \left(1 + (\tau - 1)p\right) \|\mathbf{y} - \mathbf{u}_k\|_2^2 + \frac{4\eta^2 \lambda_0 \theta(\theta - \xi^2)\tau n \kappa^2}{9p} +$$
$$\frac{\eta^2 \tau(\tau - 1)\lambda_0 p}{18} \mathbb{E}_{\mathbf{M}_k} \left[ \eta_{k,r}^2 \sum_{l=1}^{p} m_{k,r}^l \|\mathbf{u}_k - \hat{\mathbf{u}}_k^l\|_2^2 \right]$$

$$\leq \frac{\eta^2 \theta \tau \lambda_0}{18} \left(1 + (\tau - 1)p\right) \|\mathbf{y} - \mathbf{u}_k\|_2^2 + \frac{4\eta^2 \lambda_0 \theta(\theta - \xi^2)\tau n \kappa^2}{9p} + \frac{\eta^2 \tau(\tau - 1)\lambda_0 p C_1}{18}$$

$\square$

# H    Lemmas for Theorem 4

**Lemma 13.** *The $k$th global step produce the squared error satisfying*

$$\|\mathbf{y} - \mathbf{u}_{k+1}\|_2^2 = \frac{1}{p} \sum_{l=1}^{p} \|\mathbf{y} - \hat{\mathbf{u}}_{k,\tau}^l\|_2^2 - \frac{1}{p^2} \sum_{l=1}^{p} \sum_{l'=1}^{l-1} \|\hat{\mathbf{u}}_{k,\tau}^l - \hat{\mathbf{u}}_{k,\tau}^{l'}\|_2^2$$

*Proof.* We have

$$
\begin{aligned}
\|\mathbf{y} - \mathbf{u}_{k+1}\|_2^2 &= \left\| \mathbf{y} - \frac{1}{p} \sum_{l=1}^{p} \hat{\mathbf{u}}_{k,\tau}^l \right\|_2^2 \\
&= \frac{1}{p^2} \sum_{l=1}^{p} \sum_{l'=1}^{p} \left\langle \mathbf{y} - \hat{\mathbf{u}}_{k,\tau}^l, \mathbf{y} - \hat{\mathbf{u}}_{k,\tau}^{l'} \right\rangle \\
&= \frac{1}{p} \sum_{l=1}^{p} \|\mathbf{y} - \hat{\mathbf{u}}_{k,\tau}^l\|_2^2 - \frac{1}{p} \sum_{l=1}^{p} \|\mathbf{y} - \hat{\mathbf{u}}_{k,\tau}^l\|_2^2 + \frac{1}{p^2} \sum_{l=1}^{p} \sum_{l'=1}^{p} \left\langle \mathbf{y} - \hat{\mathbf{u}}_{k,\tau}^l, \mathbf{y} - \hat{\mathbf{u}}_{k,\tau}^{l'} \right\rangle \\
&= \frac{1}{p} \sum_{l=1}^{p} \|\mathbf{y} - \hat{\mathbf{u}}_{k,\tau}^l\|_2^2 - \\
&\qquad \frac{1}{2p^2} \left( \sum_{l=1}^{p} \sum_{l'=1}^{p} \left( \|\mathbf{y} - \hat{\mathbf{u}}_{k,\tau}^l\|_2^2 + \|\mathbf{y} - \hat{\mathbf{u}}_{k,\tau}^{l'}\|_2^2 \right) - \sum_{l=1}^{p} \sum_{l'=1}^{p} \left\langle \mathbf{y} - \hat{\mathbf{u}}_{k,\tau}^l, \mathbf{y} - \hat{\mathbf{u}}_{k,\tau}^{l'} \right\rangle \right) \\
&= \frac{1}{p} \sum_{l=1}^{p} \|\mathbf{y} - \hat{\mathbf{u}}_{k,\tau}^l\|_2^2 - \frac{1}{p^2} \sum_{l=1}^{p} \sum_{l'=1}^{l-1} \|\hat{\mathbf{u}}_{k,\tau}^l - \hat{\mathbf{u}}_{k,\tau}^{l'}\|_2^2
\end{aligned}
$$

$\square$

**Lemma 14.** *We have*

$$\frac{1}{p} \sum_{l=1}^{p} \sum_{l'=1}^{l-1} \|\hat{\mathbf{u}}_{k,\tau}^l - \hat{\mathbf{u}}_{k,\tau}^{l'}\|_2^2 = \sum_{l=1}^{p} \|\mathbf{u}_{k,\tau} - \hat{\mathbf{u}}_{k,\tau}^l\|_2^2$$

*Proof.* Using $\mathbf{u}_k = \frac{1}{p} \sum_{l=1}^{p} \hat{\mathbf{u}}_k^l$ we have

$$
\begin{aligned}
\sum_{l=1}^{p} \|\mathbf{u}_{k,\tau} - \hat{\mathbf{u}}_{k,\tau}^l\|_2^2 &= \sum_{l=1}^{p} \left\| \frac{1}{p} \sum_{l'=1}^{p} \hat{\mathbf{u}}_{k,\tau}^l - \hat{\mathbf{u}}_{k,\tau}^l \right\|_2^2 \\
&= \frac{1}{p^2} \sum_{l=1}^{p} \left\| \sum_{l'=1}^{p} \left( \hat{\mathbf{u}}_{k,\tau}^{l'} - \hat{\mathbf{u}}_{k,\tau}^l \right) \right\|_2^2 \\
&= \frac{1}{p^2} \sum_{l=1}^{p} \sum_{l_1=1}^{p} \sum_{l_2=1}^{p} \left\langle \hat{\mathbf{u}}_{k,\tau}^{l_1} - \hat{\mathbf{u}}_{k,\tau}^l, \hat{\mathbf{u}}_{k,\tau}^{l_2} - \hat{\mathbf{u}}_{k,\tau}^l \right\rangle \\
&= \sum_{l=1}^{p} \|\hat{\mathbf{u}}_{k,\tau}^l\|_2^2 - \frac{1}{p} \sum_{l=1}^{p} \sum_{l'=1}^{p} \left\langle \hat{\mathbf{u}}_{k,\tau} k^l, \hat{\mathbf{u}}_k^{l'} \right\rangle \\
&= \frac{1}{2p} \left( \sum_{l=1}^{p} \sum_{l'=1}^{p} \left( \|\hat{\mathbf{u}}_{k,\tau}^l\|_2^2 + \|\hat{\mathbf{u}}_{k,\tau}^{l'}\|_2^2 \right) - \sum_{l=1}^{p} \sum_{l'=1}^{p} 2 \left\langle \hat{\mathbf{u}}_{k,\tau}^l, \hat{\mathbf{u}}_{k,\tau}^{l'} \right\rangle \right) \\
&= \frac{1}{p} \sum_{l=1}^{p} \sum_{l'=1}^{l-1} \|\hat{\mathbf{u}}_{k,\tau}^l - \hat{\mathbf{u}}_{k,\tau}^{l'}\|_2^2
\end{aligned}
$$

$\square$

**Lemma 15.** *Suppose the condition of lemma 25 holds and the step size satisfies* $\eta = O\left(\frac{\lambda_0 p}{n^2(1-p^{-1})^{\frac{2}{3}}\tau}\right)$, *then with probability at least* $1 - \delta$ *it holds for all* $k \in [K]$ *that*

$$\iota_k \leq \frac{(1-p^{-1})^{\frac{1}{3}}\eta\lambda_0}{2p}\sum_{l=1}^{p}\sum_{t=0}^{\tau-1}\mathbb{E}_{\mathbf{M}_k}\left[\left\|\mathbf{y}-\hat{\mathbf{u}}_{k,t}^l\right\|_2^2\right] + 24\left(1-p^{-1}\right)^{\frac{2}{3}}n\kappa^2\eta\tau\frac{\lambda_{\max}}{\lambda_0}$$

*Proof.* Recall the definition of $\iota_k$, expanding the quadratic form gives

$$\iota_k = \frac{1}{p}\sum_{l=1}^{p}\mathbb{E}_{\mathbf{M}_k}\left[\left\|\mathbf{u}_k-\hat{\mathbf{u}}_k^l\right\|_2^2 - \left\|\mathbf{u}_{k,\tau}-\hat{\mathbf{u}}_{k,\tau}^l\right\|_2^2\right]$$

$$= \frac{1}{p}\sum_{l=1}^{p}\mathbb{E}_{\mathbf{M}_k}\left[\left\|\mathbf{u}_k-\hat{\mathbf{u}}_k^l\right\|_2^2 - \left\|\left(\mathbf{u}_k-\hat{\mathbf{u}}_k^l\right) + \left(\mathbf{u}_k-\mathbf{u}_{k+1}-\hat{\mathbf{u}}_k^l+\hat{\mathbf{u}}_{k,\tau}^l\right)\right\|_2^2\right]$$

$$= \frac{1}{p}\sum_{l=1}^{p}\mathbb{E}_{\mathbf{M}_k}\left[2\left\langle\mathbf{u}_k-\hat{\mathbf{u}}_k^l,\mathbf{u}_k-\mathbf{u}_{k+1}-\hat{\mathbf{u}}_k^l+\hat{\mathbf{u}}_{k,\tau}^l\right\rangle + \left\|\mathbf{u}_k-\mathbf{u}_{k+1}-\hat{\mathbf{u}}_k^l+\hat{\mathbf{u}}_{k,\tau}^l\right\|_2^2\right]$$

$$= \frac{1}{p}\sum_{l=1}^{p}\mathbb{E}_{\mathbf{M}_k}\left[2\left\langle\mathbf{u}_k-\hat{\mathbf{u}}_k^l,\hat{\mathbf{u}}_{k,\tau}^l-\hat{\mathbf{u}}_k^l\right\rangle\right] + \frac{1}{p}\sum_{l=1}^{p}\mathbb{E}_{\mathbf{M}_k}\left[\left\|\mathbf{u}_k-\mathbf{u}_{k+1}-\hat{\mathbf{u}}_k^l+\hat{\mathbf{u}}_{k,\tau}^l\right\|_2^2\right]$$

where in the third inequality, we use the fact that

$$\sum_{l=1}^{p}\left\langle\mathbf{u}_k-\hat{\mathbf{u}}_k^l,\mathbf{u}_k-\mathbf{u}_{k+1}\right\rangle = \left\langle p\mathbf{u}_k-\sum_{l=1}^{p}\hat{\mathbf{u}}_k^l,\mathbf{u}_k-\mathbf{u}_{k+1}\right\rangle = 0$$

For convenience, we denote

$$\sigma_{k,r}^{(i)} = \sigma\left(\langle\mathbf{W}_{k,r},\mathbf{x}_i\rangle\right); \quad \sigma_{k,\tau,r}^{l(i)} = \sigma\left(\langle\mathbf{w}_{k,\tau,r}^l,\mathbf{x}_i\rangle\right)$$

Noticing that $\frac{1}{p}\sum_{l=1}^{p}\left(\hat{\mathbf{u}}_k^l-\hat{\mathbf{u}}_{k,\tau}^l\right) = \mathbf{u}_k-\mathbf{u}_{k+1}$, we apply a trick similar to lemma 14 to get that

$$\frac{1}{p}\sum_{l=1}^{p}\mathbb{E}_{\mathbf{M}_k}\left[\left\|\mathbf{u}_k-\mathbf{u}_{k+1}-\hat{\mathbf{u}}_k^l+\hat{\mathbf{u}}_{k,\tau}^l\right\|_2^2\right] = \frac{1}{p^2}\sum_{l=1}^{p}\sum_{l'=1}^{l-1}\mathbb{E}_{\mathbf{M}_k}\left[\left\|\hat{\mathbf{u}}_k^l-\hat{\mathbf{u}}_{k,\tau}^l-\hat{\mathbf{u}}_k^l-\hat{\mathbf{u}}_{k,\tau}^l\right\|_2^2\right]$$

$$= \frac{1}{mp^2}\sum_{l=1}^{p}\sum_{l'=1}^{l-1}\sum_{i=1}^{n}\left(\sum_{r=1}^{m}a_r m_{k,r}^l\left(\sigma_{k,r}^{(i)}-\sigma_{k,\tau,r}^{l(i)}\right) - m_{k,r}^{l'}\left(\sigma_{k,r}^{(i)}-\sigma_{k,\tau,r}^{l'(i)}\right)\right)_2^2$$

$$\leq \frac{1}{mp^2}\sum_{l=1}^{p}\sum_{l'=1}^{l-1}\sum_{i=1}^{n}\sum_{r,r'=1}^{m}m_{k,r}^l\left(\sigma_{k,r}^{(i)}-\sigma_{k,\tau,r}^{l(i)}\right)^2 + m_{k,r}^{l'}\left(\sigma_{k,r}^{(i)}-\sigma_{k,\tau,r}^{l'(i)}\right)^2$$

$$\leq \frac{\eta^2\tau n^2}{mp^2}\sum_{l=1}^{p}\sum_{l'=1}^{l-1}\sum_{r=1}^{m}\sum_{t=0}^{\tau-1}\left(m_{k,r}^l\left\|\mathbf{y}-\hat{\mathbf{u}}_{k,t}^l\right\|_2^2 + m_{k,r}^{l'}\left\|\mathbf{y}-\hat{\mathbf{u}}_{k,t}^{l'}\right\|_2^2\right)$$

$$\leq \frac{\eta^2\tau n^2(p-1)}{2mp}\sum_{l=1}^{p}\sum_{r=1}^{m}\sum_{t=0}^{\tau-1}m_{k,r}^l\left\|\mathbf{y}-\hat{\mathbf{u}}_{k,t}^l\right\|_2^2$$

Notice that fixing an $l \in [p]$, we have that $m_{k,r}^l$'s are independent for all $r \in [m]$. Apply Hoeffding's inequality to get that

$$\mathbb{P}\left(\sum_{r=1}^{m}m_{k,r}^l \geq 2mp^{-1}\right) \leq \exp\left(-2mp^{-2}\right)$$

Apply the union bound over all $k \in [K]$ and $l \in [p]$, and apply the over-parameterization requirement to get that, with probability at least $1 - \delta$, it holds that

$$\frac{1}{p}\sum_{l=1}^{p} \mathbb{E}_{\mathbf{M}_k} \left[ \left\| \mathbf{u}_k - \mathbf{u}_{k+1} - \hat{\mathbf{u}}_k^l + \hat{\mathbf{u}}_{k,\tau}^l \right\|_2^2 \right] \leq \frac{\eta^2 \tau n^2 (p-1)}{p^2} \sum_{l=1}^{p}\sum_{t=0}^{\tau-1} \left\| \mathbf{y} - \hat{\mathbf{u}}_{k,t}^l \right\|_2^2$$

$$\leq \frac{\left(1 - p^{-1}\right)^{\frac{1}{3}} \eta \lambda_0}{4p} \sum_{l=1}^{p}\sum_{t=0}^{\tau-1} \left\| \mathbf{y} - \hat{\mathbf{u}}_{k,t}^l \right\|_2^2$$

by choosing $\eta = O\left( \frac{\lambda_0 p}{n^2 (1-p^{-1})^{\frac{2}{3}} \tau} \right)$ The first term can be bounded by

$$\Delta_1 = \left\langle \mathbf{u}_k - \hat{\mathbf{u}}_k^l, \hat{\mathbf{u}}_{k,\tau}^l - \hat{\mathbf{u}}_k^l \right\rangle \leq \left\| \mathbf{u}_k - \hat{\mathbf{u}}_k^l \right\| \left\| \hat{\mathbf{u}}_{k,\tau}^l - \hat{\mathbf{u}}_k^l \right\|$$

We study the term $\left\| \hat{\mathbf{u}}_{k,t+1}^l - \hat{\mathbf{u}}_{k,t}^l \right\|_2$. Following analysis in the proof of hypothesis 1, we write $\hat{\mathbf{u}}_{k,t+1}^l - \hat{\mathbf{u}}_{k,t}^l = \mathbf{I}_{1,k,t}^l + \mathbf{I}_{2,k,t}^l$. We first study the magnitude of $\mathbf{I}_{1,k,t}^l$. Its $i$th entry has

$$I_{1,k,t}^{l(i)} = \eta \sum_{j=1}^{n} \left( \mathbf{m}_k^l \circ \mathbf{H}(k,t) - \mathbf{m}_k^l \circ \mathbf{H}(k,t)^{\perp} \right)_{ij} \left( y_i - \hat{u}_{k,t}^{l(i)} \right)$$

As we have shown in the proof of theorem 2

$$\left\| \mathbf{m}_k^l \circ \mathbf{H}(k,t) - \mathbf{H}^{\infty} \right\| \leq \frac{\lambda_0}{2}$$

Therefore, $\lambda_{\max}\left( \mathbf{m}_k^l \circ \mathbf{H}(k,t) \right) \leq \lambda_{\max} + \frac{\lambda_0}{2} \leq 2\lambda_{\max}$. Moreover, in the proof of hypothesis 1 we have shown that

$$\left\| \mathbf{m}_k^l \circ \mathbf{H}(k,t)^{\perp} \right\|_2 \leq 4n\kappa^{-1}R$$

Therefore, we have

$$\left\| \mathbf{I}_{1,k,t}^l \right\|_2 = \eta \left( \mathbf{m}_k^l \circ \mathbf{H}(k,t) - \mathbf{m}_k^l \circ \mathbf{H}(k,t)^{\perp} \right) \left( \mathbf{y} - \hat{\mathbf{u}}_{k,t}^l \right)$$

$$\leq \left( 2\eta\lambda_{\max} + 4\eta\kappa^{-1}nR \right) \left\| \mathbf{y} - \hat{\mathbf{u}}_{k,t}^l \right\|_2$$

Using the bound of $\left| I_{2,k,t}^{l(i)} \right|$ in the proof of hypothesis 1, we have that

$$\left\| \mathbf{I}_{2,k,t}^l \right\|_2 \leq \left( \sum_{i=1}^{n} \left| I_{2,k,t}^{l(i)} \right|^2 \right)^{\frac{1}{2}} 4\eta\kappa^{-1}nR \left\| \mathbf{y} - \hat{\mathbf{u}}_{k,t}^l \right\|_2$$

Therefore,

$$\left\| \hat{\mathbf{u}}_{k,t+1}^l - \hat{\mathbf{u}}_{k,t}^l \right\|_2 \leq \left\| \mathbf{I}_{1,k,t}^l \right\|_2 + \left\| \mathbf{I}_{2,k,t}^l \right\|_2 \leq \left( 2\eta\lambda_{\max} + 8\eta\kappa^{-1}nR \right) \left\| \mathbf{y} - \hat{\mathbf{u}}_{k,t}^l \right\|_2 \leq 3\eta\lambda_{\max} \left\| \mathbf{y} - \hat{\mathbf{u}}_{k,t}^l \right\|_2$$

by our choice of $R$. Thus, we have that

$$\Delta_1 \leq 3\eta\lambda_{\max} \sum_{t=0}^{\tau-1} \left\| \mathbf{y} - \hat{\mathbf{u}}_{k,t}^l \right\|_2 \left\| \mathbf{u}_k - \hat{\mathbf{u}}_k^l \right\|_2 \leq \frac{\left(1 - p^{-1}\right)^{\frac{1}{3}} \eta \lambda_0}{8} \sum_{t=0}^{\tau-1} \left\| \mathbf{y} - \hat{\mathbf{u}}_{k,t}^l \right\|_2^2 + 12 \frac{\eta\tau\lambda_{\max}}{(1-p^{-1})^{\frac{1}{3}} \lambda_0} \left\| \mathbf{u}_k - \hat{\mathbf{u}}_k^l \right\|_2^2$$

Therefore, by applying lemma 25 we have that

$$\iota_k \leq \frac{\left(1 - p^{-1}\right)^{\frac{1}{3}} \eta \lambda_0}{2p} \sum_{l=1}^{p}\sum_{t=0}^{\tau-1} \mathbb{E}_{\mathbf{M}_k} \left[ \left\| \mathbf{y} - \hat{\mathbf{u}}_{k,t}^l \right\|_2^2 \right] + 24 \left( 1 - p^{-1} \right)^{\frac{2}{3}} n\kappa^2\eta\tau \frac{\lambda_{\max}}{\lambda_0}$$

$\square$

# I  Auxiliary Results

**Lemma 16.** *With probability at least $1 - ne^{-m\kappa^{-1}R}$ we have $|S_i| \le 4m\kappa^{-1}R$ for all $i \in [n]$.*

*Proof.* Note that $\mathbb{I}\{r \in S_i^{\perp}\} = \mathbb{I}\{\mathbb{I}\{A_{ir}\} \ne 0\} = \mathbb{I}\{A_{ir}\}$. Therefore, we have

$$|S_i^{\perp}| = \sum_{r=1}^{m} \mathbb{I}\{r \in S_i^{\perp}\} = \mathbb{I}\{A_{ir}\}.$$

Since $\mathbb{E}_{\mathbf{w}_{0,r}}[\mathbb{I}\{A_{ir}\}] = P(A_{ir}) \le \frac{2R}{\kappa\sqrt{2\pi}} \le \kappa^{-1}R$, we also have

$$\mathbb{E}_{\mathbf{w}_{0,r}}\left[\left(\mathbb{I}\{A_{ir}\} - \mathbb{E}_{\mathbf{w}_{0,r}}[\mathbb{I}\{A_{ir}\}]\right)^2\right] \le \mathbb{E}_{\mathbf{w}_{0,r}}\left[\mathbb{I}\{A_{ir}\}^2\right] = \frac{2R}{\kappa\sqrt{2\pi}} \le \kappa^{-1}R$$

Again apply Bernstein inequality over the random variable $\mathbb{I}\{A_{ir}\} - \mathbb{E}_{\mathbf{w}_{0,r}}[\mathbb{I}\{A_{ir}\}]$ with $t = 3m\kappa^{-1}R$ gives

$$P\left(|S_i^{\perp}| \le 4m\kappa^{-1}R\right) = P\left(\sum_{r=1}^{m} \mathbb{I}\{A_{ir}\} \ge 4m\kappa^{-1}R\right) \le \exp\left(-m\kappa^{-1}R\right)$$

$\square$

**Lemma 17.** *Define $\mathbf{H}^{\perp} \in \mathbb{R}^{n \times n}$ such that*

$$\mathbf{H}_{ij}^{\perp} = \frac{\xi}{m}\langle \mathbf{x}_i, \mathbf{x}_j\rangle \sum_{r \in S_i^{\perp}} \mathbb{I}\{\langle \mathbf{w}_r, \mathbf{x}_i\rangle \ge 0; \langle \mathbf{w}_r, \mathbf{x}_i\rangle \ge 0\}$$

*If $|S_i^{\perp}| \le 4m\kappa^{-1}R$, then we have*

$$\|\mathbf{H}^{\perp}\|_2 \le 4n\xi\kappa^{-1}R$$

*Proof.* We note that

$$\|\mathbf{H}^{\perp}\|_2^2 \le \|\mathbf{H}^{\perp}\|_F^2 = \sum_{i,j=1}^{n} |\mathbf{H}_{ij}^{\perp}|^2$$

For each $i, j$ pair we have

$$|\mathbf{H}_{ij}^{\perp}| \le \frac{\xi}{m}|S_i^{\perp}| = 4\xi\kappa^{-1}R$$

Thus

$$\|\mathbf{H}^{\perp}\|_2 \le \left(\|\mathbf{H}^{\perp}\|_F^2\right)^{-\frac{1}{2}} \le \left(16n^2\kappa^{-2}\Delta^2\right)^{-\frac{1}{2}} = 4n\xi\kappa^{-1}R$$

$\square$

**Lemma 18.** *For i.i.d Bernoulli masks with parameter $\xi$, $N_{k,r}^{\perp} \sim \mathit{Bern}(\theta)$ with*

$$\theta = P(N_{k,r}^{\perp} = 1) = 1 - (1 - \xi)^p$$

*Proof.* We have

$$P(N_{k,r}^{\perp} = 1) = 1 - P(N_{k,r}^{\perp} = 0) = 1 - \prod_{l=1}^{p} P(m_{k,r}^l = 0) = 1 - (1 - \xi)^p$$

$\square$

**Lemma 19.** *We have*

$$\mathbb{E}_{\mathbf{M}_k}\left[(\nu_{k,r,r} - \xi)^2\right] \leq \theta - \xi^2$$

*Proof.* To start, we notice that $\nu_{k,r,r} = \eta_{k,r}\sum_{l=1}^{p} m_{k,r}^{l2} = \eta_{k,r}\sum_{l=1}^{p} m_{k,r}^{l} = N_{k,r}^{\perp}$. Therefore $\mathbb{E}_{\mathbf{M}_k}[\nu_{k,r,r}] = \mathbb{E}_{\mathbf{M}_k}\left[N_{k,r}^{\perp}\right] = \theta$. Moreover, since $N_{k,r}^{\perp 2} = N_{k,r}^{\perp}$, we have $\mathbb{E}_{\mathbf{M}_k}\left[\nu_{k,r,r}^2\right] = \theta$. Thus, using $\theta \geq \xi$, we have

$$\mathbb{E}_{\mathbf{M}_k}\left[(\nu_{k,r,r} - \xi)^2\right] = \mathbb{E}_{\mathbf{M}_k}\left[\nu_{k,r,r}^2\right] - 2\xi\mathbb{E}_{\mathbf{M}_k}\left[\nu_{k,r,r}\right] + \xi^2$$
$$= \theta - 2\xi\theta + \xi^2 \leq \theta - \xi^2$$

$\square$

**Lemma 20.** *For i.i.d Bernoulli masks with parameter $\xi$, we have*

$$\mathbb{E}_{\mathbf{M}_k}\left[\nu_{k,r,r'} \mid N_{k,r}^{\perp} = 1\right] = \begin{cases} \xi & \text{if } r \neq r' \\ 1 & \text{if } r = r' \end{cases}$$

*Proof.* If $r = r'$, we have

$$\mathbb{E}_{\mathbf{M}_k}\left[\nu_{k,r,r'} \mid N_{k,r}^{\perp} = 1\right] = \mathbb{E}_{\mathbf{M}_k}\left[\eta_{k,r}\sum_{l=1}^{p} m_{k,r}^{l} \mid N_{k,r}^{\perp} = 1\right] = \mathbb{E}_{\mathbf{M}_k}\left[\frac{X_{k,r}}{N_{k,r}} \mid N_{k,r}^{\perp} = 1\right]$$
$$= \mathbb{E}_{\mathbf{M}_k}\left[N_{k,r}^{\perp} \mid N_{k,r}^{\perp} = 1\right] = 1$$

If $r' \neq r$, then we have that $m_{k,r'}^{l}$ is independent from $m_{k,r}^{l}$ and $N_{k,r}$. Therefore,

$$\mathbb{E}_{\mathbf{M}_k}\left[\nu_{k,r,r'} \mid N_{k,r}^{\perp} = 1\right] = \mathbb{E}_{\mathbf{M}_k}\left[\eta_{k,r}\sum_{l=1}^{p} m_{k,r}^{l} \mid N_{k,r}^{\perp} = 1\right]\mathbb{E}_{\mathbf{M}_k}\left[m_{k,r'}^{l}\right]$$
$$= \xi\mathbb{E}_{\mathbf{M}_k}\left[\frac{X_{k,r}}{N_{k,r}} \mid N_{k,r}^{\perp} = 1\right] = \xi$$

$\square$

**Lemma 21.** *The variance follows*

$$Var_{\mathbf{M}_k}\left(\nu_{k,r,r'} \mid N_{k,r}^{\perp} = 1\right) = \begin{cases} \frac{\theta - \xi^2}{p} & \text{if } r \neq r' \\ 0 & \text{if } r = r' \end{cases}$$

*Proof.* For $r \neq r'$, the expectation of $\nu_{k,r,r'}^2$ given $N_{k,r}^{\perp} = 1$ is

$$\mathbb{E}_{\mathbf{M}_k}\left[\nu_{k,r,r'}^2 \mid N_{k,r}^{\perp} = 1\right] = \mathbb{E}_{\mathbf{M}_k}\left[\frac{\sum_{l=1}^{p}\sum_{l'=1}^{p} m_{k,r}^{l}m_{k,r}^{l'}m_{k,r'}^{l}m_{k,r'}^{l'}}{X_{k,r}} \mid N_{k,r}^{\perp} = 1\right]$$

$$= \sum_{l=1}^{p}\sum_{l'\neq l}\mathbb{E}_{\mathbf{M}_k}[m_{k,r'}^{l}]\mathbb{E}_{\mathbf{M}_k}[m_{k,r'}^{l'}]\mathbb{E}_{\mathbf{M}_k}\left[\frac{m_{k,r}^{l}}{X_{k,r}} \mid N_{k,r}^{\perp} = 1\right]\mathbb{E}_{\mathbf{M}_k}\left[\frac{m_{k,r}^{l'}}{X_{k,r}} \mid N_{k,r}^{\perp} = 1\right] +$$

$$\sum_{l=1}^{p}\mathbb{E}_{\mathbf{M}_k}[m_{k,r'}^{l}]\mathbb{E}_{\mathbf{M}_k}\left[\frac{m_{k,r}^{l}}{X_{k,r}^2} \mid N_{k,r}^{\perp} = 1\right]$$

$$= \xi^2\sum_{l=1}^{p}\sum_{l'\neq l}\mathbb{E}_{\mathbf{M}_k}\left[\frac{m_{k,r}^{l}}{X_{k,r}} \mid N_{k,r}^{\perp} = 1\right]\mathbb{E}_{\mathbf{M}_k}\left[\frac{m_{k,r}^{l'}}{X_{k,r}} \mid N_{k,r}^{\perp} = 1\right] +$$

$$\xi\sum_{l=1}^{p}\mathbb{E}_{\mathbf{M}_k}\left[\frac{m_{k,r}^{l}}{X_{k,r}^2} \mid N_{k,r}^{\perp} = 1\right]$$

$$= \xi^2 + \xi\mathbb{E}_{\mathbf{M}_k}\left[\frac{1}{X_{k,r}} \mid N_{k,r}^{\perp} = 1\right] - \xi^2\sum_{l=1}^{p}\mathbb{E}_{\mathbf{M}_k}\left[\frac{m_{k,r}^{l}}{X_{k,r}} \mid N_{k,r}^{\perp} = 1\right]^2$$

Therefore, the variance of $\nu_{k,r,r'}$ given $N_{k,r}^\perp = 1$ has the form

$$
\mathrm{Var}\left(\nu_{k,r,r'} \mid gN_{k,r}^\perp = 1\right) = \mathbb{E}_{\mathbf{M}_k}\left[\nu_{k,r,r'}^2 \mid N_{k,r}^\perp = 1\right] - \mathbb{E}_{\mathbf{M}_k}\left[\nu_{k,r,r'} \mid N_{k,r}^\perp = 1\right]^2
$$

$$
= \xi\mathbb{E}_{\mathbf{M}_k}\left[\frac{1}{X_{k,r}} \mid N_{k,r}^\perp = 1\right] - \xi^2\sum_{l=1}^{p}\mathbb{E}_{\mathbf{M}_k}\left[\frac{m_{k,r}^l}{X_{k,r}} \mid N_{k,r}^\perp = 1\right]^2
$$

Let $X(p) = \sum_{l=1}^{p} m_{.}^l \sim \mathcal{B}(p, \xi)$, then we have

$$
\mathbb{E}_{\mathbf{M}_k}\left[\frac{1}{X_{k,r}} \mid N_{k,r}^\perp = 1\right] = \mathbb{E}_{\mathbf{M}_k}\left[\frac{1}{1 + X(p-1)}\right]
$$

$$
\mathbb{E}_{\mathbf{M}_k}\left[\frac{m_{k,r}^l}{X_r} \mid N_{k,r}^\perp = 1\right] = P(m_{k,r}^l = 1 \mid N_{k,r}^\perp = 1)\mathbb{E}_{\mathbf{M}_k}\left[\frac{1}{1 + X(p-1)}\right] = \frac{\xi}{\theta}\mathbb{E}_{\mathbf{M}_k}\left[\frac{1}{1 + X(p-1)}\right]
$$

Moreover, using reciprocal moments we have

$$
\mathbb{E}_{\mathbf{M}_k}\left[\frac{1}{1 + X(p-1)}\right] = \frac{\theta}{p\xi}
$$

Therefore

$$
\mathrm{Var}_{\mathbf{M}_k}\left(\nu_{k,r,r'} \mid N_{k,r}^\perp = 1\right) = \xi\mathbb{E}_{\mathbf{M}_k}\left[\frac{1}{1 + X(p-1)}\right] - \frac{\xi^4}{\theta^2}p\mathbb{E}_{\mathbf{M}_k}\left[\frac{1}{1 + X(p-1)}\right]^2
$$

$$
= \frac{\theta - \xi^2}{p}
$$

If $r = r'$, the variance is

$$
\mathrm{Var}_{\mathbf{M}_k}\left(\nu_{r,r} \mid g_r = 1\right) = \mathrm{Var}_{\mathbf{M}_k}\left(g_r \mid g_r = 1\right) = 0
$$

$\square$

**Lemma 22.** *Suppose $\kappa \leq 1, R \leq \kappa\sqrt{\frac{d}{32}}$. With probability at least $1 - e^{md/32}$ we have that*

$$
\|W_0\|_F \leq \kappa\sqrt{2md} - \sqrt{m}R
$$

*Proof.* For all $r \in [m], d_1 \in [d]$, we have $\mathbb{E}_{\mathbf{M}_k}\left[w_{rd_1}^2\right] = \kappa^2$. Moreover, each $w_{rd_1}^2$ is a $(2\kappa^2, 2\kappa^2)$-sub-exponential random variable

$$
\mathbb{E}\left[e^{t(w_{rd_1}^2 - \kappa^2)}\right] = \frac{1}{\kappa\sqrt{2\pi}}\int_{-\infty}^{\infty} e^{t(w_{rd_1}^2 - \kappa^2)}e^{-\frac{w_{rd_1}^2}{2\kappa^2}}dw_{rd_1}
$$

$$
= \frac{1}{\kappa\sqrt{2\pi}}\int_{-\infty}^{\infty} e^{-(\frac{1}{2\kappa^2} - t)w_{rd_1}^2 - t\kappa^2}dw_{rd_1}
$$

$$
= \frac{1}{\kappa\sqrt{2\pi}} \cdot \sqrt{\frac{\pi}{(2\kappa)^{-1} - t}} \cdot e^{-t\kappa^2}
$$

$$
= \frac{e^{-t\kappa^2}}{\sqrt{1 - 2t\kappa^2}} \leq e^{2t^2\kappa^4}
$$

with $t \leq \frac{1}{2\kappa^2}$. Thus, using independence between entries of $\mathbf{W}_0$ gives

$$
\mathbb{E}\left[e^{t(\|\mathbf{W}_0\|_F^2 - md\kappa^2)}\right] \leq \prod_{r=1}^{m}\prod_{d_1=1}^{d}\mathbb{E}\left[e^{t(w_{rd_1}^2 - \kappa^2)}\right] \leq e^{2mdt^2\kappa^4}
$$

Invoking the tail bound of sub-exponential random variable gives

$$P\left(\|\mathbf{W}_0\|_F^2 \geq md\kappa^2 + t\right) \leq \begin{cases} e^{-\frac{t^2}{8md\kappa^4}} & \text{if } 0 \leq t \leq 2md\kappa^2 \\ e^{-\frac{t^2}{4\kappa^2}} & \text{if } t > 2md\kappa^2 \end{cases}$$

Let $t = md\kappa^2 - 2m\kappa R\sqrt{2d} + mR^2$. Then

$$\|\mathbf{W}_0\|_F^2 \leq 2md\kappa^2 + mR^2 - 2m\kappa R\sqrt{2d} = (\kappa\sqrt{2md} - \sqrt{m}R)^2$$

with probability at least $1 - e^{-\frac{t^2}{8md\kappa^4}}$. Using $R \leq \kappa\sqrt{\frac{d}{32}}$ we have $t \geq \frac{1}{2}md\kappa^2$. Thus with probability at least $1 - e^{-\frac{md}{32}}$ we have

$$\|\mathbf{W}_0\|_F \leq \kappa\sqrt{2md} - \sqrt{m}R$$

$\square$

**Lemma 23.** *Assume $\kappa \leq 1$ and $R \leq \frac{\kappa}{\sqrt{2}}$. With probability at least $1 - ne^{-\frac{m}{32}}$ over initialization, it holds for all $i \in [n]$ that*

$$\sum_{r=1}^m \langle \mathbf{w}_{0,r}, \mathbf{x}_i \rangle^2 \leq 2m\kappa^2 - mR^2$$

$$\sum_{i=1}^n \sum_{r=1}^m \langle \mathbf{w}_{0,r}, \mathbf{x}_i \rangle^2 \leq 2mn\kappa^2 - mnR^2$$

*Proof.* It suffice to prove the first inequality, and the second follows by summing over $n$. To begin, we show that each $\langle \mathbf{w}_0, \mathbf{x}_i \rangle$ are Gaussian with zero mean and variance $\kappa^2$. Using independence between entries of $\mathbf{w}_{0,r}$, we have

$$\mathbb{E}\left[e^{-t\langle \mathbf{w}_{0,r}, \mathbf{x}_i \rangle}\right] = \mathbb{E}\left[\prod_{j=1}^d e^{-tw_{0,r,j}x_{i,j}}\right] = \prod_{j=1}^d \mathbb{E}\left[e^{-tw_{0,r,j}x_{i,j}}\right]$$

$$= \prod_{j=1}^d e^{-t^2 x_{i,j}^2 \kappa^2} = e^{-t^2\kappa^2 \sum_{j=1}^d x_{i,j}^2} = e^{-t^2\kappa^2}$$

where the last equality follows from our assumption that $\|\mathbf{x}_i\|_2 = 1$. Next, we treat each $\omega_{r,i} = \langle \mathbf{w}_{0,r}, \mathbf{x}_i \rangle^2$ as a random variable. First, we compute the mean of $\omega_{r,i}$

$$\mathbb{E}[\omega_{r,i}] = \mathbb{E}_{\mathbf{W}_0}\left[\langle \mathbf{w}_{0,r}, \mathbf{x}_i \rangle^2\right] = \mathbb{E}_{\mathbf{W}_0}\left[\left(\sum_{d_1=1}^d w_{0,r,d}x_{i,d}\right)^2\right]$$

$$= \sum_{d_1=1}^d \mathbb{E}_{\mathbf{W}_0}\left[w_{0,r,d}^2\right] x_{i,d}^2 = \kappa^2 \sum_{d_1=1}^d x_{i,d}^2 = \kappa^2$$

Then, we show that each $\omega_{r,i}$ is sub-exponential with parameter $(2\kappa^2, 2\kappa^2)$.

$$\mathbb{E}\left[e^{t(\omega_{r,i}-\kappa^2)}\right] = \frac{1}{\kappa\sqrt{2\pi}} \int_{-\infty}^\infty e^{t(\omega_{r,i}-\kappa^2)} e^{-\frac{\omega_{r,i}}{2\kappa^2}} d\sqrt{\omega_{r,i}}$$

$$= \frac{1}{\kappa\sqrt{2\pi}} \int_{-\infty}^\infty e^{-(\frac{1}{2\kappa^2}-t)(\sqrt{\omega_{r,i}})^2 - t\kappa^2} d\sqrt{\omega_{r,i}}$$

$$= \frac{1}{\kappa\sqrt{2\pi}} \cdot \sqrt{\frac{\pi}{(2\kappa^2)^{-1} - t}} \cdot e^{-t\kappa^2}$$

$$= \frac{e^{-t\kappa^2}}{1 - 2t\kappa^2} \leq e^{2t\kappa^4}$$

for $t \le \frac{1}{2\kappa^2}$. Since each $\mathbf{w}_{0,r}$ is independent, we have that each $\omega_{r,i}$ is independent for a fixed $i$. Thus

$$\mathbb{E}\left[e^{t\sum_{r=1}^m (\omega_{r,i}-\kappa^2)}\right] = \prod_{r=1}^m \mathbb{E}\left[e^{t(\omega_{r,i}-\kappa^2)}\right] \le e^{2mt\kappa^4}$$

Thus we have

$$P\left(\sum_{r=1}^m \omega_{r,i} \ge m\kappa^2 + t\right) \le \begin{cases} e^{-\frac{t^2}{8m\kappa^4}} & \text{if } 0 \le t \le 2m\kappa^2 \\ e^{-\frac{t^2}{2\kappa^2}} & \text{if } t \ge 2m\kappa^2 \end{cases}$$

We choose $t = m\kappa^2 - mR^2$. Since $R \le \frac{\kappa}{\sqrt{2}}$, we have that $\frac{m\kappa^2}{2} \le t \le m\kappa^2$. Thus

$$P\left(\sum_{r=1}^m \omega_{r,i} \ge 2m\kappa^2 - mR^2\right) \le e^{-\frac{m}{8}}$$

Apply a union bound over all $i \in [n]$ gives that with probability at least $1 - ne^{-\frac{m}{32}}$, it holds for all $i \in [n]$ that

$$\sum_{r=1}^m \omega_{r,i} \le 2m\kappa^2 - mR^2$$

□

**Lemma 24.** *If for some $R > 0$ and all $r \in [m]$ the initialization satisfies*

$$\sum_{r=1}^m \langle \mathbf{w}_{0,r}, \mathbf{x}_i \rangle^2 \le 2mn\kappa^2 - mnR^2$$

*and for all $r \in [m]$, it holds that $\|\mathbf{w}_{k,r} - \mathbf{w}_{0,r}\|_2 \le R$. Then with $C_1 = \frac{4\theta^2(1-\xi)n\kappa^2}{p}$, we have*

$$\mathbb{E}_{\mathbf{M}_k}\left[\eta_{k,r}^2 \sum_{l=1}^p m_{k,r}^l \|\mathbf{u}_k - \hat{\mathbf{u}}_k^l\|_2^2\right] \le C_1; \quad \mathbb{E}_{\mathbf{M}_k}\left[\eta_{k,r} \sum_{l=1}^p m_{k,r}^l \|\mathbf{u}_k - \hat{\mathbf{u}}_k^l\|_2\right] \le \sqrt{pC_1}$$

*Proof.* Using reciprocal moments, we have

$$\mathbb{E}_{\mathbf{M}_k}[\eta_{k,r}] = P\left(N_{k,r}^\perp = 1\right)\mathbb{E}_{\mathbf{M}_k}\left[\eta_{k,r} \mid N_{k,r}^\perp = 1\right] = \frac{\theta^2}{p\xi}$$

To start, we compute that for $r \ne r'$. Using the independence of $m_{k,r}^l$ and $m_{k,r'}^l$, we have

$$\mathbb{E}_{\mathbf{M}_k}\left[\eta_{k,r}^2 \sum_{l=1}^p m_{k,r}^l(\xi - m_{k,r'}^l)^2\right] = \sum_{l=1}^p \mathbb{E}_{\mathbf{M}_k}\left[\eta_{k,r}^2 m_{k,r}(\xi - m_{k,r'})^2\right]$$

$$= \sum_{l=1}^p \mathbb{E}_{\mathbf{M}_k}\left[\eta_{k,r}^2 m_{k,r}^l\right]\mathbb{E}_{\mathbf{M}_k}\left[(\xi - m_{k,r'}^l)^2\right]$$

$$= \xi(1-\xi)\mathbb{E}_{\mathbf{M}_k}\left[\eta_{k,r}^2 \sum_{l=1}^p m_{k,r}^l\right]$$

$$= \xi(1-\xi)\mathbb{E}_{\mathbf{M}_k}[\eta_{k,r}]$$

For $r = r'$, we use the idempotent

$$\mathbb{E}_{\mathbf{M}_k}\left[\eta_{k,r}^2 \sum_{l=1}^p m_{k,r}^l(\xi - m_{k,r}^l)^2\right] = (1-\xi)^2\mathbb{E}_{\mathbf{M}_k}\left[\eta_{k,r}^2 \sum_{l=1}^p m_{k,r}^l\right]$$

$$= (1-\xi)^2\mathbb{E}_{\mathbf{M}_k}[\eta_{k,r}]$$

Therefore

$$\mathbb{E}_{\mathbf{M}_k}\left[\eta_{k,r}^2\sum_{l=1}^p m_{k,r}^l\left(u_k^{(i)}-\hat{u}_k^{l(i)}\right)^2\right] \le \frac{1}{m}\mathbb{E}_{\mathbf{M}_k}\left[\eta_{k,r}^2\sum_{l=1}^p m_{k,r}^l\left(\sum_{r'=1}^m a_r(\xi-m_{k,r'}^l)\sigma(\langle\mathbf{w}_{k,r'},\mathbf{x}_i\rangle)\right)^2\right]$$

$$\le \frac{1}{m}\mathbb{E}_{\mathbf{M}_k}\left[\eta_{k,r}^2\sum_{l=1}^p m_{k,r}^l\sum_{r'=1}^m(\xi-m_{k,r'}^l)^2\sigma(\langle\mathbf{w}_{k,r'},\mathbf{x}_i\rangle)^2\right]$$

$$\le \frac{1}{m}\sum_{r'=1}^m\mathbb{E}_{\mathbf{M}_k}\left[\eta_{k,r}^2\sum_{l=1}^p m_{k,r}^l(\xi-m_{k,r'}^l)^2\right]\sigma(\langle\mathbf{w}_{k,r'},\mathbf{x}_i\rangle)^2$$

$$\le \frac{\xi(1-\xi)}{m}\mathbb{E}_{\mathbf{M}_k}[\eta_{k,r}]\sum_{r'=1}^m\sigma(\langle\mathbf{w}_{k,r'},\mathbf{x}_i\rangle)^2+$$

$$\frac{(1-\xi)(1-2\xi)}{m}\mathbb{E}_{\mathbf{M}_k}[\eta_{k,r}]\,\sigma(\langle\mathbf{w}_{k,r},\mathbf{x}_i\rangle)^2$$

$$\le \frac{\theta^2(1-\xi)}{mp}\sum_{r'=1}^m\langle\mathbf{w}_{k,r'},\mathbf{x}_i\rangle^2+\frac{(1-\xi)^2\theta^2}{mp\xi}\langle\mathbf{w}_{k,r},\mathbf{x}_i\rangle^2$$

$$\le \frac{2\theta^2(1-\xi)\kappa^2}{p}+\frac{2\theta^2(1-\xi)^2\kappa^2}{mp\xi}$$

$$\le \frac{4\theta^2(1-\xi)\kappa^2}{p}$$

where in the last inequality we use $m\ge\xi^{-1}$. Thus, we have

$$\mathbb{E}_{\mathbf{M}_k}\left[\eta_{k,r}^2\sum_{l=1}^p m_{k,r}^l\|\mathbf{u}_k-\hat{\mathbf{u}}_k^l\|_2^2\right]=\sum_{i=1}^n\mathbb{E}_{\mathbf{M}_k}\left[\eta_{k,r}^2\sum_{l=1}^p m_{k,r}^l\left(u_k^{(i)}-\hat{u}_k^{l(i)}\right)^2\right]\le C_1$$

Also, we have

$$\mathbb{E}_{\mathbf{M}_k}\left[\eta_{k,r}\sum_{l=1}^p m_{k,r}^l\|\mathbf{u}_k-\hat{\mathbf{u}}_k^l\|_2\right]\le\mathbb{E}_{\mathbf{M}_k}\left[\left(\eta_{k,r}\sum_{l=1}^p m_{k,r}^l\|\mathbf{u}_k-\hat{\mathbf{u}}_k^l\|_2\right)^2\right]^{\frac{1}{2}}$$

$$\le\sqrt{p}\mathbb{E}_{\mathbf{M}_k}\left[\eta_{k,r}^2\sum_{l=1}^p m_{k,r}^l\|\mathbf{u}_k-\hat{\mathbf{u}}_k^l\|_2^2\right]^{\frac{1}{2}}$$

Plugging in the previous bound gives the desired result. □

**Lemma 25.** *If for some $R>0$ and all $r\in[m]$ the initialization satisfies*

$$\sum_{r=1}^m\langle\mathbf{w}_{0,r},\mathbf{x}_i\rangle^2\le 2mn\kappa^2-mnR^2$$

*and for all $r\in[m]$, it holds that $\|\mathbf{w}_{k,r}-\mathbf{w}_{0,r}\|_2\le R$. Then we have*

$$\mathbb{E}_{\mathbf{M}_k}\left[\|\mathbf{u}_k-\hat{\mathbf{u}}_k^l\|_2^2\right]\le 4\xi(1-\xi)n\kappa^2$$

*Proof.* To start, we have

$$\mathbb{E}_{\mathbf{M}_k}\left[\left(u_k^{(i)} - \hat{u}_k^{l(i)}\right)^2\right] = \frac{1}{m}\mathbb{E}_{\mathbf{M}_k}\left[\left(\sum_{r=1}^m a_r(\xi - m_{k,r}^l)\sigma(\langle \mathbf{w}_{k,r}, \mathbf{x}_i \rangle)\right)^2\right]$$

$$\leq \frac{1}{m}\sum_{r=1}^m \mathbb{E}_{\mathbf{M}_k}\left[(\xi - m_{k,r}^l)^2\right]\sigma(\langle \mathbf{w}_{k,r}, \mathbf{x}_i \rangle)^2$$

$$\leq \frac{\xi(1-\xi)}{m}\sum_{r=1}^m \langle \mathbf{w}_{k,r}, \mathbf{x}_i \rangle^2$$

$$\leq \frac{2\xi(1-\xi)}{m}\sum_{r=1}^m \langle \mathbf{w}_{0,r}, \mathbf{x}_i \rangle^2 + 2\xi(1-\xi)R^2$$

$$\leq 4\xi(1-\xi)\kappa^2$$

Therefore,

$$\mathbb{E}_{\mathbf{M}_k}\left[\|\mathbf{u}_k - \hat{\mathbf{u}}_k^l\|_2^2\right] = \sum_{i=1}^n \mathbb{E}_{\mathbf{M}_k}\left[\left(u_k^{(i)} - \hat{u}_k^{l(i)}\right)^2\right] \leq 4\xi(1-\xi)n\kappa^2$$

$\square$

**Lemma 26.** *Assume that for all $i \in [n]$, $y_i$ satisfies $|y_i| \leq C - 1$ for some $C \geq 1$. Then, we have*

$$\mathbb{E}_{\mathbf{W}_0,\mathbf{a}}\left[\|\mathbf{y} - \mathbf{u}_0\|_2^2\right] \leq C^2 n$$

*Proof.* It is easy to see that $\mathbb{E}_{\mathbf{W}_0,\mathbf{a}}\left[u_0^{(i)}\right] = 0$. Now, note that

$$\mathbb{E}_{\mathbf{W}_0,\mathbf{a}}\left[\left(u_0^{(i)}\right)^2\right] = \frac{\xi^2}{m}\mathbb{E}_{\mathbf{W}_0,\mathbf{a}}\left[\left(\sum_{r=1}^m a_r\sigma(\langle \mathbf{w}_{0,r}, \mathbf{x}_i \rangle)\right)^2\right]$$

$$= \frac{\xi^2}{m}\sum_{r=1}^m \mathbb{E}_{\mathbf{W}_0}\left[\langle \mathbf{w}_{0,r}, \mathbf{x}_i \rangle^2\right]$$

$$= \frac{\xi^2}{m}\sum_{r=1}^m \mathbb{E}_{\mathbf{W}_0}\left[\left(\sum_{d'=1}^d w_{0,r,d'}x_{i,d'}\right)^2\right]$$

$$= \frac{\xi^2}{m}\sum_{r=1}^m\sum_{d'=1}^d \mathbb{E}_{\mathbf{W}_0}\left[w_{0,r,d'}^2 x_{i,d'}^2\right]$$

$$= \frac{\xi^2}{m}\sum_{r=1}^m\sum_{d'=1}^d x_{i,d'}^2$$

$$= \xi^2$$

Therefore,

$$\mathbb{E}_{\mathbf{W}_0,\mathbf{a}}\left[\left(y_i - u_0^{(i)}\right)^2\right] = y_i^2 - 2y_i\mathbb{E}_{\mathbf{W}_0,\mathbf{a}}\left[u_0^{(i)}\right] + \mathbb{E}_{\mathbf{W}_0,\mathbf{a}}\left[\left(u_0^{(i)}\right)^2\right] = y_i^2 + \xi^2$$

Thus,

$$\mathbb{E}_{\mathbf{W}_0,\mathbf{a}}\left[\|\mathbf{y} - \mathbf{u}_0\|_2^2\right] = \sum_{i=1}^n \mathbb{E}_{\mathbf{W}_0,\mathbf{a}}\left[\left(y_i - u_0^{(i)}\right)^2\right] = \sum_{i=1}^n y_i^2 + \xi^2 n \leq C^2 n$$

$\square$

