# OpenReview forum: "On the Convergence of Shallow Neural Network Training with Randomly Masked Neurons"
_TMLR — Accepted by TMLR_

### Review · Reviewer_M1SE · 2022-06-15

**Summary Of Contributions:**

This paper focuses on analyzing training error and establishing convergence rates when training a perceptron with a single-hidden layer and multiple randomly masked subnetworks. Masks are randomly sampled at each iteration. The authors used techniques based on the NTK analysis. However, unlike standard NTK analysis, they handled the additional randomness due to mask sampling. They defined a new notion of masked-NTK induced by another mask and show that it becomes positive definite under some assumptions in Theorem 2. Then they consider masked drawn from Bernoulli and Categorical distributions and established convergence guarantees when the number of hidden neurons is sufficiently large and the step size is sufficiently small. They provided results on important and interesting special cases including standard Dropout, Multi-Sample Dropout, and  Multi-Worker Independent Subnetwork Training.

**Requested Changes:**

- Problem formulation: all compute nodes have access to the entire training samples. It is important that the authors further motivate and provide practical settings that fit to the model considered in this paper.

- It seems underlying distributions over which masks are sampled directly impact training and maybe test error. The authors considered two well-known distributions, which is certainly interesting. Beyond those distributions, are there principles to design optimal masks?

- How is the aggregation of weights  performed? Does it depend on the construction of masks and the number of local updates?

- Page 3 "Distributed ML .... communication inefficient" This is partially true. If the tuning/convergence of IST takes a lot of time, the overall speedup gain may disappear.

- Loss at iteration $k$ $L_k$: This loss is in the form of $||\mathrm{E}[\dots] ||^2$ where the expectation insider inside the quadratic function is over randomess of masks. It is more common to analyze the expected loss, i.e.,  $\mathrm{E}[||\dots ||^2]$. Is it possible to extend the analysis?

- Theorem 2: This boundedness assumption 2 is a bit strange. Could you please clarify when this assumption holds in general?

- Theorem 3: The second term in $B_1$ increases with $n$. Is it possible to improve the bound? In addition, Theorem 3 shows that increasing the number of local steps does not help. Why do they authors consider multiple local steps?

- Section 5: It is not clear how the variance is shown in Figure 2a.

Minor comments:

- $\eta$ and $\eta_{k,r}$ in Algorithm 1 should be defined.

- Please specify the precise definition of co-alignment in Assumption 1.

-  $\xi$ should be red in two equations right after Theorem 1.

- Corollary 1: Please provide citations for the convergence rate under $\zeta=1$. Do other work also require  $m=\Omega(n^5)$?

**Strengths And Weaknesses:**

Strengths:

This paper is sound and provides a rigorous analysis for a challenging problem. The authors clearly defined the problem and training algorithm. The assumptions are stated explicitly and notations are mostly clear. The paper is mostly well-written. In particular, proof sketches are helpful to help the readers get the main ideas. The related work are mostly provided though there is room for improvement.

Weakness:

The setting should be better motivated, in particular, the design of masks and the fact that all workers have access to the entire training samples. The sufficient number of neurons for convergence is another concern. It is also unclear whether local updates are useful given the convergence analysis. Finally, related work and experimental results can be improved.

---

> ### Author Response · Authors · 2022-07-06
> **Response to Reviewer M1SE**
>
> We thank the reviewer for his/her thoughtful comments. We respond below to the two weaknesses raised, the requested changes, as well as minor comments. The changes/edits based on the comments of this review are indicated in orange color in the main text:
>
> **Respond to Weakness**
>
> *The setting should be better motivated, in particular, the design of masks and the fact that all workers have access to the entire training samples*
>
> We comment on this concern in our response to requested changes (1),(2), and (4) below
>
> *The sufficient number of neurons for convergence is another concern.*
>
> The level of over-parameterization is typical in previous work. In particular, [1,2] assumes that the number of hidden neurons is $\Omega(n^6)$ and $\Omega(n^4)$ respectively. Our work is based on their analysis and does not require larger over-parameterization: when $\xi=1$ and choosing $\kappa=1$, the over-parameterization in our result also reduces to $\Omega(n^4)$. We also would like to point out that we do not claim any improvement on the required number of hidden neurons, but extend the analysis in previous work to a more general training framework.
>
> *It is also unclear whether local updates are useful given the convergence analysis.*
>
> We commented on the effect of local updates in the response to the requested change (7) below.
>
> *Finally, related work and experimental results can be improved.*
>
> We have improved the Related Works and the experiments in the updated version.
>
> **Respond to Requested Changes (1)-(5)**
>
> (1) We do admit that, for the case of IST, it rarely appears in the practical scenario that every node has access to all the data. However, IST is proposed as a data/model parallel training algorithm that works under both partial and full data access. While analyzing the case of partial data access is closer to a practical setting, our work focuses on the technical difficulty brought up by the model partition process, and thus makes the simplified assumption of full data access. We believe that our analysis can be extended to partial data access by combining the techniques of [3], but to keep the proof from being too lengthy we did not include that case. Lastly, for the case of dropout and multi-sample dropout, each subnetwork will indeed have full data access.
>
> (2) First, we would like to point out that the main focus of our work is on the use/choice of random masks over hidden layers, as this setting generalizes current algorithms. Our analysis is centered around the characterization of masked-NTK and the subnetwork local training dynamics. The theoretical results require i) the sampling of masks in each iteration to be independent of other iterations, and ii) the sampling of masks for each neuron to be independent of other neurons. To make the analysis concrete and to endow it with a more practical setting, we consider the Bernoulli mask and the categorical mask.
>
> (3) We have added clarification of the aggregation step at the end of section 3. It depends on the construction of the masks, and not on the number of local updates. Overall, the number of local iterations could lead to faster convergence, based on our analysis.
>
> (4) We thank the reviewer for pointing this out. The efficiency of distributed training indeed depends on both the communication cost and the convergence time. In the case of low communication cost, algorithms that enjoy faster convergence achieve better efficiency. However, for training models that require high computation/communication costs, distributed training algorithms that reduce the communication rounds achieve a better overall (wall-clock time) speedup. This is observed empirically in previous works [4,5].
>
> (5) We use the definition of $L_k$ only to shed light on the current NTK-based analysis technique for neural networks. While we point out that the expectation of the subnetwork is the same as the whole network, in $L_k$ we are focusing on the behavior of the whole network instead of the expected loss on the subnetwork. In Theorems 2,3, and 4, as well as in the corollaries, the convergence is based on $E[||\cdot||]$.
>
> **References**
>
> [1] Simon S Du, Xiyu Zhai, Barnabas Poczos, and Aarti Singh. Gradient descent provably optimizes over-parameterized neural networks. In *International Conference on Learning Representations*, 2018b.
>
> [2] Zhao Song and Xin Yang. Quadratic suffices for over-parametrization via matrix chernoff bound, 2020.
>
> [3] Baihe Huang, Xiaoxiao Li, Zhao Song, and Xin Yang. Fl-ntk: A neural tangent kernel-based framework for federated learning convergence analysis, 2021.
>
> [4] Chen Dun, Cameron R. Wolfe, Christopher M. Jermaine, and Anastasios Kyrillidis. Resist: Layer-wise decomposition of resnets for distributed training, 2021.
>
> [5] Cameron R Wolfe, Jingkang Yang, Arindam Chowdhury, Chen Dun, Artun Bayer, Santiago Segarra, and Anastasios Kyrillidis. GIST: Distributed training for large-scale graph convolutional networks. *arXiv preprint arXiv:2102.10424, 2021.*

---

> > ### Author Response · Authors · 2022-07-06
> > **Further Response to Reviewer M1SE**
> >
> > **Respond to Requested Changes (6)-(8)**
> >
> > (6) The boundedness is typical in previous work [1,2,3], and is shown to hold when the neural network is sufficiently over-parameterized. We used this in Theorems 2, 3, and 4. In Theorem 2, this assumption is satisfied by equation (3), while in Theorems 3 and 4, we explicitly showed that the boundedness holds.
> >
> > (7) The second term does scale with n. However, $\kappa$ also appears in that term. This term can be arbitrarily small, as long as the initialization scale is small. In the Corollaries of Theorem 3, we choose $\kappa = \frac{1}{\sqrt{n}}$ , and this choice brings the second term to the constant scale $O(1)$. Moreover, increasing the number of local steps does increase $B_1$. Indeed the upper bound on $\eta$ scales inversely with $\tau$. However, fixing an $\eta$ that is small enough and increasing $\tau$ also leads to a further decrease in the MSE. In our work, we only consider constant step size for local training, and we believe that the dependence of $\eta$ on $\tau$ would be eliminated if the same analysis is carried out on a decaying stepsize $\eta_t$ = $\frac{\eta}{t+1}$. For the brevity of our proof, we did not include this case.
> >
> > (8) We are sorry for the confusion caused here. We did plot the variance in the figure. However, since the error decreases in a rather stable behavior, the variance is too small to be observed. We have also added the clarification in the updated version of the paper.
> >
> > **Respond to Minor Comments**
> >
> > (1) We have added the clarification in the updated version.
> >
> > (2) We have added the clarification of co-alignment in Assumption 1. To be specific, we mean that the two vectors are not parallel, i.e., there does not exist $\zeta\in R$ such that $x_i\neq \zeta x_j$ . This assumption is the same as in [1,2].
> >
> > (3) Thank you for the advice. We have fixed this in the updated version.
> >
> > (4) We have provided the citation for the convergence rate under $\xi=1$ in the updated version. In particular, [1] requires $m = \Omega(n^6)$ and [2] requires $m = \Omega(n^4)$. However, notice that when $\xi=1$, we can choose $\kappa=1$ and our over-parameterization requirement reduces to $m = \Omega(n^4)$, as in [2].
> >
> > **References**
> >
> > [1] Simon S Du, Xiyu Zhai, Barnabas Poczos, and Aarti Singh. Gradient descent provably optimizes over-parameterized neural networks. In *International Conference on Learning Representations*, 2018b.
> >
> > [2] Zhao Song and Xin Yang. Quadratic suffices for over-parametrization via matrix chernoff bound, 2020.
> >
> > [3] Samet Oymak and Mahdi Soltanolkotabi. Toward moderate overparameterization: Global convergence guarantees for training shallow neural networks. *IEEE Journal on Selected Areas in Information Theory*, 1(1):84–105, 2020.

---

> > > ### Comment · Reviewer_M1SE · 2022-07-08
> > > **Comment after revision**
> > >
> > > I would like to thank the authors for their response and careful revision. The revised version addresses the reviewer's concerns. I think in the current version, the claims match the contributions, which is the main acceptance criterion of TMLR.

---

### Review · Reviewer_uQXy · 2022-06-15

**Summary Of Contributions:**

The authors study different algorithms that iteratively train subnetworks of a neural network as a way of training the full network. Examples of this include Dropout and the Independent Subnet Training protocol. Using the neural tangent kernel approach, the main contributions of the analysis are to prove convergence guarantees resulting in linear convergence of the training loss. The other contributions are in service of this main result but are of some independent interest also. For example, the authors identify the NTK corresponding to the subnetworks and show it is close to the infinite-width limit. They also derive the bias in the surrogate gradient compared to the ideal gradient, and show that this bias decreases to zero as the number of subnetworks grows.

**Broader Impact Concerns:**

None.

**Requested Changes:**

1. Can the authors discuss the differences between their work and Mohtashami et al., 2021? Is the only difference in the differentiability of the activation function? Is there any difference in technique?

2. Can the authors comment on the scale of \kappa? I believe the standard NTK initialization would set \kappa=1/sqrt(d), but the discussion around B_1 indicates \kappa=1/sqrt(n) is the relevant scale.

3. As mentioned above, the experimental section should include some analysis of the effect of width of the results. In particular, there should be finite-size effects from the width.

4. Another claim the paper makes is that categorical sampling has a more desirable error term. The paper would be improved by an experiment supporting this claim.

5. What is the purpose of using a pre-trained model as a feature extractor in the experimental section?

**Minor**
- Is “error region” a standard term? It was not clear to me what this meant in the abstract.
- In Fig. 1(c), f_{m_k^1} should be f_{m_k^2}
- Page 4, “minima” is already plural
- a_r should be a_m in the vector on page 5
- “right hand side” should be hyphenated on page 8
- In Eq 7, why is there an expectation over M_{k-1}? I thought u_k and y did not depend on the mask.
- I think the y-axis in Fig. 2(a) is incorrect and should just be Training Error, as the axis’ ticks are already in log-scale.
- "Conbine” -> Combine on page 21

**Strengths And Weaknesses:**

**Strengths**

The paper’s primary contributions are Thms. 3 and 4, which involve some generalization of the standard NTK analysis. The authors do a good job of explaining the steps involved in this proof at a high level and discussing the different terms that emerge in the results. For example, the discussion of B_1 after Thm. 3 is appreciated.

In summary, the main claims outlined in the contributions are supported by the theorems. Although, since the bias of the surrogate gradient is identified in the contributions section, it might make sense to move some of the analysis of this into the main text from the supplement.


**Weaknesses**

One point of confusion I have is whether the paper investigates Dropout in a regime that is practically relevant. Setting aside the assumptions required for the NTK analysis, it seems like showing that the surrogate gradient concentrates around the ideal gradient implies that the updates to the weights are the same with and without Dropout. However, Dropout is commonly used as a regularizer, which would require that the weights learned depend on the Dropout rate. This concern does not matter for IST, where the motivation is to reduce computation time.

The experimental section could do a better job of supporting the paper’s main contributions. While the results are consistent with Thm. 3, it would be better to understand the assumptions of Thms. 3 and 4.

---

> ### Author Response · Authors · 2022-07-06
> **Response to Reviewer uQXy**
>
> We thank the reviewer for his/her thoughtful comments. We respond below to the two weaknesses raised, the requested changes, as well as some minor comments mentioned in the review. The changes/edits based on the comments of this review are indicated with purple color in the main text:
>
> **Respond to Weakness**
>
> *...it seems like showing that the surrogate gradient concentrates around the ideal gradient implies that the updates to the weights are the same with and without Dropout...*
>
> By “concentrates around the ideal gradient”, we do not mean that the expectation of the surrogate gradient equals the ideal gradient. Instead, we mean that the expectation of the surrogate gradient lies near the ideal gradient. We are sorry for the ambiguity here, and we have edited the paper to avoid further confusion. We would like to highlight that the fact that the expectation of the updates of a gradient-based algorithm equals the ideal gradient does not prevent it from having the regularization effect. One example is SGD, where in the analysis the expectation of the noise is assumed to be zero, but the algorithm still shows an implicit regularization effect. Although regularization is not shown in our work, the analysis does not exclude any regularization behavior of the Dropout technique.
>
> *...it would be better to understand the assumptions of Thms. 3 and 4....*
>
> We have added a plot in the updated version showing how the over-parameterization affects the error term. In particular, Theorem 3, implies that the error term decreases when we choose a smaller initialization scale $\kappa$, while a smaller $\kappa$ would require a larger over-parameterization. This is consistent with our new experiments as shown in Figure 2c: as we increase the number of hidden neurons and adjust the initialization scale, we observe that the training converges to a smaller error. Further, Figure 2d shows how the convergence rate changes, as we increase the number of subnetworks under the categorical mask assumption. We observe that the training error improvement decreases consistently across the first three global steps, as we increase the number of subnetworks. This corresponds to what we have shown in Theorem 4, where $1 - \gamma$ decreases, as we increase $p$. More details are included in the main text.
>
> **Respond to Requested Changes (1)-(3)**
>
> (1) We thank the reviewer for this comment. We have added a discussion on the difference between the two works in the Related Works section. In short, the main difference between our work and [1] is that, in the latter case, the authors consider a more general class of objectives with the assumption of differentiability. Yet, some assumptions made for both the objective, such as smoothness, and the algorithm, such as bounded perturbation, are not straightforward to check, especially under the setting of optimizing neural networks, This fact and the fact that the analysis followed is radically different, constitutes our work different than theirs, and a different contribution in this field.
>
> (2) Our initialization choice is actually quite standard. First, we point out that using the $\frac{1}{\sqrt{m}}$ scaling factor in the neural network function is standard in previous literature [2,3,4]. Amongst these literatures, the standard NTK [2] and its following work on finite width shallow neural network [3] uses $\kappa=1$, while another work [4] uses $\kappa=\frac{1}{\sqrt{n}}$. The analysis of our work does not depend on a specific κ. However, choosing $\kappa=\frac{1}{\sqrt{n}}$, as in [4] makes our results easier to interpret.
>
> (3) Since our work is based on [3,5], the reader can refer to their work for some finite-width effect experiments. What is unique in our theorem is the error term in the convergence result of Theorem 3. For this, we have added an experiment on how changing the width of the neural network affects this error term; see e.g., the new plot in Figure 2c.
>
> **Reference**
>
> [1] Amirkeivan Mohtashami, Martin Jaggi, and Sebastian U Stich. Simultaneous training of partially masked neural networks. *arXiv preprint arXiv:2106.08895, 2021.*
>
> [2] Arthur Jacot, Franck Gabriel, and Clement Hongler. Neural tangent kernel: convergence and generalization in neural networks. In *Proceedings of the 32nd International Conference on Neural Information Processing Systems*, pp. 8580–8589, 2018.
>
> [3] Simon S Du, Xiyu Zhai, Barnabas Poczos, and Aarti Singh. Gradient descent provably optimizes over-parameterized neural networks. In *International Conference on Learning Representations*, 2018b.
>
> [4] Sanjeev Arora, Simon S. Du, Wei Hu, Zhiyuan Li, and Ruosong Wang. Fine-grained analysis of optimization and generalization for overparameterized two-layer neural networks, 2019.
>
> [5] Zhao Song and Xin Yang. Quadratic suffices for over-parametrization via matrix chernoff bound, 2020.

---

> > ### Author Response · Authors · 2022-07-06
> > **Further Response to Reviewer uQXy**
> >
> > **Respond to Requested Changes (4), (5)**
> >
> > (4) We thank the reviewer for focusing more on this point. Based on this comment, we have revisited the proof of Theorem 4, which we have edited, and now it leads to a slightly different conclusion. In particular, the decrease in training error in each iteration becomes smaller, as we increase the number of subnetworks. To empirically validate this, we have added an experimental result showing how the loss improvement changes, as we change the number of subnetworks. Overall, though, this additional edit does not change the narrative of the paper, which is to provide theoretical justification for why subnetwork training works (for this case, why IST works).
> >
> > (5) The purpose of using a pre-trained model for the experiments is to find a task that is representative of the current machine learning tasks and can be easily solved using an MLP. We have added a corresponding discussion in the updated version of the paper. E.g., the recent work by [1], proposes the use of a library of pre-trained networks to extract useful features, which later on are processed by adding the topmost layers used for classification. This results in a procedure that takes an image, creates an embedding and then uses that embedding to build a classifier by feeding it into a multi-layer perceptron with single/multiple hidden layers.
> >
> > **Respond to Minors**
> >
> > For bullet points (1)-(5) and (7)-(9), we have resolved them in the updated version of the paper. For bullet point (6), we take the expectation over $M_{k-1}$ because $u_k$ depends on $M_{k-1}$. In particular, $W_k$ is computed by aggregating the results of $W_{k-1,\tau}^l$, and $W_{k-1,\tau}^l$'s are weights trained on subnetwork in the $(k-1)$th global iteration; thus, these weights depend on the mask $M_{k-1}$. Therefore $u_k$ also depends on $M_{k-1}$.
> >
> > **References**
> >
> > [1] Arkabandhu Chowdhury, Mingchao Jiang, Swarat Chaudhuri, and Chris Jermaine. Few-shot image classification: Just use a library of pre-trained feature extractors and a simple classifier. In *Proceedings of the IEEE/CVF International Conference on Computer Vision*, pp. 9445–9454, 2021.

---

> > > ### Comment · Reviewer_uQXy · 2022-07-22
> > > **Response to authors**
> > >
> > > I am grateful to the authors for their detailed response to my questions. I feel that this in addition to the revision of the manuscript addresses my concerns with the paper. I believe the listed contributions are supported by the paper.

---

### Review · Reviewer_1me7 · 2022-06-23

**Summary Of Contributions:**

This paper studies the convergence of two-layer neural network training with randomly masked neurons. In particular, several contributions are claimed by the authors:

This paper shows that with randomly masked neurons, the training loss decreases at a linear rate to a certain error region level.

The developed convergence guarantees imply the convergence rate for a class of empirical training approaches, such as dropout, multi-sample dropout, and multi-worker IST.

The authors further compare the convergence performance when using categorical masks and Bernoulli masks and show that using  the categorical mask can lead to a more desirable error term.

**Broader Impact Concerns:**

No broader impact concerns.

**Requested Changes:**

The authors may need to make the modification according to the aforementioned weakness. In particular, I would like to see

1. More rigorous motivations.
2. Why study the convergence in the NTK regime.
3. If it's possible to remove/improve the dependency on $K$.
4. Discussion on the generalization.


**Strengths And Weaknesses:**

Strength:

This paper develops a theoretical analysis for a unified training scheme, which can be adapted to many practical scenarios.

Weakness:

1. More motivations are needed to support the goal of this work. Currently, the authors only mention that regularization techniques and distributed training methods are widely studied in recent years. However, this may not be enough to motivate the study of convergence rates of training neural networks with random masks (in the NTK regime). The authors may need to (1) mention the advantage/improvements of using distributed training methods; (2) summarize the existing convergence results (not limited to NN models); and (3) point out why studying convergence is important in the introduction section (before the stated question).

2. I do not quite see whether studying the convergence in the NTK regime is a good approach for training the random masked network (or dropout regularization). Take the dropout regularization as an example, the regularization typically encourages the algorithm to find a solution with lower complexity from a global perspective, while NTK analysis requires the solution to be extremely close to the initialization, which contradicts the spirit of the regularization.

3. I am also not satisfied with the overparameterization condition stated in Theorem 3. Particularly, the number of hidden neurons needs to be larger than at least $O(K)$, where $K$ is the number of outer iterations. This implies the theory cannot support the case that the neural network width is fixed while the training iteration number becomes extremely large.

4. I believe the benefit of using dropout or other random masking approaches is to get a better "generalization" ability. The authors may also need to discuss the generalization performance of the solution found by using random masked neurons. Otherwise, as shown in Corollary 1, the optimization guarantee of using dropout is definitely worse than that without using dropout.

---

> ### Author Response · Authors · 2022-07-06
> **Response to Reviewer 1me7**
>
> We thank the reviewer for his/her thoughtful comments. We respond below to the weaknesses raised, the requested changes, as well as some minor comments mentioned in the review. The changes/edits based on the comments of this review are indicated with olive color in the main text:
>
> (1) **“More motivations are needed to support the goal of this work.”** For this point, we have added a discussion and cited related works in our edited version. In particular, we have added text that highlights the value of using distributed methods. Regarding existing convergence results, we added a paragraph in the ”Related Works” section, focusing on current works that analyze convergence in distributed training. Regarding why studying convergence rates is an important topic, such comment questions the optimization research overall: Studying achievable convergence rates is one of the main areas in optimization research. Convergence rate results signify which algorithms are faster than others, and when/why someone would use one algorithm over another. Focusing on our setting, the problem we consider is a non-trivial, non-convex setting, which makes it an interesting case by itself. Many recent works in optimization and machine learning theory focus on such open questions, and the current work goes in this direction.
>
> (2) **“Whether studying the convergence in the NTK regime is a good approach”**: We choose the NTK-based technique to analyze the convergence because it is one of the most widely-used techniques to analyze gradient-based algorithms on neural networks [2,3,4,5,6]. As we show in Theorem 2, the masked-NTK of the randomly sampled subnetworks has the nice property of concentrating around the NTK of the whole network that facilitates our analysis. The reviewer also pointed out that the small perturbation of the weights contradicts the intuition of the regularization behavior of dropout. Yet, most works that analyze the generalization behavior of gradient descent consider bounding the Rademacher complexity [1,4,7], using the property that the parameters –learned by gradient descent– stay in a small neighborhood of the initialization. Finally, we point out that the majority of existing techniques, including those going beyond NTK, analyze the convergence of training neural networks based on small weight perturbation properties from initialization [7,8,10].
>
> (3) **“Not satisfied with the overparameterization condition stated in Theorem 3”**: The over-parameterization on $K$ is quite common in previous work. Please see the added remark below Theorem 3 in our edited version of the paper. In particular, under a fixed $m$, the convergence is guaranteed for a bounded number of iterations. This is not a concern in general since to guarantee ε small training error we only need $K$ to be $\frac{\log \epsilon^{-1} + \log n}{\log (1 - O(\eta\theta\tau\lambda_0)^{-1})}$. This is termed early-stopping and is used in previous literature [6,9].
>
> (4) **“Get a better ”generalization” ability”**: We thank the reviewer for this point. Indeed having a generalization guarantee for dropout is ideal, since it is proposed as a regularization technique. However, the focus of our work is to show the convergence of using subnetwork training to optimize a training objective. Thus, in our work, we view dropout as an optimization method. We certainly believe that studying the generalization property is interesting, but we believe that showing the convergence of training error is a necessary step toward this interesting and challenging task. Studying the generalization property is considered as our future work, which may stand as another paper by itself.

---

> > ### Author Response · Authors · 2022-07-06
> > **References for Response to Reviewer 1me7**
> >
> > **References**
> >
> > [1] Yuan Cao and Quanquan Gu. Generalization error bounds of gradient descent for learning over-parameterized deep relu networks, 2019. URL https://arxiv.org/abs/1902.01384.
> >
> > [2] Simon S. Du, Jason D. Lee, Haochuan Li, Liwei Wang, and Xiyu Zhai. Gradient descent finds global minima of deep neural networks, 2018a. URL https://arxiv.org/abs/1811.03804.
> >
> > [3] Simon S Du, Xiyu Zhai, Barnabas Poczos, and Aarti Singh. Gradient descent provably optimizes over-parameterized neural networks. In *International Conference on Learning Representations*, 2018b.
> >
> > [4] Sanjeev Arora, Simon S. Du, Wei Hu, Zhiyuan Li, and Ruosong Wang. Fine-grained analysis of optimization and generalization for overparameterized two-layer neural networks, 2019.
> >
> > [5] Zhao Song and Xin Yang. Quadratic suffices for over-parametrization via matrix chernoff bound, 2020.
> >
> > [6] Lili Su and Pengkun Yang. On learning over-parameterized neural networks: A functional approximation perspective, 2019.
> >
> > [7] Ziwei Ji and Matus Telgarsky. Polylogarithmic width suffices for gradient descent to achieve arbitrarily small test error with shallow relu networks, 2020.
> >
> > [8] Chaehwan Song, Ali Ramezani-Kebrya, Thomas Pethick, Armin Eftekhari, and Volkan Cevher. Subquadratic overparameterization for shallow neural networks, 2021. URL https://arxiv.org/abs/2111.01875.
> >
> > [9] Zeyuan Allen-Zhu, Yuanzhi Li, and Yingyu Liang. Learning and generalization in overparameterized neural networks, going beyond two layers, 2018. URL https://arxiv.org/abs/1811.04918.
> >
> > [10] Quynh Nguyen. On the proof of global convergence of gradient descent for deep relu networks with linear widths, 2021. URL https://arxiv.org/abs/2101.09612.

---

### Decision · Action_Editors · 2022-08-05

**Recommendation:** Accept with minor revision

**Comment:**

Dear Authors,

thank you submitting your manuscript, your response and revision to the reviewers questions.  After your response, the reviewers have all agreed that, comparing the papers contents to its claims, it should be accepted, and I agree.

May I only ask one very minor modifications. Two of the reviewers raised questions about if the assumptions needed for the NTK analysis are well suited to the setting of dropout/random masking. I find the authors response satisfactory, but I would also encourage the authors to include in the abstract a small sentence on the assumptions under which their analysis holds. For instance, perhaps a comment on the need for having a small weight change for the NTK analysis to hold. Or something else similar, just in the spirit of further matching claims of the paper to the results.

The reviewers also asked for more background material and better motivations as relating to the distributed setting. But with the much increased "Related Work" section, they agree that it is now clear and satisfactory.